# FedX: Federated Learning for Compositional Pairwise Risk Optimization

## Abstract

In this paper, we tackle a novel federated learning (FL) problem for optimizing a family of compositional pairwise risks, to which no existing FL algorithms are applicable. In particular, the objective has the form of $\mathbb{E}_{\mathbf{z} \sim \mathcal{S}_1} f(\mathbb{E}_{\mathbf{z}' \sim \mathcal{S}_2} \ell(\mathbf{w}; \mathbf{z}, \mathbf{z}'))$, where two sets of data $\mathcal{S}_1, \mathcal{S}_2$ are distributed over multiple machines, $\ell(\cdot; \cdot, \cdot)$ is a pairwise loss that only depends on the prediction outputs of the input data pairs $(\mathbf{z}, \mathbf{z}')$, and $f(\cdot)$ is possibly a non-linear non-convex function. This problem has important applications in machine learning, e.g., AUROC maximization with a pairwise loss, and partial AUROC maximization with a compositional loss. The challenges for designing an FL algorithm lie in the non-decomposability of the objective over multiple machines and the interdependency between different machines. We propose two provable FL algorithms (FedX) for handling linear and nonlinear $f$, respectively. To address the challenges, we decouple the gradient's components with two types, namely active parts and lazy parts, where the *active* parts depend on local data that are computed with the local model and the *lazy* parts depend on other machines that are communicated/computed based on historical models and samples. We develop a novel theoretical analysis to combat the latency of the lazy parts and the interdependency between the local model parameters and the involved data for computing local gradient estimators. We establish both iteration and communication complexities and show that using the historical samples and models for computing the lazy parts do not degrade the complexities. We conduct empirical studies of FedX for deep AUROC and partial AUROC maximization, and demonstrate their performance compared with several baselines.

## 1 Introduction

This work is motivated by solving the following optimization problem arising in many ML applications in a **federated learning (FL)** setting:

$$\min_{\mathbf{w} \in \mathbb{R}^d} \frac{1}{|\mathcal{S}_1|} \sum_{\mathbf{z} \in \mathcal{S}_1} f\left( \underbrace{\frac{1}{|\mathcal{S}_2|} \sum_{\mathbf{z}' \in \mathcal{S}_2} \ell(\mathbf{w}; \mathbf{z}, \mathbf{z}')}_{g(\mathbf{w}; \mathbf{z}, \mathcal{S}_2)} \right), \tag{1}$$

where $\mathcal{S}_1$ and $\mathcal{S}_2$ denote two sets of data points that are distributed over many machines, $\mathbf{w}$ denotes the model parameter of a prediction function $h_{\mathbf{w}}(\cdot) \in \mathbb{R}^{d_o}$, $f(\cdot)$ is a deterministic function that could be linear or non-linear (possibly non-convex), and $\ell(\mathbf{w}; \mathbf{z}, \mathbf{z}') = \ell(h_{\mathbf{w}}(\mathbf{z}), h_{\mathbf{w}}(\mathbf{z}'))$ denotes a pairwise loss that only depends the prediction outputs of the input data $\mathbf{z}, \mathbf{z}'$. We refer to the above problem as compositional pairwise risk (CPR) minimization problem.

When $f$ is a linear function, the above problem is the classic pairwise loss minimization problem, which has applications in AUROC (AUC) maximization (Gao et al., 2013; Zhao et al., 2011; Gao & Zhou, 2015; Calders & Jaroszewicz, 2007; Charoenphakdee et al., 2019; Yang et al., 2021b), bipartite ranking (Cohen et al., 1997; Clémençon et al., 2008; Kotlowski et al., 2011; Dembczynski et al., 2012), distance metric learning (Radenović et al., 2016; Wu et al., 2017; Yang et al., 2021b). When $f$ is a non-linear function, the above problem is a special case of finite-sum coupled compositional optimization problem (Wang & Yang, 2022a), which has found applications in various performance measure optimization such as partial AUC maximization (Zhu et al., 2022), average precision maximization (Qi et al., 2021; Wang et al., 2022), NDCG maximization (Qiu et al., 2022), and p-norm

push optimization (Rudin, 2009; Wang & Yang, 2022a) and distance metric learning (Sohn, 2016). We provide details of some examples of CPR minimization applications in Appendix A.

This is in sharp contrast with most existing studies on FL algorithms (Yang, 2013; Konevcnỳ et al., 2016; McMahan et al., 2017; Kairouz et al., 2021; Smith et al., 2018; Stich, 2018; Yu et al., 2019a;b; Khaled et al., 2020; Woodworth et al., 2020b;a; Karimireddy et al., 2020b; 2021; Haddadpour et al., 2019), which focus on the following empirical risk minimization (ERM) problem with the data set $\mathcal{S}$ distributed over different machines: $\min_{\mathbf{w} \in \mathbb{R}^d} \frac{1}{|\mathcal{S}|} \sum_{\mathbf{z} \in \mathcal{S}} \ell(\mathbf{w}; \mathbf{z})$. The major differences between CPR and ERM are (1) the ERM's objective is decomposable over training data, while the CPR is not decomposable over training examples; and (2) the data-dependent losses in ERM are decoupled between different data points; in contrast the data-dependent loss in CPR couples different training data points. These differences pose a big challenge for optimizing CPR in the FL setting, where the training data are distributed on different machines and are prohibited to be moved to a central server. In particular, the gradient of CPR cannot be written as the sum of local gradients at individual machines that only depend on the local data in those machines. Instead, the gradient of CPR at each machine not only depends on local data but also on data in other machines. As a result, the design of communication-efficient FL algorithms for optimizing CPR is much more complicated than that for ERM. In addition, the presence of non-linear function $f$ makes the algorithm design and analysis even more challenging than that with linear $f$. There are two levels of coupling in CPR with nonlinear $f$ with one level at the pairwise loss $\ell(h_{\mathbf{w}}(\mathbf{z}), h_{\mathbf{w}}(\mathbf{z}'))$ and another level at the non-linear risk of $f(g(\mathbf{w}; \mathbf{z}, \mathcal{S}_2))$, which makes estimation of stochastic gradient more tricky.

Although optimization of CPR can be solved by existing algorithms in a centralized learning setting (Wang et al., 2017; Ghadimi et al., 2020; Hu et al., 2020; Wang & Yang, 2022a; Qi et al., 2021; Wang et al., 2022; Zhu et al., 2022; Chen et al., 2021), extension of the existing algorithms to the FL setting is **non-trivial**. This is different from the extension of centralized algorithms for ERM to the FL setting. In the design and analysis of FL algorithms for ERM, the individual machines compute local gradients and update local models and communicate periodically for averaging models. The rationale of local FL algorithms for ERM is that as long as the gap error between local models and the averaged model is on par with the noise in the stochastic gradients by controlling the communication frequency, the convergence of local FL algorithms will not be sacrificed and is able to enjoy the parallel speed-up of using multiple machines. However, this rationale is not sufficient for developing FL algorithms for CPR optimization due to the challenges mentioned above.

To address the challenges, we propose two novel FL algorithms named **FedX1 and FedX2** for optimizing CPR with linear and non-linear $f$, respectively. The main innovation in the algorithm design lies at that we decouple the gradient of the objective with two types, active parts and lazy parts. The active parts depend on data in local machines and the lazy parts depend on data in other machines. We estimate the active parts using the local data and the local model and estimate the lazy parts using the information with delayed communications from other machines that are computed at historical models in the previous round. In terms of analysis, the challenge is that the model used in the computation of stochastic gradient estimator depends on the (historical) samples for computing the lazy parts at the current iteration, which is only exacerbated in the presence of non-linear function $f$. We develop a novel analysis that allows us to transfer the error of the gradient estimator into the latency error of the lazy parts and the gap error between local models and the global model. Hence, the rationale is that as long as the latency error of the lazy parts and the gap error between local models and the global model is on par with the noise in the stochastic gradient estimator we are able to achieve convergence and linear speed-up.

The main contributions of this work are summarized as follows:

- We propose two novel communication-efficient algorithms, FedX1 and FedX2, for optimizing the CPR with linear and nonlinear $f$, respectively. Besides communicating local models, the proposed algorithms need to communicate local prediction outputs only periodically.

- We perform novel technical analysis to prove the convergence of both algorithms. We show that both algorithms enjoy parallel speed-up in terms of the iteration complexity, and a lower-order communication complexity.

- We conduct empirical studies on two tasks for federated deep partial AUC optimization with a compositional loss and federated deep AUC optimization with a pairwise loss, and demonstrate the advantages of the proposed algorithms over several baselines.

## 2 RELATED WORK

**FL for ERM.** The challenge of FL is how to utilize the distributed data to learn a ML model with light communication cost without harming the data privacy (Konevcnỳ et al., 2016; McMahan et al., 2017). To reduce the communication cost, many algorithms have been proposed to skip communications (Stich, 2018; Yu et al., 2019a;b; Yang, 2013; Karimireddy et al., 2020b) or compress the communicated statistics (Stich et al., 2018; Basu et al., 2019; Jiang & Agrawal, 2018; Wangni et al., 2018; Bernstein et al., 2018). Tight analysis has been performed in various studies (Kairouz et al., 2021; Yu et al., 2019a;b; Khaled et al., 2020; Woodworth et al., 2020b;a; Karimireddy et al., 2020b; Haddadpour et al., 2019). However, most of these works target at ERM.

**FL for Non-ERM Problems.** In (Guo et al., 2020; Yuan et al., 2021a; Deng & Mahdavi, 2021; Deng et al., 2020; Liu et al., 2020; Sharma et al., 2022), federated minimax optimization algorithms are studied, which are not applicable to our problem when $f$ is non-convex. Gao et al. (2022) have considered a much simpler federated compositional optimization in the form of $\sum_k \mathbb{E}_{\zeta \sim \mathcal{D}_f^k} f_k(\mathbb{E}_{\xi \sim \mathcal{D}_g^k} g_k(\mathbf{w}; \xi); \zeta)$, where $k$ denotes the machine index. We can see that compared with our CPR risk, their objective does not involve interdependence between different machines. Li et al. (2022); Huang et al. (2022) have analyzed FL algorithms for bi-level problems where only the low-level objective involves distribution over many machines. Tarzanagh et al. (2022) considered another federated bilevel problem, where both upper and lower level objective are distributed many machines, but the lower level objective is not coupled with the data in the upper objective. Xing et al. (2022) studied a federated bilevel optimization in a server-clients setting, where the central server solves an objective that depends on optimal solutions of local clients. Our problem cannot be mapped into these federated bilevel optimization problems.

**Centralized Compositional Pairwise Risk Minimization.** In the centralized setting CPR minimization has been considered in recent works (Qi et al., 2021; Wang et al., 2022; Wang & Yang, 2022a; Qiu et al., 2022; Jiang et al., 2022). However, it is non-trivial to extend these algorithms to the FL setting due to the challenges mentioned earlier. We provide a summary of state-of-the-art sample complexities for solving ERM and CPR in both centralized and FL setting in Appendix B.

## 3 FEDX FOR OPTIMIZING CPR

We assume $\mathcal{S}_1, \mathcal{S}_2$ are split into $N$ non-overlapping subsets that are distributed over $N$ clients [1], i.e., $\mathcal{S}_1 = \mathcal{S}_1^1 \cup \mathcal{S}_1^2 \ldots \cup \mathcal{S}_1^N$ and $\mathcal{S}_2 = \mathcal{S}_2^1 \cup \mathcal{S}_2^2 \ldots \cup \mathcal{S}_2^N$. We denote by $\mathbb{E}_{\mathbf{z} \sim \mathcal{S}} = \frac{1}{|\mathcal{S}|} \sum_{\mathbf{z} \in \mathcal{S}}$. Denote by $\omega_{1i} = N|\mathcal{S}_1^i|/|\mathcal{S}_1|$ and $\omega_{2j} = N|\mathcal{S}_2^j|/|\mathcal{S}_2|, i = 1, \ldots, N, j = 1, \ldots, N$. We assume that these quantities $\omega_1 = (\omega_{11}, \ldots, \omega_{1N})$ and $\omega_2 = (\omega_{21}, \ldots, \omega_{2N})$ are available on all clients. If not, they can be easily computed and communicated once between the $N$ clients. Denote by $\nabla_1 \ell(\cdot, \cdot)$ and $\nabla_2 \ell(\cdot, \cdot)$ the partial gradients in terms of the first argument and the second argument, respectively. Without loss of generality, we assume the dimensionality of $h(\mathbf{w}; \mathbf{z})$ is 1 (i.e., $d_o = 1$) in the following presentation. For our discussion of complexity, we will simply assume $\omega_{1i}, \omega_{2j} \approx O(1)$.

### 3.1 FEDX1 FOR OPTIMIZING CPR WITH LINEAR $f$

With linear $f$, we rewrite the CPR risk into an equivalent form that is tailored to the FL setting:

$$\min_{\mathbf{w} \in \mathbb{R}^d} F(\mathbf{w}) = \frac{1}{N} \sum_{i=1}^N \mathbb{E}_{\mathbf{z} \in \mathcal{S}_1^i} \frac{1}{N} \sum_{j=1}^N \mathbb{E}_{\mathbf{z}' \in \mathcal{S}_2^j} \ell_{ij}(h(\mathbf{w}, \mathbf{z}), h(\mathbf{w}, \mathbf{z}')), \quad (2)$$

where $\ell_{ij}(h_{\mathbf{w}}(\mathbf{z}), h(\mathbf{w}, \mathbf{z}')) = \omega_{1i}\omega_{2j}\ell(h(\mathbf{w}, \mathbf{z}), h(\mathbf{w}, \mathbf{z}'))$. To highlight the challenge and motivate FedX, we compute the gradient of the objective function and decompose it into two terms:

$$\nabla F(\mathbf{w}) = \frac{1}{N} \sum_{i=1}^N \underbrace{\mathbb{E}_{\mathbf{z} \in \mathcal{S}_1^i} \frac{1}{N} \sum_{j=1}^N \mathbb{E}_{\mathbf{z}' \in \mathcal{S}_2^j} \nabla_1 \ell_{ij}(h(\mathbf{w}, \mathbf{z}), h(\mathbf{w}, \mathbf{z}')) \nabla h_{\mathbf{w}}(\mathbf{z})}_{\Delta_{i1}}$$

$$+ \frac{1}{N} \sum_{i=1}^N \underbrace{\mathbb{E}_{\mathbf{z}' \in \mathcal{S}_2^i} \frac{1}{N} \sum_{j=1}^N \mathbb{E}_{\mathbf{z} \in \mathcal{S}_1^j} \nabla_2 \ell_{ji}(h(\mathbf{w}, \mathbf{z}), h(\mathbf{w}, \mathbf{z}')) \nabla h(\mathbf{w}, \mathbf{z}')}_{\Delta_{i2}}.$$

---

[1]We use clients and machines interchangeably.

With the above decomposition, we can see that the main task at the local client $i$ is to estimate the gradient terms $\Delta_{i1}$ and $\Delta_{i2}$. Due to the symmetry between $\Delta_{i1}$ and $\Delta_{i2}$, below, we only use $\Delta_{i1}$ as an illustration for explaining the proposed algorithm. The difficulty in computing $\Delta_{i1}$ lies at it relies on data in other machines due to the presence of $\mathbb{E}_{\mathbf{z}' \in \mathcal{S}_2^j}$ for all $j$. To overcome this difficulty, we decouple the data-dependent factors in $\Delta_{i1}$ into two types marked by green and blue shown below:

$$\Delta_{i1} = \underbrace{\mathbb{E}_{\mathbf{z} \in \mathcal{S}_1^i}}_{\text{local1}} \underbrace{\frac{1}{N} \sum_{j=1}^N \mathbb{E}_{\mathbf{z}' \in \mathcal{S}_2^j}}_{\text{global1}} \nabla_1 \ell_{ij}( \underbrace{h(\mathbf{w}, \mathbf{z})}_{\text{local2}}, \underbrace{h(\mathbf{w}, \mathbf{z}')}_{\text{global2}} ) \underbrace{\nabla h(\mathbf{w}, \mathbf{z})}_{\text{local3}}. \tag{3}$$

It is notable that the three green terms can be estimated or computed based the local data. In particular, local1 can be estimated by sampling data from $\mathcal{S}_1^i$ and local2 and local3 can be computed based on the sampled data $\mathbf{z}$ and the local model parameter. The difficulty springs from estimating and computing the two blue terms that depend on data on all machines. *We would like to avoid communicating $h(\mathbf{w}; \mathbf{z}')$ at every iteration for estimating the blue terms as each communication would incur additional communication overhead.* To tackle this, we propose to leverage the historical information computed in the previous round [2]. To put this into context of optimization, we consider the update at the $k$-th iteration during the $r$-th round, where $k = 0, \ldots, K - 1$. Let $\mathbf{w}_{i,k}^r$ denote the local model in $i$-th client at the $k$-th iteration within $r$-th round. Let $\mathbf{z}_{i,k,1}^r \in \mathcal{S}_1^i, \mathbf{z}_{i,k,2}^r \in \mathcal{S}_2^i$ denote the data sampled at the $k$-th iteration from $\mathcal{S}_1^i$ and $\mathcal{S}_2^i$, respectively. Each local machine will compute $h(\mathbf{w}_{i,k}^r; \mathbf{z}_{i,k,1}^r)$ and $h(\mathbf{w}_{i,k}^r; \mathbf{z}_{i,k,2}^r)$, which will be used for computing the active parts. Across all iterations $k = 0, \ldots, K - 1$, we will accumulate the computed prediction outputs over sampled data and stored in two sets $\mathcal{H}_{i,1}^r = \{h(\mathbf{w}_{i,k}^r; \mathbf{z}_{i,k,1}^r), k = 0, \ldots, K - 1\}$ and $\mathcal{H}_{i,2}^r = \{h(\mathbf{w}_{i,k}^r; \mathbf{z}_{i,k,2}^r), k = 0, \ldots, K - 1\}$. At the end of round $r$, we will communicate $\mathbf{w}_{i,K}^r$ and $\mathcal{H}_{i,1}^r$ and $\mathcal{H}_{i,2}^r$ to the central server, which will average the local models to get a global model $\mathbf{w}_r$ and also aggregate $\mathcal{H}_1^r = \mathcal{H}_{1,1}^r \cup \mathcal{H}_{2,1}^r \ldots \cup \mathcal{H}_{N,1}^r$ and $\mathcal{H}_2^r = \mathcal{H}_{1,2}^r \cup \mathcal{H}_{2,2}^r \ldots \cup \mathcal{H}_{N,2}^r$. These aggregated information will be broadcast to each individual client. Then, at the $k$-th iteration in the $r$-th round, we estimate the blue term by sampling $h_{2,\xi}^{r-1} \in \mathcal{H}_2^{r-1}$ without replacement and compute an estimator of $\Delta_{i1}$ by

$$G_{i,k,1}^r = \nabla_1 \ell_{ij}( \underbrace{h(\mathbf{w}_{i,k}^r; \mathbf{z}_{i,k,1}^r)}_{\text{active}}, \underbrace{h_{2,\xi}^{r-1}}_{\text{lazy}} ) \underbrace{\nabla h(\mathbf{w}_{i,k}^r; \mathbf{z}_{i,k,1}^r)}_{\text{active}}, \tag{4}$$

where $\xi = (j, t, \mathbf{z}_{j,t,2}^{r-1})$ represents a random variable that captures the randomness in the sampled client $j \in \{1, \ldots, N\}$, iteration index $k \in \{0, \ldots, K - 1\}$ and data sample $\mathbf{z}_{j,t,2}^{r-1} \in \mathcal{S}_2^j$, which is used for estimating the global1 in (3). We refer to the green factors in $G_{i,k,1}$ as the active parts and the blue factor in $G_{i,k,1}$ as the lazy part. Similarly, we can estimate $\Delta_{i2}$ by $G_{i,k,2}$

$$G_{i,k,2}^r = \nabla_2 \ell_{j'i}( \underbrace{h_{1,\zeta}^{r-1}}_{\text{lazy}}, \underbrace{h(\mathbf{w}_{i,k}^r; \mathbf{z}_{i,k,2}^r)}_{\text{active}} ) \underbrace{\nabla h(\mathbf{w}_{i,k}^r; \mathbf{z}_{i,k,2}^r)}_{\text{active}}, \tag{5}$$

where $h_{1,\zeta}^{r-1} \in \mathcal{H}_1^{r-1}$ is a randomly sampled prediction output in the previous round with $\zeta = (j', t', \mathbf{z}_{j',t',1}^{r-1})$ representing a random variable including a client sample $j'$ and iteration sample $t'$ and the data sample $\mathbf{z}_{j',t',1}^{r-1}$. Then we will update the local model parameter $\mathbf{w}_{i,k}^r$ by using a gradient estimator $G_{i,k,1}^r + G_{i,k,2}^r$.

We present the detailed steps of the proposed algorithm FedX1 in Algorithm 1. Several remarks are following: (i) at every round, the algorithm needs to communicate both the model parameters $\mathbf{w}_{i,K}^r$ and the historical prediction outputs $\mathcal{H}_{i,1}^{r-1}$ and $\mathcal{H}_{i,2}^{r-1}$, where $\mathcal{H}_{i,*}^{r-1}$ is constructed by collecting all or sub-sampled computed predictions in the $(r-1)$-th round. The bottom line for constructing $\mathcal{H}_{i,*}^{r-1}$ is to ensure that $\mathcal{H}_*^{r-1}$ contains at least $K$ independently sampled predictions that are from the previous round on all machines such that the corresponding data samples involved in $\mathcal{H}_*^{r-1}$ can be used to approximate $\frac{1}{N} \sum_{i=1}^N \mathbb{E}_{\mathbf{z} \in \mathcal{S}_*^i}$ $K$ times. Hence, to keep the communication costs minimal, each client at least needs to sample $O(\lceil K/N \rceil)$ sampled predictions from all iterations $k = 0, 1, \ldots, K - 1$ and send them to the server for constructing $\mathcal{H}_*^{r-1}$, which is then broadcast to all clients for computing the lazy parts in the round $r$. As a result, the minimal communication costs per-round per-client is

---

[2] A round is defined as a sequence of local updates between two consecutive communications.

---

**Algorithm 1** FedX1: Federated Learning for CPR with linear $f$

---

1: On Client $i$: **Require** parameters $\eta, K$
2: Initialize model $\mathbf{w}_{i,0}^0$ and initialize Buffer $\mathcal{B}_{i,1} = \emptyset$ and $\mathcal{B}_{i,2} = \emptyset$
3: Sample $K$ points from $\mathcal{S}_1^i$, compute their predictions using model $\mathbf{w}_{i,0}^0$ denoted by $\mathcal{H}_{i,1}^0$
4: Sample $K$ points from $\mathcal{S}_2^i$, compute their predictions using model $\mathbf{w}_{i,0}^0$ denoted by $\mathcal{H}_{i,2}^0$
5: **for** $r = 1, ..., R$ **do**
6:     Send $\mathcal{H}_{i,1}^{r-1}, \mathcal{H}_{i,2}^{r-1}$ to the server
7:     Receive $\mathcal{R}_{i,1}^{r-1}, \mathcal{R}_{i,2}^{r-1}$ from the server
8:     Update buffer $\mathcal{B}_{i,1}, \mathcal{B}_{i,2}$ using $\mathcal{R}_{i,1}^{r-1}, \mathcal{R}_{i,2}^{r-1}$ with shuffling   $\diamond$ see text for updating the buffer
9:     Set $\mathcal{H}_{i,1}^r = \emptyset, \mathcal{H}_{i,2}^r = \emptyset$
10:    **for** $k = 0, .., K - 1$ **do**
11:        Sample $\mathbf{z}_{i,k,1}^r$ from $\mathcal{S}_1^i$, sample $\mathbf{z}_{i,k,2}^r$ from $\mathcal{S}_2^i$         $\diamond$ or sample two mini-batches of data
12:        Take next $h_\xi^{r-1}$ and $h_\zeta^{r-1}$ from $\mathcal{B}_{i,1}$ and $\mathcal{B}_{i,2}$, respectively
13:        Compute $h(\mathbf{w}_{i,k}^r, \mathbf{z}_{i,k,1}^r)$ and $h(\mathbf{w}_{i,k}^r, \mathbf{z}_{i,k,2}^r)$
14:        Add $h(\mathbf{w}_{i,k}^r, \mathbf{z}_{i,k,1}^r)$ into $\mathcal{H}_{i,1}^r$ and add $h(\mathbf{w}_{i,k}^r, \mathbf{z}_{i,k,2}^r)$ into $\mathcal{H}_{i,2}^r$
15:        Compute $G_{i,k,1}^r$ and $G_{i,k,2}^r$ according to (4) and (5)
16:        $\mathbf{w}_{i,k+1}^r = \mathbf{w}_{i,k}^r - \eta(G_{i,k,1}^r + G_{i,k,2}^r)$
17:    **end for**
18:    Sends $\mathbf{w}_{i,K}^r$ to the server
19:    Receives $\bar{\mathbf{w}}^r$ from the server and set $\mathbf{w}_{i,0}^{r+1} = \bar{\mathbf{w}}_r$
20: **end for**

---

21: On Server
22: **for** $r = 0, ..., R - 1$ **do**
23:     Collects $\mathcal{H}_1^r = \mathcal{H}_{1,1}^r \cup \mathcal{H}_{2,1}^r \ldots \cup \mathcal{H}_{N,1}^r$ and $\mathcal{H}_2^r = \mathcal{H}_{1,2}^r \cup \mathcal{H}_{1,2}^r \ldots \cup \mathcal{H}_{N,2}^r$
24:     Set $\mathcal{R}_{i,1}^r = \mathcal{H}_1^r, \mathcal{R}_{i,2}^r = \mathcal{H}_2^r$
25:     Send $\mathcal{R}_{i,1}^r, \mathcal{R}_{i,2}^r$ to client $i$ for all $i \in [N]$
26:     Receive $\mathbf{w}_{i,K}^{r+1}$, from client $i$, compute $\bar{\mathbf{w}}^{r+1} = \frac{1}{N} \sum_{i=1}^N \mathbf{w}_{i,K}^{r+1}$ and broadcast it to all clients.
27: **end for**

---

$O(d + Kd_o/N)$. Nevertheless, for simplicity in Algorithm 1 we simply put all historical predictions into $\mathcal{H}_{i,*}^{r-1}$.

Similar to all other FL algorithms, FedX1 does not require communicating the raw input data, hence protects the privacy of the data. However, compared with most FL algorithms for ERM, FedX1 for CPR has an additional communication overhead at least $O(d_o K/N)$ which depends on the dimensionality of prediction output $d_o$. For learning a high-dimensional model (e.g. deep neural network with $d \gg 1$) with score-based pairwise losses ($d_o = 1$), the additional communication cost $O(K/N)$ could be marginal. For updating the buffer $\mathcal{B}_{i,1}$ and $\mathcal{B}_{i,2}$, we can simply flush the history and add the newly received $\mathcal{R}_{i,1}^{r-1}$ with random shuffling to $\mathcal{B}_{i,1}$ and add $\mathcal{R}_{i,2}^{r-1}$ with random shuffling to $\mathcal{B}_{i,2}$. However, we can keep the history up to a certain limit as long as the latency error can be well controlled, which will be analyzed in Appendix E.

Next, we present the theoretical results of FedX1 with more formal results given in appendix.

**Theorem 1.** *(Informal) Under appropriate conditions, by setting $\eta = O(\frac{N}{R^{2/3}})$ and $K = O(\frac{R^{1/3}}{N})$, Algorithm 1 ensures that $\mathbb{E}\left[\frac{1}{R} \sum_{r=1}^R \|\nabla F(\bar{\mathbf{w}}^r)\|^2\right] \leq O(\frac{1}{R^{2/3}})$.*

**Remark.** To get $\mathbb{E}[\frac{1}{R} \sum_{r=1}^R \|\nabla F(\bar{\mathbf{w}}^r)\|^2] \leq \epsilon^2$, we just need to set $R = O(\frac{1}{\epsilon^3})$, $\eta = N\epsilon^2$ and $K = \frac{1}{N\epsilon}$. The number of communications is much less than the total number of iterations i.e., $O(\frac{1}{N\epsilon^4})$ as long as $N \leq O(\frac{1}{\epsilon})$. And the sample complexity on each machine is $\frac{1}{N\epsilon^4}$, which is linearly reduced by the number of machines $N$.

**Novelty of Analysis.** As the lazy parts are computed in different machines in a previous round, the gradient estimators $G_{i,k,1}^r$ and $G_{i,k,2}^r$ will involve the dependency between the local model parameter $\mathbf{w}_{i,k}^r$ and the historical data contained in $\xi, \zeta$ used for computing $G_{i,k,1}^r$ and $G_{i,k,2}^r$, which makes the analysis more involved. We need to make sure that using the gradient estimator based on them can still result in "good" results. To this end, we borrow an analysis technique in (Yang et al., 2021b)

to decouple the dependence between the current model parameter and the data used for computing the current gradient estimator, in which they used data in previous iteration to couple the data in the current iteration in order to compute a gradient of the pairwise loss $\ell(h(\mathbf{w}_t; \mathbf{z}_t), h(\mathbf{w}_t; \mathbf{z}_{t-1}))$. Nevertheless, in federated CPR controlling the error brought by the lazy parts is more challenging since the delay is much longer and they were computed on different machines. In our analysis, we replace $\mathbf{w}_{i,k}^r$ with $\bar{\mathbf{w}}^{r-1}$ to decouple the dependence between the model parameter $\bar{\mathbf{w}}^{r-1}$ and the historical data $\xi, \zeta$, then we need to control the latency error $\|\bar{\mathbf{w}}^{r-1} - \bar{\mathbf{w}}^r\|^2$ and the gap error between different machines $\sum_i \sum_k \mathbb{E}\|\bar{\mathbf{w}}^r - \mathbf{w}_{i,k}^r\|^2$ such that the complexities are not compromised.

### 3.2  FEDX2 FOR OPTIMIZING CPR WITH NONLINEAR $f$

Similarly, we re-write the objective into an equivalent form that is tailored to the FL setting, i.e.,

$$F(\mathbf{w}) = \frac{1}{N} \sum_{i=1}^{N} \mathbb{E}_{\mathbf{z} \in \mathcal{S}_1^i} f_i \left( \frac{1}{N} \sum_{j=1}^{N} \mathbb{E}_{\mathbf{z}' \in \mathcal{S}_2^j} \ell_j(h(\mathbf{w}; \mathbf{z}), h(\mathbf{w}; \mathbf{z}')) \right), \tag{6}$$

where $f_i(\cdot) = \omega_{1i} f(\cdot)$ and $\ell_j(\cdot, \cdot) = \omega_{2j} \ell(\cdot, \cdot)$. We compute the gradient and decompose it into two terms:

$$\nabla F(\mathbf{w}) = \frac{1}{N} \sum_{i=1}^{N} \mathbb{E}_{\mathbf{z} \in \mathcal{S}_1^i} \frac{1}{N} \sum_{j=1}^{N} \mathbb{E}_{\mathbf{z}' \in \mathcal{S}_2^j} \underbrace{\nabla f_i(g(\mathbf{w}; \mathbf{z}, \mathcal{S}_2))\, \nabla_1 \ell_j(h(\mathbf{w}; \mathbf{z}), h(\mathbf{w}; \mathbf{z}')) \nabla h(\mathbf{w}; \mathbf{z})}_{\Delta_{i1}}$$

$$+ \frac{1}{N} \sum_{i=1}^{N} \mathbb{E}_{\mathbf{z}' \in \mathcal{S}_2^i} \frac{1}{N} \sum_{j=1}^{N} \mathbb{E}_{\mathbf{z} \in \mathcal{S}_1^j} \underbrace{\nabla f_j(g(\mathbf{w}; \mathbf{z}, \mathcal{S}_2))\, \nabla_2 \ell_i(h(\mathbf{w}; \mathbf{z}), h(\mathbf{w}; \mathbf{z}')) \nabla h(\mathbf{w}; \mathbf{z}')}_{\Delta_{i2}}. \tag{7}$$

Compared to that in (3) for CPR with linear $f$, the $\Delta_{i1}$ term above involves another factor $\nabla f_i(g(\mathbf{w}; \mathbf{z}, \mathcal{S}_2))$, which cannot be computed locally as it depends on $\mathcal{S}_2$ distributed over all machines. Similarly, the $\Delta_{i2}$ term above involves another non-locally computable factor $\nabla f_j(g(\mathbf{w}; \mathbf{z}, \mathcal{S}_2))$. To address the challenge of estimating $g(\mathbf{w}; \mathbf{z}, \mathcal{S}_2)$, we leverage the similar technique in the centralized setting (Wang & Yang, 2022b) by tracking it using a moving average estimator based on random samples. In a centralized setting, one can maintain and update $u(\mathbf{z})$ for estimating $g(\mathbf{w}, \mathbf{z}, \mathcal{S}_2)$ by $\mathbf{u}(\mathbf{z}) \leftarrow (1 - \gamma) \mathbf{u}(\mathbf{z}) + \gamma \ell(h(\mathbf{w}, \mathbf{z}), h(\mathbf{w}, \mathbf{z}'))$, where $\mathbf{z}'$ is a random sample from $\mathcal{S}_2$. However, this is not possible in an FL setting as $\mathcal{S}_2$ is distributed over many machines. To tackle this, we leverage the same delay communication technique used in the last subsection. In particular, at the $k$-th iteration in the $r$-th round, we can update $\mathbf{u}(\mathbf{z}_{i,k,1}^r)$ for a sampled $\mathbf{z}_{i,k,1}^r$ by

$$\mathbf{u}_{i,k}^r(\mathbf{z}_{i,k,1}^r) = (1 - \gamma) \mathbf{u}_{i,k}^r(\mathbf{z}_{i,k,1}^r) + \gamma \ell_j(h(\mathbf{w}_{i,k}^r, \mathbf{z}_{i,k,1}^r), h_{\xi,2}^{r-1}), \tag{8}$$

where $h_{\xi,2}^{r-1}$ is a random sample from $\mathcal{H}_2^{r-1}$ where $\xi = (j', t', \mathbf{z}_{j',t',2}^{r-1})$ captures the randomness in client, iteration index and data sample in the last round. Then, we can use $\nabla f_i(\mathbf{u}_{i,k}^r(\mathbf{z}_{i,k,1}^r))$ in place of $\nabla f_i(g(\mathbf{w}_{i,k}^r; \mathbf{z}_{i,k,1}^r))$ for estimating $\Delta_{i1}$. However, it is more nuanced for estimating $\nabla f_j(g(\mathbf{w}; \mathbf{z}, \mathcal{S}_2))$ in $\Delta_{2i}$ since $\mathbf{z} \in \mathcal{S}_j^2$ is not local random data. To address this, we propose to communicate $\mathcal{U}^{r-1} = \{\mathbf{u}_{i,k}^{r-1}(\mathbf{z}_{i,k,1}^{r-1}), i \in [N], k \in [K] - 1\}$. Then at the $k$-iteration in the $r$-th round of the $i$-th client, we can estimate $\nabla f_j(g(\mathbf{w}; \mathbf{z}, \mathcal{S}_2))$ with a random sample from $\mathcal{U}^{r-1}$ denoted by $u_\zeta^{r-1}$, where $\zeta = (j', t', \mathbf{z}_{j',t',1}^{r-1})$, i.e., by using $\nabla f_{j'}(\mathbf{u}_\zeta^{r-1})$. Then we estimate $\Delta_{1i}$ and $\Delta_{2i}$ by

$$G_{i,k,1}^r = \underbrace{\nabla f_i(\mathbf{u}_{i,k}^r(\mathbf{z}_{i,k,1}^r))}_{\text{active}}\, \nabla_1 \ell_j \big( \underbrace{h(\mathbf{w}_{i,k}^r; \mathbf{z}_{i,k,1}^r)}_{\text{active}}, \underbrace{h_{2,\xi}^{r-1}}_{\text{lazy}} \big)\, \underbrace{\nabla h(\mathbf{w}_{i,k}^r; \mathbf{z}_{i,k,1}^r)}_{\text{active}}$$

$$G_{i,k,2}^r = \underbrace{\nabla f_{j'}(\mathbf{u}_\zeta^{r-1})}_{\text{lazy}}\, \nabla_2 \ell_i \big( \underbrace{h_{1,\zeta}^{r-1}}_{\text{lazy}}, \underbrace{h(\mathbf{w}_{i,k}^r; \mathbf{z}_{i,k,2}^r)}_{\text{active}} \big)\, \underbrace{\nabla h(\mathbf{w}_{i,k}^r; \mathbf{z}_{i,k,2}^r)}_{\text{active}} \tag{9}$$

where $j, \xi, j', \zeta$ are random variables. Another difference from CPR with linear $f$ is that even in the centralized setting directly using $G_{i,k,1}^r + G_{i,k,2}^r$ will lead to a worse complexity due to that non-linear $f$ make the stochastic gradient estimator biased (Wang et al., 2017). Hence, in order to improve the convergence, we follow existing state-of-the-art algorithms for stochastic compositional optimization (Ghadimi et al., 2020; Wang & Yang, 2022b) to compute a moving average estimator

---

**Algorithm 2** FedX2: Federated Learning for CPR with non-linear $f$

---

1: On Client $i$: **Require** parameters $\eta, K$
2: Initialize model $\mathbf{w}_{i,0}^0$, $\mathcal{U}_i^0 = \{u^0(\mathbf{z}) = 0, \mathbf{z} \in \mathcal{S}_1^i\}$, $G_{i,0}^0 = 0$, and buffer $\mathcal{B}_{i,1}, \mathcal{B}_{i,2}, \mathcal{C}_i = \emptyset$
3: Sample $K$ points from $S_1^i$, compute their predictions using model $\mathbf{w}_{i,0}^0$ denoted by $\mathcal{H}_{i,1}^0$
4: Sample $K$ points from $S_2^i$, compute their predictions using model $\mathbf{w}_{i,0}^0$ denoted by $\mathcal{H}_{i,2}^0$
5: **for** $r = 1, ..., R$ **do**
6: $\quad$ Send $\mathcal{H}_{i,1}^{r-1}, \mathcal{H}_{i,2}^{r-1}, \mathcal{U}_i^{r-1}$ to the server
7: $\quad$ Receive $\mathcal{R}_{i,1}^{r-1}, \mathcal{R}_{i,2}^{r-1}, \mathcal{P}^{r-1}$ from the server
8: $\quad$ Update the buffer $\mathcal{B}_{i,1}, \mathcal{B}_{i,2}, \mathcal{C}_i$ using $\mathcal{R}_{i,1}^{r-1}, \mathcal{R}_{i,2}^{r-1}, \mathcal{P}^{r-1}$ with shuffling, respectively
9: $\quad$ Set $\mathcal{H}_{i,1}^r = \emptyset, \mathcal{H}_{i,2}^r = \emptyset, \mathcal{U}_i^r = \emptyset$
10: $\quad$ **for** $k = 0, .., K - 1$ **do**
11: $\quad\quad$ Sample $\mathbf{z}_{i,k,1}^r$ from $\mathcal{S}_1^i$, sample $\mathbf{z}_{i,k,2}^r$ from $\mathcal{S}_2^i$ $\quad\quad\quad \diamond$ or sample two mini-batches of data
12: $\quad\quad$ Take next $h_\xi^{r-1}, h_\zeta^{r-1}$ and $u_\zeta^{r-1}$ from $\mathcal{B}_{i,1}$ and $\mathcal{B}_{i,2}$ and $\mathcal{C}_i$, respectively
13: $\quad\quad$ Compute $h(\mathbf{w}_{i,k}^r, \mathbf{z}_{i,k,1}^r)$ and $h(\mathbf{w}_{i,k}^r, \mathbf{z}_{i,k,2}^r)$
14: $\quad\quad$ Compute $h(\mathbf{w}_{i,k}^r, \hat{\mathbf{z}}_{i,k,1}^r)$ and $h(\mathbf{w}_{i,k}^r, \hat{\mathbf{z}}_{i,k,2}^r)$ and add them to $\mathcal{H}_{i,1}^r, \mathcal{H}_{i,2}^r$, respectively
15: $\quad\quad$ Compute $\mathbf{u}_{i,k}^r(\mathbf{z}_{i,k,1}^r)$ according to (8 and add it to $\mathcal{U}_i^r$
16: $\quad\quad$ Compute $G_{i,k,1}^r$ and $G_{i,k,2}^r$ according to (9)
17: $\quad\quad$ $G_{i,k}^r = (1 - \beta)G_{i,k-1}^r + \beta(G_{i,k,1}^r + G_{i,k,2}^r)$
18: $\quad\quad$ $\mathbf{w}_{i,k+1}^r = \mathbf{w}_{i,k}^r - \eta G_{i,k}^r$
19: $\quad$ **end for**
20: $\quad$ Sends $\mathbf{w}_{i,K}^r, G_{i,k}^r$ to the server
21: $\quad$ Receives $\bar{\mathbf{w}}^r, \bar{G}^r$ from the server and set $\mathbf{w}_{i,0}^{r+1} = \bar{\mathbf{w}}_r, G_{i,0}^{r+1} = \bar{G}^r$
22: **end for**

---

23: On Server
24: **for** $r = 0, ..., R - 1$ **do**
25: $\quad$ Collects $\mathcal{H}_*^r = \mathcal{H}_{1,*}^r \cup \mathcal{H}_{2,*}^r \ldots \cup \mathcal{H}_{N,*}^r$ and $\mathcal{U}^r = \mathcal{U}_1^r \cup \mathcal{U}_1^r \ldots \cup \mathcal{U}_N^r$, where $* = 1, 2$
26: $\quad$ Set $\mathcal{R}_{i,1}^r = \mathcal{H}_1^r, \mathcal{R}_{i,2}^r = \mathcal{H}_2^r, \mathcal{P}_i^r = \mathcal{U}^r$ and send them to Client $i$ for all $i \in [N]$
27: $\quad$ Receive $\mathbf{w}_{i,K}^{r+1}, G_{i,K}^{r+1}$ from client $i$, compute $\bar{\mathbf{w}}^{r+1} = \frac{1}{N}\sum_{i=1}^N \mathbf{w}_{i,K}^{r+1}, G^{r+1} = \frac{1}{N}\sum_{i=1}^N G_{i,K}^{r+1}$
$\quad\quad$ and broadcast them to all clients.
28: **end for**

---

for the gradient at local machines, i.e., Step 17 in Algorithm 3. With these changes, we present the detailed steps of FedX2 for solving CPR with non-linear $f$ in Algorithm 3. The buffers $\mathcal{B}_{i,*}$ and $\mathcal{C}_i$ are updated similar to that for FedX1. Different from FedX1, there is an additional communication cost for communicating $\mathcal{U}_i^{r-1}$ and an additional buffer $\mathcal{C}_i$ at each local machine to store the received $\mathcal{P}_i^{r-1}$ from aggregated $\mathcal{U}^{r-1}$. Nevertheless, these additional costs are marginal compared with communicating $\mathcal{H}_*^{r-1}$ and maintaining the buffer $\mathcal{B}_{i,*}$.

We present the convergence result of FedX2 below with more formal results given in appendix.

**Theorem 2.** *(Informal) Under appropriate conditions, denoting $M = \max_i |\mathcal{S}_i^1|$ as the largest number of data on a single machine, by setting $\gamma = O(\frac{M^{1/3}}{R^{2/3}})$, $\beta = O(\frac{1}{M^{1/6}R^{2/3}})$, $\eta = O(\frac{1}{M^{2/3}R^{2/3}})$ and $K = O(M^{1/3}R^{1/3})$, Algorithm 2 ensures that $\mathbb{E}\left[\frac{1}{R}\sum_{r=1}^R \|\nabla F(\bar{\mathbf{w}}^r)\|^2\right] \leq O(\frac{1}{R^{2/3}})$.*

**Remark.** To get $\mathbb{E}[\frac{1}{R}\sum_{r=1}^R \|\nabla F(\bar{\mathbf{w}}^r)\|^2] \leq \epsilon^2$, we just set $R = O(\frac{M^{1/2}}{\epsilon^3}), \eta = O(\frac{\epsilon^2}{M}), \gamma = O(\epsilon^2)$, $\beta = \frac{\epsilon^2}{\sqrt{M}}$ and $K = \frac{M^{1/2}}{\epsilon}$. The number of communications $R = O(\frac{M^{1/2}}{\epsilon^3})$ is less than the total number of iterations i.e., $O(\frac{M}{\epsilon^4})$ by a factor of $O(M^{1/2}/\epsilon)$. And the sample complexity on each machine is $\frac{M}{\epsilon^4}$, which is less than that in Wang & Yang (2022b) which has a sample complexity of $O(\sum_{i=1}^N |\mathcal{S}_i^1|/\epsilon^4)$. When the data are evenly distributed on different machines, we have achieved a linear speedup property. And in an extreme case where all data are on one machine, we see that the sample complexity of FedX2 matches that established in (Wang & Yang, 2022b), which is expected. Compared with FedX1, the analysis of FedX2 has to deal with several extra difficulties. First, with non-linear $f$, the coupling between the inner function and outer function adds to the complexity of interdependence between different rounds and different machines. Second, we have to deal with the error for the lazy part related to $\mathbf{u}$.

It is notable that our analysis for FedX2 with moving average gradient estimator for solving CPR is different from previous studies for local momentum methods (Yu et al., 2019a; Karimireddy et al., 2020a), which used a moving average with a fixed momentum parameter for computing a gradient estimator in local steps for the ERM problem. In contrast, in FedX2 the momentum parameter $\beta$ is decreasing as $R$ increases, which is similar to centralized algorithms for solving compositional problems (Ghadimi et al., 2020; Wang & Yang, 2022b).

## 4 EXPERIMENTS

To verify our algorithms, we run experiments on two tasks: federated deep partial AUC maximization and federated deep AUC maximization with a pairwise surrogate loss, which corresponds to (1) with non-linear $f$ and linear $f$, respectively.

**Datasets and Neural Networks.** We use four datasets: Cifar10, Cifar100 (Krizhevsky, 2009), CheXpert (Irvin et al., 2019), and ChestMNIST (Yang et al., 2021a), where the latter two datasets are large-scale medical image data. The statistics of these datasets are reported in Appendix. For Cifar10 and Cifar100, we sample 20% of the training data as validation set, and construct imbalanced binary versions with positive:negative = 1:5 in the training set similar to (Yuan et al., 2021b). For CheXpert, we consider the task of predicting Consolidation and use the last 1000 images in the training set as the validation set and use the original validation set as the testing set. For ChestMNIST, we consider the task of Mass prediction and use the provided train/valid/test split. We distribute training data to $N = 16$ machines unless specified otherwise. To increase the heterogeneity of data on different machines, we add random Gaussian noise of $\mathcal{N}(\mu, 0.04)$ to all training images, where $\mu \in \{-0.08 : 0.01 : 0.08\}$ that varies on different machines, i.e., for the $i$-th machine out of the $N = 16$ machines, its $\mu = -0.08 + i * 0.01$. We train ResNet18 from scratch for CIFAR-10 and CIFAR-100 data, and initialize DenseNet121 by an ImageNet pretrained model for CheXpert and ChestMNIST data. All experiments use the PyTorch framework (Paszke et al., 2019).

**Baselines.** We compare our algorithms with three local baselines: 1) *Local SGD* which optimizes a Cross-Entropy loss using classical local SGD algorithm; 2) *CODASCA* - a state-of-the-art FL algorithm for optimizing a min-max formulated AUC loss (Yuan et al., 2021a); and 3) *Local Pair* which optimizes the CPR risk using only local pairs. As a reference, we also compare with the *Centralized* methods, i.e., mini-batch SGD for CPR with linear $f$ and SOX for CPR with non-linear $f$. For each algorithm, we tune the initial step size in $[1e^{-3}, 1]$ using grid search and decay it by a factor of 0.1 after every 5K iterations. All algorithms are run for 20k iterations. The mini-batch sizes $B_1, B_2$ (as in Step 11 of FedX1 and FedX2) are set to 32. The $\beta$ parameter of FedX2 (and corresponding Local Pair and Centralized method) is set to 0.1. In the Centralized method, we tune the batch size $B_1$ and $B_2$ from $\{32, 64, 128, 256, 512\}$ in an effort to benchmark the best performance of the centralized setting. For CODASCA and Local SGD which are not using pairwise losses, we set the batch size to 64 for the sake of fair comparison with FedX. For all the non-centralized algorithms, we set the communication interval $K = 32$ unless specified otherwise. In every run of any algorithm, we use the validation set to select the best performing model and finally use the selected model to evaluate on the testing set. For each algorithm, we repeat 3 times with different random seeds and report the averaged performance.

**FedX2 for Federated Deep Partial AUC Maximization.** First, we consider the task of one way partial AUC maximization, which refers to the area under the ROC curve with false positive rate (FPR) restricted to be less than a threshold. We consider the KL-OPAUC loss function proposed in (Zhu et al., 2022), which is the formulation of (1) where $\mathcal{S}_1^i$ denotes the set of positive data, $\mathcal{S}_2^i$ denotes the set of negative data and $\ell(a, b) = \exp((b + 1 - a)_+^2/\lambda)$ and $f(\cdot) = \lambda \log(\cdot)$ where $\lambda$ is a parameter tuned in $[1 : 5]$. The experimental results are reported in Table 1. We have the following observations: (i) FedX2 is better than all local methods (i.e., Local SGD, Local Pair and CODASCA), and achieves competitive performance as the Centralized method, which indicates the our algorithm can effectively utilize data on all machines. The better performance of FedX2 on CIFAR100 and CheXpert than the Centralized method is probably due to that the Centralized method may overfit the training data; (ii) FedX2 is better than the Local Pair method, which implies that using data pairs from all machines are helpful for improving the performance in terms of partial AUC maximization; and (iii) FedX2 is better than CODASCA, which is not surprising since CODASCA is designed to optimize AUC loss, while FedX2 is used to optimize partial AUC loss.

**FedX1 for Federated Deep AUC maximization with Corrupted Labels.** Second, we consider the task of federated deep AUC maximization. Since deep AUC maximization for solving a min-max

Table 1: Comparison for Federated Deep Partial AUC Maximization. All reported results are partial AUC scores on testing data.

| $K = 32, N = 16$ | | Centralized (OPAUC Loss) | Local SGD (CE Loss) | CODASCA (Min-Max AUC) | Local Pair (OPAUC Loss) | FedX2 (OPAUC Loss) |
|---|---|---|---|---|---|---|
| Cifar10 | FPR $\leq 0.3$ | 0.7655±0.0039 | 0.6825±0.0047 | 0.7288±0.0035 | 0.7487±0.0059 | **0.7580±0.0034** |
| | FPR $\leq 0.5$ | 0.8032±0.0039 | 0.7279±0.0050 | 0.7702±0.0029 | 0.7888±0.0052 | **0.7978±0.0026** |
| Cifar100 | FPR $\leq 0.3$ | 0.6287±0.0037 | 0.5875±0.0016 | 0.6131±0.0054 | 0.6281±0.0032 | **0.6332±0.0024** |
| | FPR $\leq 0.5$ | 0.6487±0.0026 | 0.6124±0.0021 | 0.6406±0.0041 | 0.6569±0.0017 | **0.6623±0.0022** |
| CheXpert | FPR $\leq 0.3$ | 0.7220±0.0035 | 0.6495±0.0039 | 0.6903±0.0059 | 0.6902±0.0053 | **0.7344±0.0042** |
| | FPR $\leq 0.5$ | 0.7861±0.0040 | 0.7017±0.0042 | 0.7770±0.0071 | 0.7483±0.0033 | **0.7918±0.0037** |
| ChestMNIST | FPR $\leq 0.3$ | 0.6344±0.0053 | 0.5904±0.0012 | 0.6071±0.0040 | 0.5802±0.0039 | **0.6228±0.0048** |
| | FPR $\leq 0.5$ | 0.6622±0.0029 | 0.6072±0.0034 | 0.6272±0.0038 | 0.6026±0.0025 | **0.6490±0.0039** |

Table 2: Comparison for Federated Deep AUC maximization under corrupted labels. All reported results are AUC scores on testing data.

| $K = 32, N = 16$ | Centralized (PSM Loss) | Local SGD (CE Loss) | CODASCA (Min-Max AUC) | Local Pair (PSM Loss) | FedX1 (PSM Loss) |
|---|---|---|---|---|---|
| Cifar10 | 0.7352±0.0043 | 0.6501±0.0024 | 0.6407±0.0044 | 0.7287±0.0027 | **0.7344±0.0038** |
| Cifar100 | 0.6114±0.0038 | 0.5700±0.0031 | 0.5950±0.0039 | 0.6175±0.0045 | **0.6208±0.0041** |
| CheXpert | 0.8149±0.0031 | 0.6782±0.0032 | 0.7062±0.0085 | 0.7924±0.0043 | **0.8431±0.0027** |
| ChestMNIST | 0.7227±0.0026 | 0.5642±0.0041 | 0.6509±0.0033 | 0.6766±0.0019 | **0.6925±0.0030** |

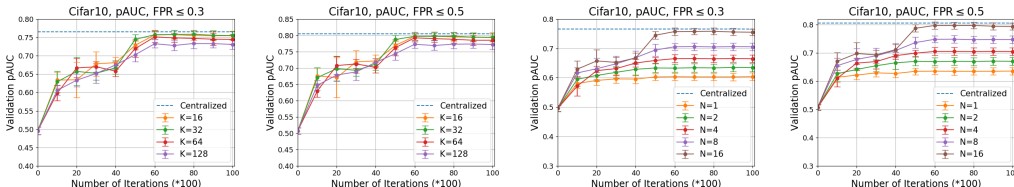

Figure 1: Ablation study: Left two: Fix $N$ and Vary $K$; Right two: Fix $K$ and Vary $N$

loss (an equivalent form for the pairwise square loss) has been developed in previous works (Yuan et al., 2021a), we aim to justify the benefit of using the general pairwise loss formulation. According to (Charoenphakdee et al., 2019), a symmetric loss can be more robust to data with corrupted labels for AUC maximization, where a symmetric loss is one such that $\ell(z) + \ell(-z)$ is a constant. Since the square loss is not symmetric, we conjecture that that min-max federated deep AUC maximization algorithm CODASCA is not robust to the noise in labels. In contrast, our algorithm FedX1 can optimize a symmetric pairwise loss; hence we expect FedX1 is better than CODASCA in the presence of corrupted labels. To verify this hypothesis, we generate corrupted data by flipping the labels of 20% of both the positive and negative training data. We use FedX1/Local Pair to optimize the symmetric pairwise sigmoid (PSM) loss (Calders & Jaroszewicz, 2007), which corresponds to (1) with linear $f(s) = s$ and $\ell(a, b) = (1 + \exp((a - b)))^{-1}$, where $a$ is a positive data score and $b$ is a negative data score. The results are reported in Table 2. We observe that FedX1 is more robust to label noises compared to other local methods, including Local SGD, Local Pair, and CODASCA that optimizes a min-max AUC loss. As before, FedX1 has competitive performance with the Centralized method.

**Ablation Study.** Third, we show an ablation study to further verify our theory. In particular, we show the benefit of using multiple machines and the lower communication complexity by using $K > 1$ local updates between two communications. To verify the first effect, we fix $K$ and vary $N$, and for the latter we fix $N$ and vary $K$. We conduct experiments on the CIFAR-10 data for optimizing the CPR risk corresponding to partial AUC loss and the results are plotted in Figure 1. The left two figures demonstrate that our algorithm can tolerate a certain value of $K$ for skipping communications without harming the performance; and the right two figures demonstrate the advantage of FL by using FedX2, i.e., using data from more sources can dramatically improve the performance.

## 5 CONCLUSION

In this paper, we have considered federated learning (FL) for compositional pairwise risk minimization problems. We have developed communication-efficient FL algorithms to alleviate the interdependence between different machines. Novel convergence analysis is performed to address the technical challenges and to improve both iteration and communication complexities of proposed algorithms. We have conducted empirical studies of the proposed FL algorithms for solving deep partial AUC maximization and deep AUC maximization and achieved promising results compared with several baseline algorithms.

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

# A APPLICATIONS OF CPR PROBLEMS

We now present some concrete applications of the CPR minimization problems, including AUROC maximization, partial AUROC maximization and AUPRC maximization. A more comprehensive list of CPR minimization problems is discussed in the Intrduction section and can also be found in a recent survey (Yang, 2022).

_AUROC Maximization_ The area under ROC curve (AUROC) is defined (Hanley & McNeil, 1982) as
$$\text{AUROC}(\mathbf{w}) = \mathbb{E}[\mathbb{I}(h(\mathbf{w}, \mathbf{z}) \geq h(\mathbf{w}, \mathbf{z}'))|y = +1, y' = -1], \tag{10}$$
where $\mathbf{z}, \mathbf{z}'$ are a pair of data features and $y, y'$ are the corresponding labels. To maximize the AUROC, there are a number of surrogate losses $\ell(\cdot)$, e.g. $\ell(\mathbf{w}; \mathbf{z}, \mathbf{z}') = (1 - h(\mathbf{w}, \mathbf{z}) + h(\mathbf{w}, \mathbf{z}'))^2$, that have proposed in the literature (Gao et al., 2013; Zhao et al., 2011; Gao & Zhou, 2015; Calders & Jaroszewicz, 2007; Charoenphakdee et al., 2019; Yang et al., 2021b), which formulates the problem into
$$\min_{\mathbf{w}} \frac{1}{|\mathcal{S}_1|} \sum_{\mathbf{z}_i \in S_1} \frac{1}{|\mathcal{S}_2|} \sum_{\mathbf{z}_j \in S_2} \ell(\mathbf{w}, \mathbf{z}_i, \mathbf{z}_j), \tag{11}$$
where $\mathcal{S}_1$ is the set of data with positive labels and $\mathcal{S}_2$ is the set of data with negative labels. This is a CPR problem of (1) with $f(x) = x$.

_Partial AUROC Maximization_ In medical diagnosis, high false positive rates (FPR) and low true positive rates (TPR) may cause a large cost. To alleviate this, we will also consider optimizing partial AUC (pAUC). This task considers to maximize the area under ROC curve with the restriction that the false positive rate to be less than a certain level. In Zhu et al. (2022), it has been shown that the partial AUROC maximization problem can be solved by the
$$\min_{\mathbf{w}} \frac{1}{|\mathcal{S}_1|} \sum_{\mathbf{x}_i \in \mathcal{S}_1} \lambda \log \left( \frac{1}{|\mathcal{S}_2|} \sum_{\mathbf{z}_j \in \mathcal{S}_2} \exp(\frac{\tilde{\ell}(\mathbf{w}, \mathbf{z}_i, \mathbf{z}_j)}{\lambda}) \right), \tag{12}$$
where $\mathcal{S}_1$ is the set of positive data, $\mathcal{S}_2$ is the set of negative data, $\tilde{\ell}(\cdot)$ is surrogate loss, and $\lambda$ is associated with the tolerance level of false positive rate. This is a CPR problem of (1) with $f(x) = \lambda \log(x)$, and $\ell(\mathbf{w}, \mathbf{z}_i, \mathbf{z}_j) = \exp(\frac{\tilde{\ell}(\mathbf{w}, \mathbf{z}_i, \mathbf{z}_j)}{\lambda})$.

_AUPRC Maximization_ According to (Boyd et al., 2013), the area under the precision-recall curve (AUPRC) can be approximated by
$$\frac{1}{|\mathcal{S}|} \sum_{(\mathbf{z}_i, y_i) \in \mathcal{S}} \mathbb{I}(y_i = 1) \frac{\sum\limits_{(\mathbf{z}_j, y_j) \in \mathcal{S}} \mathbb{I}(y_j = 1)\mathbb{I}(h(\mathbf{w}, \mathbf{z}_i) \geq h(\mathbf{w}, \mathbf{z}_j))}{\sum\limits_{(\mathbf{z}_j, y_j) \in \mathcal{S}} \mathbb{I}(h(\mathbf{w}, \mathbf{z}_i) \geq h(\mathbf{w}, \mathbf{z}_j))}. \tag{13}$$
Then using a surrogate loss, the AUPRC maximization problem becomes
$$\min_{\mathbf{w}} -\frac{1}{|\mathcal{S}|} \sum_{(\mathbf{z}_i, y_i) \in \mathcal{S}} \mathbb{I}(y_i = 1) \frac{\sum\limits_{(\mathbf{z}_j, y_j) \in \mathcal{S}} \mathbb{I}(y_j = 1)\tilde{\ell}(\mathbf{w}, \mathbf{z}_i, \mathbf{z}_j)}{\sum\limits_{(\mathbf{z}_j, y_j) \in \mathcal{S}} \tilde{\ell}(\mathbf{w}, \mathbf{z}_i \mathbf{z}_j)}, \tag{14}$$
which is a CPR problem of (1) with $\ell(\mathbf{w}, \mathbf{z}_i, \mathbf{z}_j) = [(\mathbb{I}_{y_j=1})\tilde{\ell}(\mathbf{w}, \mathbf{z}_i, \mathbf{z}_j), \tilde{\ell}(\mathbf{w}, \mathbf{z}_i, \mathbf{z}_j)]$ and $f(x_1, x_2) = \frac{x_1}{x_2}$ (Qi et al., 2021).

# B COMPLEXITY FOR SOLVING CPR AND ERM PROBLEMS

In Table 3, we summarize state-of-the-art results for ERM problems and CPR problems, in both centralized setting and federated setting. We cover the cases when the data comes with/without a finite-sum structure. For the CPR problem, we consider the finite-sum form for both the inner function and the outer function.

Here we focuses on non-convex problems. For federated learning in convex/strongly-convex cases, please refer to (Shamir et al., 2014; Li et al., 2019; 2020; Khaled et al., 2020; Karimireddy et al., 2020b; 2021; Mishchenko et al., 2022; Khaled & Jin, 2022) and reference therein.

In Table 3, the $*$ notion indicates that an algorithm matches a known lower bound complexity. The Spider algorithm (Fang et al., 2018) matches the lower bound result in (Zhou & Gu, 2019) for the

Table 3: Comparison for sample complexity on each machine for solving ERM problem and CPR problem to a $\epsilon$-stationary point, i.e., $\mathbb{E}[\|F(\mathbf{w})\|^2] \leq \epsilon^2$. $N$ is the number of machines in federated setting. $n$ is the number of finite-sum components in outer finite-sum setting, which in ERM is the number of all data and in CPR is the number of data on the outer function. $n_{\text{in}}$ denotes the number of finite-sum components for the inner function $g$ when it is of finite-sum structure. In federated learning setting with a finite-sum structure, each machine $i$ has $n_i$ data in the outer function. $*$ indicates the complexity matches a known lower bound.

| | | Sample Complexity | Setting |
|---|---|---|---|
| ERM | Centralized | SGD* $O(1/\epsilon^4)$ (Ghadimi & Lan, 2013) | Expectation |
| | | SPIDER*: $O(\sqrt{n}/\epsilon^2)$ (Fang et al., 2018) | Finite-sum |
| | Federated | PR-SGD*: $O(1/N\epsilon^4)$ (Yu et al., 2019b) | Expectation |
| | | VRL-SGD*: $O(1/N\epsilon^4)$ (Liang et al., 2019) | Expectation |
| | | SCAFFOLD*: $O(1/N\epsilon^4)$ (Karimireddy et al., 2020b) | Expectation |
| | | FedProx: $O(1/N\epsilon^4)$ (Li et al., 2020) | Finite-sum |
| | | Mime: $O(1/N\epsilon^4)$ (Karimireddy et al., 2021) | Finite-sum |
| CPR | Centralized | BSGD: $O(1/\epsilon^6)$ (Hu et al., 2020) | Inner Expectation + Outer Expection |
| | | BSpiderBoost*: $O(1/\epsilon^5)$ (Hu et al., 2020) | Inner Expectation + Outer Expectation |
| | | MSVR: $O(\max(1/\epsilon^4, n/\epsilon^3))$ (Jiang et al., 2022) | Inner Expectation + Outer Finite-sum |
| | | MSVR: $O(n\sqrt{n_{\text{in}}}/\epsilon^2)$ (Jiang et al., 2022) | Inner Finite-sum + Outer Finite-sum |
| | | SOX: $O(n/\epsilon^4)$ (Wang & Yang, 2022b) | Inner Expectation + Outer Finite-sum |
| | Federated | This Work: $O(\max_i n_i/\epsilon^4)$ | Inner Expectation + Outer Finite-sum |

finite-sum setting and the SGD algorithm (Ghadimi & Lan, 2013) matches the lower bound in (Arjevani et al., 2022) for the expectation setting. In finite-sum setting, the federated ERM algorithms, i.e., PR-SGD, VRL-SGD, SCAFFOLD, matches the lower bound in (Woodworth et al., 2020b;a; Glasgow et al., 2022). BSpiderBoost matches the lower bound in (Hu et al., 2020). For CPR problems with a finite-sum structure on the outer function, the tight lower bounds are still unclear. After submitting to ICLR 2023, we noticed a later work Jiang et al. (2022) has propose a MSVR algorithm (in Table 3) that further improves the sample complexities by utilizing variance reduce techniques SVRG and STORM. However, naively implementing MSVR in federated setting would have a much higher communication cost than our algorithm. Actually, even for those ERM algorithms which have used similar variance reduction techniques, it remains an open problem whether any communication-efficient algorithm could be feasible.

## C  ANALYSIS OF FEDX1 FOR OPTIMIZING CPR WITH LINEAR $f$

In this section, we present the analysis of the FedX1 algorithm. For $\mathbf{z} \in \mathcal{S}_1^i$ and $\mathbf{z}' \in \mathcal{S}_2^j$, we define
$$G_1(\mathbf{w}, \mathbf{z}, \mathbf{w}', \mathbf{z}') = \nabla_1 \ell_{ij}(h(\mathbf{w}; \mathbf{z}), h(\mathbf{w}'; \mathbf{z}'))^\top \nabla h(\mathbf{w}; \mathbf{z})$$
$$G_2(\mathbf{w}, \mathbf{z}, \mathbf{w}', \mathbf{z}') = \nabla_2 \ell_{ij}(h(\mathbf{w}, \mathbf{z}), h(\mathbf{w}'; \mathbf{z}'))^\top \nabla h(\mathbf{w}'; \mathbf{z}'). \tag{15}$$
Therefore, the
$$G_{i,k,1}^r = \nabla_1 \ell_{ij}(h(\mathbf{w}_{i,k}^r; \mathbf{z}_{i,k,1}^r), h_{2,\xi}^{r-1}) \nabla h(\mathbf{w}_{i,k}^r; \mathbf{z}_{i,k,1}^r),$$
defined in (3) is equivalent to $G_1(\mathbf{w}_{i,k}^r, \mathbf{z}_{i,k,1}^r, \mathbf{w}_{j,t}^{r-1}, \mathbf{z}_{j,t,2}^{r-1})$, where $h_{2,\xi}^{r-1} = h(\mathbf{w}_{j,t}^{r-1}; \mathbf{z}_{j,t,2}^{r-1})$ is a scored of a randomly sampled data that in computed in the round $r-1$ at machine $j$ and iteration $t$. Technically, notations $j$ and $t$ are associated with $i$ and $k$, but we omit this dependence when the context is clear to simplify notations.

Similarly, the
$$G_{i,k,2}^r = \nabla_2 \ell_{j'i}(h_{1,\zeta}^{r-1}, h(\mathbf{w}_{i,k}^r; \mathbf{z}_{i,k,2}^r),) \nabla h(\mathbf{w}_{i,k}^r; \mathbf{z}_{i,k,2}^r),$$

defined in (5) is equivalent to $G_2(\mathbf{w}_{j',t'}^{r-1}, \mathbf{z}_{j',t',1}^{r-1}, \mathbf{w}_{i,k}^r, \mathbf{z}_{i,k,2}^r)$. Denote

$$\nabla F_i(\mathbf{w}) := \underbrace{\mathbb{E}_{\mathbf{z} \in \mathcal{S}_1^i} \frac{1}{N} \sum_{j=1}^{N} \mathbb{E}_{\mathbf{z}' \in \mathcal{S}_2^j} \nabla_1 \ell_{ij}(h(\mathbf{w}, \mathbf{z}), h(\mathbf{w}, \mathbf{z}')) \nabla h_{\mathbf{w}}(\mathbf{z})}_{\Delta_{i1}} \tag{16}$$

$$+ \underbrace{\mathbb{E}_{\mathbf{z}' \in \mathcal{S}_2^i} \frac{1}{N} \sum_{j=1}^{N} \mathbb{E}_{\mathbf{z} \in \mathcal{S}_1^j} \nabla_2 \ell_{ji}(h(\mathbf{w}, \mathbf{z}), h(\mathbf{w}, \mathbf{z}')) \nabla h(\mathbf{w}, \mathbf{z}')}_{\Delta_{i2}}. \tag{17}$$

We make the following assumptions regarding the CPR with linear $f$ problem, i.e., problem 2.

**Assumption 1.**

- $\ell_{ij}(\cdot)$ *is differentiable,* $L_\ell$*-smooth and* $C_\ell$*-Lipschitz.*

- $h(\cdot, \mathbf{z})$ *is differentiable,* $L_h$*-smooth and* $C_h$*-Lipschitz on* $\mathbf{w}$ *for any* $\mathbf{z} \in \mathcal{S}_1 \cup \mathcal{S}_2$.

- $\mathbb{E}_{\mathbf{z} \in \mathcal{S}_1^i, \mathbf{z}' \in \mathcal{S}_2} \| \nabla \ell_{ij}(h(\mathbf{w}; \mathbf{z}), h(\mathbf{w}; \mathbf{z}')) \nabla h(\mathbf{w}; \mathbf{z}) + \nabla \ell_{ji}(h(\mathbf{w}; \mathbf{z}), h(\mathbf{w}; \mathbf{z}')) \nabla h(\mathbf{w}; \mathbf{z}') - \nabla F_i(\mathbf{w}) \|^2 \leq \sigma^2$.

- $\| \nabla F_i(\mathbf{w}) - \nabla F(\mathbf{w}) \|^2 \leq D^2$.

Under Assumption 1, it follows that $F(\cdot)$ is $L_F$-smooth, with $L_F := 2(L_\ell C_h + C_\ell L_h)$. Similarly, $G_1, G_2$ also Lipschtz in $\mathbf{w}$ with some constant $L_1$ that depend on $C_h, C_\ell, L_\ell, L_h$. Let $\tilde{L} := \max\{L_F, L_1\}$. Basically, we consider well-conditioned problems where $\omega_{i,1} = \frac{N|\mathcal{S}_1^i|}{|\mathcal{S}_2|}$ and $\omega_{i,2} = \frac{N|\mathcal{S}_2^i|}{|\mathcal{S}_2|}$ are of $O(1)$, therefore the above constants are appropriate. Nevertheless, we can also directly consider the FL objective where $\omega_{1,i} = 1$ and $\omega_{2,i} = 1$ similar to existing FL studies for the ERM problem. We re-present Theorem 1 as below.

**Theorem 3.** *Under Assumption 1, by setting* $\eta = O(\frac{N}{R^{2/3}})$ *and* $K = O(\frac{R^{1/3}}{N})$*, Algorithm 2 ensures that*

$$\mathbb{E}[\frac{1}{R} \sum_{r=1}^{R} \|\nabla F(\bar{\mathbf{w}}^r)\|^2] \leq O(\frac{1}{R^{2/3}}). \tag{18}$$

*Proof.* Denote $\tilde{\eta} = \eta K$. Using the $\tilde{L}$-smoothness of $F(\mathbf{w})$, we have

$$F(\bar{\mathbf{w}}^{r+1}) - F(\bar{\mathbf{w}}^r) \le \nabla F(\bar{\mathbf{w}}^r)^\top (\bar{\mathbf{w}}^{r+1} - \bar{\mathbf{w}}^r) + \frac{\tilde{L}}{2} \|\bar{\mathbf{w}}^{r+1} - \bar{\mathbf{w}}^r\|^2$$

$$= -\tilde{\eta} \nabla F(\bar{\mathbf{w}}^r)^\top \left( \frac{1}{NK} \sum_i \sum_k (G_{i,k,1}^r + G_{i,k,2}^r) \right) + \frac{\tilde{L}}{2} \|\bar{\mathbf{w}}^{r+1} - \bar{\mathbf{w}}^r\|^2$$

$$= -\tilde{\eta} (\nabla F(\bar{\mathbf{w}}^r) - \nabla F(\bar{\mathbf{w}}^{r-1}) + \nabla F(\bar{\mathbf{w}}^{r-1}))^\top \left( \frac{1}{NK} \sum_i \sum_k (G_{i,k,1}^r + G_{i,k,2}^r) \right) + \frac{\tilde{L}}{2} \|\bar{\mathbf{w}}^{r+1} - \bar{\mathbf{w}}^r\|^2$$

$$\le \frac{1}{2\tilde{L}} \|\nabla F(\bar{\mathbf{w}}^r) - \nabla F(\bar{\mathbf{w}}^{r-1})\|^2 + 2\tilde{\eta}^2 L \|\frac{1}{NK} \sum_i \sum_k (G_{i,k,1}^r + G_{i,k,2}^r)\|^2$$

$$- \tilde{\eta} \nabla F(\bar{\mathbf{w}}^{r-1})^\top \left( \frac{1}{NK} \sum_i \sum_k (G_{i,k,1}^r + G_{i,k,2}^r) \right) + \frac{L}{2} \|\bar{\mathbf{w}}^{r+1} - \bar{\mathbf{w}}^r\|^2$$

$$\le 2\tilde{\eta}^2 \tilde{L} \|\frac{1}{NK} \sum_i \sum_k (G_{i,k,1}^r + G_{i,k,2}^r)\|^2 + \tilde{L} \|\bar{\mathbf{w}}^{r+1} - \bar{\mathbf{w}}^r\|^2$$

$$- \tilde{\eta} \nabla F(\bar{\mathbf{w}}^{r-1})^\top \left( \frac{1}{NK} \sum_i \sum_k (G_{i,k,1}^r + G_{i,k,2}^r) \right),$$

$$(19)$$

where

$$- \mathbb{E}\left[ \tilde{\eta} \nabla F(\bar{\mathbf{w}}^{r-1})^\top \left( \frac{1}{NK} \sum_i \sum_k (G_{i,k,1}^r + G_{i,k,2}^r) \right) \right]$$

$$= -\mathbb{E}\Big[ \tilde{\eta} \nabla F(\bar{\mathbf{w}}^{r-1})^\top \Big( \frac{1}{NK} \sum_i \sum_k (G_1(\mathbf{w}_{i,k}^r, \mathbf{z}_{i,k,1}^r, \mathbf{w}_{j,t}^{r-1}, \mathbf{z}_{j,t,2}^{r-1}) + G_2(\mathbf{w}_{j',t'}^{r-1}, \mathbf{z}_{j',t',1}^{r-1}, \mathbf{w}_{i,k}^r, \mathbf{z}_{i,k,2}^r)$$

$$- G_1(\bar{\mathbf{w}}^{r-1}, \mathbf{z}_{i,k,1}^r, \mathbf{w}^{r-1}, \mathbf{z}_{j,t,2}^{r-1}) - G_2(\bar{\mathbf{w}}^{r-1}, \mathbf{z}_{j',t',1}^{r-1}, \bar{\mathbf{w}}^{r-1}, \mathbf{z}_{i,k,2}^r)$$

$$+ G_1(\bar{\mathbf{w}}^{r-1}, \mathbf{z}_{i,k,1}^r, \bar{\mathbf{w}}^{r-1}, \mathbf{z}_{j,t,2}^{r-1}) + G_2(\bar{\mathbf{w}}^{r-1}, \mathbf{z}_{j',t',1}^{r-1}, \bar{\mathbf{w}}^{r-1}, \mathbf{z}_{i,k,2}^r))) \Big) \Big]$$

$$\le \frac{\tilde{\eta}}{4} \mathbb{E} \|\nabla F(\bar{\mathbf{w}}^{r-1})\|^2 + 8\tilde{\eta} \tilde{L}^2 \mathbb{E} \|\bar{\mathbf{w}}^r - \bar{\mathbf{w}}^{r-1}\|^2 + 8\tilde{\eta} \tilde{L}^2 \frac{1}{NK} \sum_i \sum_k \mathbb{E} \|\bar{\mathbf{w}}^r - \mathbf{w}_{i,k}^r\|^2$$

$$- \mathbb{E}[\tilde{\eta} \nabla F(\bar{\mathbf{w}}^{r-1})^\top \Big( \frac{1}{NK} \sum_i \sum_k (G_1(\bar{\mathbf{w}}^{r-1}, \mathbf{z}_{i,k,1}^r, \bar{\mathbf{w}}^{r-1}, \mathbf{z}_{j,t,2}^{r-1}) + G_2(\bar{\mathbf{w}}^{r-1}, \mathbf{z}_{j',t',1}^{r-1}, \mathbf{w}^{r-1}, \mathbf{z}_{i,k,2}^r) - \nabla F_i(\mathbf{w}^{r-1}))$$

$$+ \nabla F(\bar{\mathbf{w}}^{r-1}) \Big)]$$

$$= \frac{\tilde{\eta}}{4} \mathbb{E} \|\nabla F(\bar{\mathbf{w}}^{r-1})\|^2 + 8\tilde{\eta} \tilde{L}^2 \mathbb{E} \|\bar{\mathbf{w}}^r - \bar{\mathbf{w}}^{r-1}\|^2 + 8\tilde{\eta} \tilde{L}^2 \frac{1}{NK} \sum_i \sum_k \mathbb{E} \|\bar{\mathbf{w}}^r - \mathbf{w}_{i,k}^r\|^2 - \tilde{\eta} \mathbb{E} \|\nabla F(\bar{\mathbf{w}}^{r-1})\|^2$$

$$\le -\mathbb{E}\left[ \frac{\tilde{\eta}}{2} \|\nabla F(\bar{\mathbf{w}}^{r-1})\|^2 \right] + 8\tilde{\eta} \tilde{L}^2 \mathbb{E} \|\bar{\mathbf{w}}^r - \bar{\mathbf{w}}^{r-1}\|^2 + 8\tilde{\eta} \tilde{L}^2 \frac{1}{NK} \sum_i \sum_k \mathbb{E} \|\bar{\mathbf{w}}^r - \mathbf{w}_{i,k}^r\|^2,$$

$$(20)$$

where the second equality holds because that data samples $\mathbf{z}_{i,k,1}^r, \mathbf{z}_{j,t}^{r-1}, \mathbf{z}_{j',t',1}^{r-1}, \mathbf{z}_{i,k,2}^r$ are independent samples after $\bar{\mathbf{w}}^{r-1}$, therefore

$$\mathbb{E}[(G_1(\bar{\mathbf{w}}^{r-1}, \mathbf{z}_{i,k,1}^r, \bar{\mathbf{w}}^{r-1}, \mathbf{z}_{j,t,2}^{r-1}) + G_2(\bar{\mathbf{w}}^{r-1}, \mathbf{z}_{j',t',1}^{r-1}, \bar{\mathbf{w}}^{r-1}, \mathbf{z}_{i,k,2}^r) - \nabla F_i(\bar{\mathbf{w}}^{r-1})] = \mathbf{0}. \quad (21)$$

To bound the updates of $\bar{\mathbf{w}}^r$ after one round, we have

$$\mathbb{E}\|\bar{\mathbf{w}}^{r+1} - \bar{\mathbf{w}}^r\|^2 = \tilde{\eta}^2 \mathbb{E}\|\frac{1}{NK}\sum_i\sum_k(G_{i,k,1}^r + G_{i,k,2}^r)\|^2$$

$$= \tilde{\eta}^2\mathbb{E}\|\frac{1}{NK}\sum_i\sum_k(G_1(\mathbf{w}_{i,k}^r, \mathbf{z}_{i,k,1}^r, \mathbf{w}_{j,t}^{r-1}, \mathbf{z}_{j,t,2}^{r-1}) + G_2(\mathbf{w}_{j',t'}^{r-1}, \mathbf{z}_{j',t',1}^{r-1}, \mathbf{w}_{i,k}^r, \mathbf{z}_{i,k,2}^r))\|^2$$

$$\leq 5\tilde{\eta}^2\mathbb{E}\left\|\frac{1}{NK}\sum_i\sum_k[G_1(\mathbf{w}_{i,k}^r, \mathbf{z}_{i,k,1}^r, \mathbf{w}_{j,t}^{r-1}, \mathbf{z}_{j,t,2}^{r-1}) + G_2(\mathbf{w}_{j',t'}^{r-1}, \mathbf{z}_{j',t',1}^{r-1}, \mathbf{w}_{i,k}^r, \mathbf{z}_{i,k,2}^r)]\right.$$
$$\left. - \frac{1}{NK}\sum_i\sum_k[G_1(\bar{\mathbf{w}}^r, \mathbf{z}_{i,k,1}^r, \mathbf{w}_{j,t}^{r-1}, \mathbf{z}_{j,t,2}^{r-1}) + G_2(\mathbf{w}_{j',t'}^{r-1}, \mathbf{z}_{j',t',1}^{r-1}, \bar{\mathbf{w}}^r, \mathbf{z}_{i,k,2}^r)]\right\|^2$$

$$+ 5\tilde{\eta}^2\mathbb{E}\left\|\frac{1}{NK}\sum_i\sum_k[G_1(\bar{\mathbf{w}}^r, \mathbf{z}_{i,k,1}^r, \mathbf{w}_{j,t}^{r-1}, \mathbf{z}_{j,t,2}^{r-1}) + G_2(\mathbf{w}_{j',t'}^{r-1}, \mathbf{z}_{j',t',1}^{r-1}, \bar{\mathbf{w}}^r, \mathbf{z}_{i,k,2}^r)]\right.$$
$$\left. - \frac{1}{NK}\sum_i\sum_k[G_1(\bar{\mathbf{w}}^r, \mathbf{z}_{i,k,1}^r, \bar{\mathbf{w}}^{r-1}, \mathbf{z}_{j,t,2}^{r-1}) + G_2(\bar{\mathbf{w}}^{r-1}, \mathbf{z}_{j',t',1}^{r-1}, \bar{\mathbf{w}}^r, \mathbf{z}_{i,k,2}^r)]\right\|$$

$$+ 5\tilde{\eta}^2\mathbb{E}\left\|\frac{1}{NK}\sum_i\sum_k[G_1(\bar{\mathbf{w}}^r, \mathbf{z}_{i,k,1}^r, \bar{\mathbf{w}}^{r-1}, \mathbf{z}_{j,t,2}^{r-1}) + G_2(\bar{\mathbf{w}}^{r-1}, \mathbf{z}_{j',t',1}^{r-1}, \bar{\mathbf{w}}^r, \mathbf{z}_{i,k,2}^r)]\right.$$
$$\left. - \frac{1}{NK}\sum_i\sum_k[G_1(\bar{\mathbf{w}}^{r-1}, \mathbf{z}_{i,k,1}^r, \bar{\mathbf{w}}^{r-1}, \mathbf{z}_{j,t,2}^{r-1}) + G_2(\bar{\mathbf{w}}^{r-1}, \mathbf{z}_{j',t',1}^{r-1}, \bar{\mathbf{w}}^{r-1}, \mathbf{z}_{i,k,2}^r)]\right\|$$

$$+ 5\tilde{\eta}^2\mathbb{E}\left\|\frac{1}{NK}\sum_i\sum_k[G_1(\bar{\mathbf{w}}^{r-1}, \mathbf{z}_{i,k,1}^r, \bar{\mathbf{w}}^{r-1}, \mathbf{z}_{j,t,2}^{r-1}) + G_2(\bar{\mathbf{w}}^{r-1}, \mathbf{z}_{j',t',1}^{r-1}, \bar{\mathbf{w}}^{r-1}, \mathbf{z}_{i,k,2}^r) - \nabla F_i(\bar{\mathbf{w}}^{r-1})]\right\|^2$$

$$+ 5\tilde{\eta}^2\mathbb{E}\left\|\nabla F(\bar{\mathbf{w}}^{r-1})\right\|^2$$

$$\leq 10\tilde{\eta}^2\frac{\tilde{L}^2}{NK}\sum_i\sum_k\mathbb{E}\|\mathbf{w}_{i,k}^r - \bar{\mathbf{w}}^r\|^2 + 10\tilde{\eta}^2\frac{\tilde{L}^2}{NK}\sum_i\sum_k\mathbb{E}\|\mathbf{w}_{i,k}^{r-1} - \bar{\mathbf{w}}^{r-1}\|^2 + 10\tilde{\eta}^2\tilde{L}^2\mathbb{E}\|\bar{\mathbf{w}}^r - \bar{\mathbf{w}}^{r-1}\|^2$$

$$+ 10\tilde{\eta}^2\frac{\sigma^2}{NK} + 10\tilde{\eta}^2\mathbb{E}\|F(\bar{\mathbf{w}}^{r-1})\|^2.$$
$$(22)$$

Thus,

$$\frac{1}{R}\sum_r\mathbb{E}\|\bar{\mathbf{w}}^{r+1} - \bar{\mathbf{w}}^r\|^2 \leq \frac{1}{R}\sum_r\left[40\tilde{\eta}^2\tilde{L}^2\frac{1}{NK}\sum_i\sum_k\mathbb{E}\|\mathbf{w}_{i,k}^r - \bar{\mathbf{w}}^r\|^2 + 20\tilde{\eta}^2\frac{\sigma^2}{NK} + 20\tilde{\eta}^2\mathbb{E}\|F(\bar{\mathbf{w}}^{r-1})\|^2\right].$$
$$(23)$$

Then we bound the updates in one round and one machine as

$$\|\bar{\mathbf{w}}^r - \mathbf{w}_{i,k}^r\|^2 = \|\mathbf{w}_{i,k-1}^r - \eta(G_1(\mathbf{w}_{i,k-1}^r, \mathbf{z}_{i,k-1,1}^r, \mathbf{w}_{j,t}^{r-1}, \mathbf{z}_{j,t,2}^{r-1}) + G_2(\mathbf{w}_{j',t'}^{r-1}, \mathbf{z}_{j',t',1}^{r-1}, \mathbf{w}_{i,k-1}^r, \mathbf{z}_{i,k-1,2}^r)) - \bar{\mathbf{w}}^r\|^2$$

$$\leq \|\mathbf{w}_{i,k-1}^r - \bar{\mathbf{w}}^r - \eta(G_1(\bar{\mathbf{w}}^{r-1}, \mathbf{z}_{i,k-1,1}^r, \bar{\mathbf{w}}^{r-1}, \mathbf{z}_{j,t,2}^{r-1}) + G_2(\bar{\mathbf{w}}^{r-1}, \mathbf{z}_{j',t',1}^{r-1}, \bar{\mathbf{w}}^{r-1}, \mathbf{z}_{i,k-1,2}^r))$$
$$+ \eta([G_1(\bar{\mathbf{w}}^{r-1}, \mathbf{z}_{i,k-1,1}^r, \bar{\mathbf{w}}^{r-1}, \mathbf{z}_{j,t,2}^{r-1}) + G_2(\bar{\mathbf{w}}^{r-1}, \mathbf{z}_{j',t',1}^{r-1}, \bar{\mathbf{w}}^{r-1}, \mathbf{z}_{i,k-1,2}^r)]$$
$$- [G_1(\bar{\mathbf{w}}^r, \mathbf{z}_{i,k-1,1}^r, \bar{\mathbf{w}}^{r-1}, \mathbf{z}_{j,t,2}^{r-1}) + G_2(\bar{\mathbf{w}}^{r-1}, \mathbf{z}_{j',t',1}^{r-1}, \bar{\mathbf{w}}^r, \mathbf{z}_{i,k-1,2}^r)])$$
$$+ \eta([G_1(\bar{\mathbf{w}}^r, \mathbf{z}_{i,k-1,1}^r, \bar{\mathbf{w}}^{r-1}, \mathbf{z}_{j,t,2}^{r-1}) + G_2(\bar{\mathbf{w}}^{r-1}, \mathbf{z}_{j',t',1}^{r-1}, \bar{\mathbf{w}}^r, \mathbf{z}_{i,k-1,2}^r)]$$
$$- [G_1(\mathbf{w}_{i,k-1}^r, \mathbf{z}_{i,k-1,1}^r, \bar{\mathbf{w}}^{r-1}, \mathbf{z}_{j,t,2}^{r-1}) + G_2(\bar{\mathbf{w}}^{r-1}, \mathbf{z}_{j',t',1}^{r-1}, \mathbf{w}_{i,k-1}^r, \mathbf{z}_{i,k-1,2}^r)])$$
$$+ \eta([G_1(\mathbf{w}_{i,k-1}^r, \mathbf{z}_{i,k-1,1}^r, \bar{\mathbf{w}}^{r-1}, \mathbf{z}_{j,t,2}^{r-1}) + G_2(\bar{\mathbf{w}}^{r-1}, \mathbf{z}_{j',t',1}^{r-1}, \mathbf{w}_{i,k-1}^r, \mathbf{z}_{i,k-1,2}^r)]$$
$$- [G_1(\mathbf{w}_{i,k-1}^r, \mathbf{z}_{i,k-1,1}^r, \mathbf{w}_{j,t}^{r-1}, \mathbf{z}_{j,t,2}^{r-1}) + G_2(\mathbf{w}_{j',t'}^{r-1}, \mathbf{z}_{j',t',1}^{r-1}, \mathbf{w}_{i,k-1}^r, \mathbf{z}_{i,k-1,2}^r)])\|^2$$
$$(24)$$

Using Young's inequality, we continue this inequality as

$$\mathbb{E}\|\bar{\mathbf{w}}^r - \mathbf{w}_{i,k}^r\|^2$$

$$\leq (1 + \frac{1}{4K})\mathbb{E}\|\mathbf{w}_{i,k-1}^r - \bar{\mathbf{w}}^r - \eta(G_1(\bar{\mathbf{w}}^{r-1}, \mathbf{z}_{i,k-1,1}^r, \bar{\mathbf{w}}^{r-1}, \mathbf{z}_{j,t,2}^{r-1}) + G_2(\bar{\mathbf{w}}^{r-1}, \mathbf{z}_{j',t',1}^{r-1}, \bar{\mathbf{w}}^{r-1}, \mathbf{z}_{i,k-1,2}^r))\|^2$$

$$+ (4K+1)\eta^2\mathbb{E}\|([G_1(\bar{\mathbf{w}}^{r-1}, \mathbf{z}_{i,k-1,1}^r, \bar{\mathbf{w}}^{r-1}, \mathbf{z}_{j,t,2}^r) + G_2(\bar{\mathbf{w}}^{r-1}, \mathbf{z}_{j',t',1}^{r-1}, \bar{\mathbf{w}}^{r-1}, \mathbf{z}_{i,k-1,2}^r)]$$

$$- [G_1(\bar{\mathbf{w}}^r, \mathbf{z}_{i,k-1,1}^r, \bar{\mathbf{w}}^{r-1}, \mathbf{z}_{j,t,2}^{r-1}) + G_2(\bar{\mathbf{w}}^{r-1}, \mathbf{z}_{j',t',1}^{r-1}, \bar{\mathbf{w}}^r, \mathbf{z}_{i,k-1,2}^r)])$$

$$+ ([G_1(\bar{\mathbf{w}}^r, \mathbf{z}_{i,k-1,1}^r, \bar{\mathbf{w}}^{r-1}, \mathbf{z}_{j,t,2}^r) + G_2(\bar{\mathbf{w}}^{r-1}, \mathbf{z}_{j',t',1}^{r-1}, \bar{\mathbf{w}}^r, \mathbf{z}_{i,k-1,2}^r)]$$

$$- [G_1(\mathbf{w}_{i,k-1}^r, \mathbf{z}_{i,k-1,1}^r, \bar{\mathbf{w}}^{r-1}, \mathbf{z}_{j,t,2}^{r-1}) + G_2(\bar{\mathbf{w}}^{r-1}, \mathbf{z}_{j',t',1}^{r-1}, \mathbf{w}_{i,k-1}^r, \mathbf{z}_{i,k-1,2}^r)])$$

$$+ ([G_1(\mathbf{w}_{i,k-1}^r, \mathbf{z}_{i,k-1,1}^r, \bar{\mathbf{w}}^{r-1}, \mathbf{z}_{j,t,2}^{r-1}) + G_2(\bar{\mathbf{w}}^{r-1}, \mathbf{z}_{j',t',1}^{r-1}, \mathbf{w}_{i,k-1}^r, \mathbf{z}_{i,k-1,2}^r)]$$

$$- [G_1(\mathbf{w}_{i,k-1}^r, \mathbf{z}_{i,k-1,1}^r, \mathbf{w}_{j,t}^{r-1}, \mathbf{z}_{j,t,2}^{r-1}) + G_2(\mathbf{w}_{j',t'}^{r-1}, \mathbf{z}_{j',t',1}^{r-1}, \mathbf{w}_{i,k-1}^r, \mathbf{z}_{i,k-1,2}^r)])\|^2$$

$$\leq (1 + \frac{1}{4K})\mathbb{E}\|\mathbf{w}_{i,k-1}^r - \bar{\mathbf{w}}^r - \eta(G_1(\bar{\mathbf{w}}^{r-1}, \mathbf{z}_{i,k-1,1}^r, \bar{\mathbf{w}}^{r-1}, \mathbf{z}_{j,t,2}^{r-1}) + G_2(\bar{\mathbf{w}}^{r-1}, \mathbf{z}_{j',t',1}^{r-1}, \bar{\mathbf{w}}^{r-1}, \mathbf{z}_{i,k-1,2}^r))\|^2$$

$$+ 18K\eta^2\tilde{L}^2\mathbb{E}(\|\bar{\mathbf{w}}^{r-1} - \bar{\mathbf{w}}^r\|^2 + \|\bar{\mathbf{w}}^r - \mathbf{w}_{i,k-1}^r\|^2 + \|\bar{\mathbf{w}}^{r-1} - \bar{\mathbf{w}}_{j,t}^{r-1}\|^2)$$

$$\leq (1 + \frac{1}{K})\mathbb{E}\|\mathbf{w}_{i,k-1}^r - \bar{\mathbf{w}}^r - \eta\nabla F_i(\bar{\mathbf{w}}^{r-1})\|^2 + 5\eta^2 K\sigma^2$$

$$+ 18K\eta^2\tilde{L}^2\mathbb{E}(\|\bar{\mathbf{w}}^{r-1} - \bar{\mathbf{w}}^r\|^2 + \|\bar{\mathbf{w}}^r - \mathbf{w}_{i,k-1}^r\|^2 + \|\bar{\mathbf{w}}^{r-1} - \bar{\mathbf{w}}_{j,t}^{r-1}\|^2)$$

$$\leq (1 + \frac{2}{K})\mathbb{E}\|\mathbf{w}_{i,k-1}^r - \bar{\mathbf{w}}^r\|^2 + 4K\eta^2\mathbb{E}\|\nabla F_i(\bar{\mathbf{w}}^{r-1})\|^2 + 5K\eta^2\sigma^2$$

$$+ 18K\eta^2\tilde{L}^2\mathbb{E}(\|\bar{\mathbf{w}}^{r-1} - \bar{\mathbf{w}}^r\|^2 + \|\bar{\mathbf{w}}^r - \mathbf{w}_{i,k-1}^r\|^2 + \|\bar{\mathbf{w}}^{r-1} - \bar{\mathbf{w}}_{j,t}^{r-1}\|^2)$$

$$\leq (1 + \frac{2}{K})\mathbb{E}\|\mathbf{w}_{i,k-1}^r - \bar{\mathbf{w}}^r\|^2 + 8K\eta^2\mathbb{E}\|\nabla F(\bar{\mathbf{w}}^{r-1})\|^2 + 8K\eta^2(D^2 + \sigma^2)$$

$$+ 18K\eta^2\tilde{L}^2\mathbb{E}(\|\bar{\mathbf{w}}^{r-1} - \bar{\mathbf{w}}^r\|^2 + \|\bar{\mathbf{w}}^r - \mathbf{w}_{i,k-1}^r\|^2 + \|\bar{\mathbf{w}}^{r-1} - \bar{\mathbf{w}}_{j,t}^{r-1}\|^2)$$

$$= (1 + \frac{2}{K} + 18K\eta^2\tilde{L}^2)\mathbb{E}\|\mathbf{w}_{i,k-1}^r - \bar{\mathbf{w}}^r\|^2 + 8K\eta^2\mathbb{E}\|\nabla F(\bar{\mathbf{w}}^{r-1})\|^2 + 8K\eta^2(D^2 + \sigma^2)$$

$$+ 18K\eta^2\tilde{L}^2\mathbb{E}(\|\bar{\mathbf{w}}^{r-1} - \bar{\mathbf{w}}^r\|^2 + \|\bar{\mathbf{w}}^{r-1} - \bar{\mathbf{w}}_{j,t}^{r-1}\|^2).$$

$$(25)$$

Thus,

$$\mathbb{E}\|\bar{\mathbf{w}}^r - \mathbf{w}_{i,k}^r\|^2$$

$$\leq \left(8K\eta^2\mathbb{E}\|\nabla F(\bar{\mathbf{w}}^{r-1})\|^2 + 8K\eta^2(D^2 + \sigma^2) + 18K\eta^2\tilde{L}^2\mathbb{E}[\|\bar{\mathbf{w}}^{r-1} - \bar{\mathbf{w}}^r\|^2\right.$$

$$\left. + \|\bar{\mathbf{w}}^{r-1} - \bar{\mathbf{w}}_{j,t}^{r-1}\|^2]\right)\left(\sum_{m=0}^{k-1}(1 + \frac{2}{K} + 18K\eta^2)^m\right)$$

$$\leq (8K\eta^2\mathbb{E}\|\nabla F(\bar{\mathbf{w}}^{r-1})\|^2 + 8K\eta^2(D^2 + \sigma^2) + 18K\eta^2\tilde{L}^2\mathbb{E}(\|\bar{\mathbf{w}}^{r-1} - \bar{\mathbf{w}}^r\|^2 + \|\bar{\mathbf{w}}^{r-1} - \bar{\mathbf{w}}_{j,t}^{r-1}\|^2))5K$$

$$\leq 40K^2\eta^2\mathbb{E}\|\nabla F(\bar{\mathbf{w}}^{r-1})\|^2 + 40K^2\eta^2(D^2 + \sigma^2) + 100K^2\eta^2\tilde{L}^2\mathbb{E}(\|\bar{\mathbf{w}}^{r-1} - \bar{\mathbf{w}}^r\|^2 + \|\bar{\mathbf{w}}^{r-1} - \bar{\mathbf{w}}_{j,t}^{r-1}\|^2),$$

$$(26)$$

where the second inequality is due to $18K\eta^2 \leq \frac{1}{K}$.

Then,

$$\frac{1}{RNK}\sum_{r=1}^R\sum_{i=1}^N\sum_{k=1}^K\mathbb{E}\|\bar{\mathbf{w}}^r - \mathbf{w}_{i,k}^r\|^2 \leq 80K^2\eta^2\mathbb{E}\|\nabla F(\bar{\mathbf{w}}^{r-1})\|^2 + 200K^2\eta^2(D^2 + \sigma^2), \quad (27)$$

and

$$\frac{1}{R}\sum_r\mathbb{E}\|\bar{\mathbf{w}}^{r+1} - \bar{\mathbf{w}}^r\|^2 \leq 80K^2\eta^2\mathbb{E}\|\nabla F(\bar{\mathbf{w}}^{r-1})\|^2 + 80\tilde{\eta}^2K^2\eta^2(D^2 + \sigma^2) + 20\tilde{\eta}^2\frac{\sigma^2}{NK}. \quad (28)$$

Recalling (67) and (20), we obtain

$$\frac{1}{R}\sum_{r=1}^R\mathbb{E}\|F(\bar{\mathbf{w}}_r)\|^2 \leq O\left(\frac{2(F(\bar{\mathbf{w}}_0) - F_*)}{\tilde{\eta}R} + \tilde{\eta}^2(D^2 + \sigma^2) + 40\tilde{\eta}\frac{\sigma^2}{NK}\right). \quad (29)$$

If we set $\eta = O(N\epsilon^2)$, $K = O(1/N\epsilon)$, thus $\tilde{\eta} = O(\epsilon)$, to ensure $\frac{1}{R} \sum_{r=1}^{R} \mathbb{E}\|F(\bar{\mathbf{w}}_r)\|^2 \leq \epsilon^2$, it takes communication rounds of $R = O(\frac{1}{\epsilon^3})$, and sample complexity on each machine $O(\frac{1}{N\epsilon^4})$. $\qquad\square$

# D    FEDX2 FOR OPTIMIZING CPR WITH NON-LINEAR $f$

In this section, we define the following notations:
$$
\begin{aligned}
G_{i,1}(\mathbf{w}_1, \mathbf{z}_1, \mathbf{u}, \mathbf{w}_2, \mathbf{z}_2) &= \nabla f_i(\mathbf{u})\nabla\ell(h(\mathbf{w}_1, \mathbf{z}_1), h(\mathbf{w}_2, \mathbf{z}_2))\nabla h(\mathbf{w}_1, \mathbf{z}_1) \\
G_{i,2}(\mathbf{w}_1, \mathbf{z}_1, \mathbf{u}, \mathbf{w}_2, \mathbf{z}_2) &= -\nabla f_i(\mathbf{u})\nabla\ell(h(\mathbf{w}_1, \mathbf{z}_1), h(\mathbf{w}_2, \mathbf{z}_2))\nabla h(\mathbf{w}_2, \mathbf{z}_2).
\end{aligned}
\tag{30}
$$
Denote
$$
\nabla F_i(\mathbf{w}) := \underbrace{\mathbb{E}_{\mathbf{z}\in\mathcal{S}_1^i} \frac{1}{N} \sum_{j=1}^{N} \mathbb{E}_{\mathbf{z}'\in\mathcal{S}_2^j} \nabla f_i(g(\mathbf{w}; \mathbf{z}, \mathcal{S}_2))\nabla_1\ell_j(h(\mathbf{w}; \mathbf{z}), h(\mathbf{w}; \mathbf{z}'))\nabla h(\mathbf{w}; \mathbf{z})}_{\Delta_{i1}} \tag{31}
$$
$$
+ \underbrace{\mathbb{E}_{\mathbf{z}'\in\mathcal{S}_2^i} \frac{1}{N} \sum_{j=1}^{N} \mathbb{E}_{\mathbf{z}\in\mathcal{S}_1^j} \nabla f_j(g(\mathbf{w}; \mathbf{z}, \mathcal{S}_2))\nabla_2\ell_i(h(\mathbf{w}; \mathbf{z}), h(\mathbf{w}; \mathbf{z}'))\nabla h(\mathbf{w}; \mathbf{z}')}_{\Delta_{i2}}. \tag{32}
$$
We make the following assumptions regarding the CPR with non-linear $f$, i.e., problem (6).

**Assumption 2.**

- $\ell_j(\cdot)$ is differentiable, $L_\ell$-smooth and $C_\ell$-Lipschitz. $|\ell(\cdot)| \leq C_0$.

- $f_i(\cdot)$ is differentiable, $L_f$-smooth and $C_f$-Lipschitz.

- $h(\cdot, \mathbf{z})$ is differentiable, $L_h$-smooth and $C_h$-Lipschitz on $\mathbf{w}$ for any $\mathbf{z} \in \mathcal{S}_1 \cup \mathcal{S}_2$.

- $\mathbb{E}_{\mathbf{z}\in\mathcal{S}_1^i, \mathbf{z}'\in\mathcal{S}_2}\|\nabla f_i(g(\mathbf{w}; \mathbf{z}, \mathcal{S}_2))\nabla\ell_j(h(\mathbf{w}; \mathbf{z}), h(\mathbf{w}; \mathbf{z}'))\nabla h(\mathbf{w}; \mathbf{z})$ $\qquad +$
  $\nabla f_i(g(\mathbf{w}; \mathbf{z}, \mathcal{S}_2))\nabla\ell_j(h(\mathbf{w}; \mathbf{z}), h(\mathbf{w}; \mathbf{z}'))\nabla h(\mathbf{w}; \mathbf{z}) - \nabla F_i(\mathbf{w})\|^2 \leq \sigma^2$.

- $\|\nabla F_i(\mathbf{w}) - \nabla F(\mathbf{w})\|^2 \leq D^2$.

Based on this assumption, it follows that $G_{i,1}, G_{i,2}$ are Lipschitz with some constant modulus $C_1$ and are bounded by $C_2$, $F$ is $L_F$-smooth, where $C_1, C_2, L_F$ are some proper constants depend on Assumption 2. We denote $\tilde{L} = \max\{C_1, C_2, L_F\}$ to simplify notations.

## D.1    ANALYSIS OF THE MOVING AVERAGE ESTIMATOR $\mathbf{u}$

For $\mathbf{z}_1 \in \mathcal{S}_1^i, \mathbf{z}_2 \in \mathcal{S}_2^j$, define $g(\mathbf{w}_1, \mathbf{z}_1, \mathbf{w}_2, \mathbf{z}_2) = \ell_j(h(\mathbf{w}_1; \mathbf{z}_1), h(\mathbf{w}_2, \mathbf{z}_2))$ and for $\mathbf{z}_1 \in \mathcal{S}_1^i$, we define
$$
g(\mathbf{w}_1, \mathbf{z}_1, \mathbf{w}_2, \mathcal{S}_2) = \frac{1}{N} \sum_{j=1}^{N} \mathbb{E}_{\mathbf{z}'\in\mathcal{S}_2^j} \ell_j(h(\mathbf{w}_1; \mathbf{z}_1), h(\mathbf{w}_2, \mathbf{z}')) \tag{33}
$$

**Lemma 1.** *Under Assumption 2, the moving average estimator* $\mathbf{u}$ *satisfies*

$$\frac{1}{N}\sum_{i=1}^{N}\frac{1}{|\mathcal{S}_1^i|}\sum_{\mathbf{z}\in|\mathcal{S}_1^i|}\mathbb{E}\|\mathbf{u}_{i,k}^r(\mathbf{z})-g(\bar{\mathbf{w}}_k^r,\mathbf{z},\bar{\mathbf{w}}_k^r,\mathcal{S}_2)\|^2$$

$$\leq(1-\frac{\gamma}{4|\mathcal{S}_1^i|})\frac{1}{N}\sum_{i=1}^{N}\frac{1}{|\mathcal{S}_1^i|}\sum_{\mathbf{z}\in|\mathcal{S}_1^i|}[\mathbb{E}\|\mathbf{u}_{i,k-1}^r(\mathbf{z})-g(\bar{\mathbf{w}}_{k-1}^r,\mathbf{z},\bar{\mathbf{w}}_{k-1}^r,\mathcal{S}_2)\|^2$$

$$+\frac{\gamma\beta^2K^2C_0}{|\mathcal{S}_1^i|}+2\frac{\gamma^2}{|\mathcal{S}_1^i|}(\sigma^2+C_0^2)$$

$$+(1+\frac{4|\mathcal{S}_1^i|}{\gamma})\tilde{L}^2\|\bar{\mathbf{w}}_{k-1}^r-\bar{\mathbf{w}}_k^r\|^2]+2\gamma^2\|\bar{\mathbf{w}}^r-\bar{\mathbf{w}}^{r-1}\|^2+2\gamma^2\|\bar{\mathbf{w}}^r-\mathbf{w}_{i,k}^r\|^2+2\|\bar{\mathbf{w}}^r-\bar{\mathbf{w}}_k^r\|^2.$$

$$(34)$$

*Proof.* By update rules, we have

$$\mathbf{u}_{i,k}^r(\mathbf{z})=\begin{cases}\mathbf{u}_{i,k-1}^r(\mathbf{z})-\gamma(\mathbf{u}_{i,k-1}^r(\mathbf{z})-\ell(h(\mathbf{w}_{i,k}^r;\mathbf{z}_{i,k,1}^r),h(\mathbf{w}_{j,t}^{r-1};\mathbf{z}_{j,t,2}^{r-1})))&\mathbf{z}=\mathbf{z}_{i,k,1}^r\\\mathbf{u}_{i,k-1}^r(\mathbf{z})&\mathbf{z}\neq\mathbf{z}_{i,k,1}^r.\end{cases}\quad(35)$$

Or equivalently,

$$\mathbf{u}_{i,k}^r(\mathbf{z})=\begin{cases}\mathbf{u}_{i,k-1}^r(\mathbf{z})-\gamma(\mathbf{u}_{i,k-1}^r(\mathbf{z})-g(\mathbf{w}_{i,k}^r,\mathbf{z}_{i,k,1}^r,\mathbf{w}_{j,t}^{r-1},\mathbf{z}_{j,t,2}^{r-1}))&\mathbf{z}=\mathbf{z}_{i,k,1}^r\\\mathbf{u}_{i,k-1}^r(\mathbf{z})&\mathbf{z}\neq\mathbf{z}_{i,k,1}^r\end{cases}\quad(36)$$

Define $\bar{\mathbf{u}}_k^r=(\mathbf{u}_{1,k}^r,\mathbf{u}_{2,k}^r,...,\mathbf{u}_{N,k}^r)$, $\bar{\mathbf{w}}_k^r=\frac{1}{N}\sum_{i=1}^{N}\mathbf{w}_{i,k}^r$, and

$$\phi_k^r(\bar{\mathbf{u}}_k^r)=\frac{1}{2N}\sum_{i=1}^{N}\frac{1}{|\mathcal{S}_i|}\sum_{\mathbf{z}\in\mathcal{S}_1^i}\|\mathbf{u}_{i,k}^r(\mathbf{z})-g(\bar{\mathbf{w}}_k^r,\mathbf{z},\bar{\mathbf{w}}_k^r,\mathcal{S}_2)\|^2.\quad(37)$$

Then it follows that

$$\frac{1}{2}\phi_k^r(\bar{\mathbf{u}}_k^r)=\frac{1}{2N}\sum_{i=1}^{N}\frac{1}{|\mathcal{S}_1^i|}\sum_{\mathbf{z}\in|\mathcal{S}_1^i|}\mathbb{E}\|\mathbf{u}_{i,k}^r(\mathbf{z})-g(\bar{\mathbf{w}}_k^r,\mathbf{z},\bar{\mathbf{w}}_k^r,\mathcal{S}_2)\|^2$$

$$=\frac{1}{N}\sum_{i}\frac{1}{|\mathcal{S}_i|}\sum_{\mathbf{z}\in|\mathcal{S}_1^i|}\mathbb{E}\Big[\frac{1}{2}\|\mathbf{u}_{i,k-1}^r(\mathbf{z})-g(\bar{\mathbf{w}}_k^r,\mathbf{z},\bar{\mathbf{w}}_k^r,\mathcal{S}_2)\|^2+\langle\mathbf{u}_{i,k-1}^r(\mathbf{z})-g(\bar{\mathbf{w}}_k^r,\mathbf{z},\bar{\mathbf{w}}_k^r,\mathcal{S}_2),\mathbf{u}_{i,k}^r(\mathbf{z})-\mathbf{u}_{i,k-1}^r(\mathbf{z})\rangle$$

$$+\frac{1}{2}\|\mathbf{u}_{i,k}^r(\mathbf{z})-\mathbf{u}_{i,k-1}^r(\mathbf{z})\|^2\Big]$$

$$=\frac{1}{N}\sum_{i}\frac{1}{|\mathcal{S}_i|}\sum_{\mathbf{z}\in\mathcal{S}_1^i}\mathbb{E}\Big[\frac{1}{2}\|\mathbf{u}_{i,k-1}^r(\mathbf{z})-g(\bar{\mathbf{w}}_k^r,\mathbf{z},\bar{\mathbf{w}}_k^r,\mathcal{S}_2)\|^2$$

$$+\frac{1}{|\mathcal{S}_1^i|}\langle\mathbf{u}_{i,k-1}^r(\mathbf{z}_{i,k,1}^r)-g(\bar{\mathbf{w}}_k^r,\mathbf{z},\bar{\mathbf{w}}_k^r,\mathcal{S}_2),\mathbf{u}_{i,k}^r(\mathbf{z}_{i,k,1}^r)-\mathbf{u}_{i,k-1}^r(\mathbf{z}_{i,k,1}^r)\rangle$$

$$+\frac{1}{2|\mathcal{S}_1^i|}\|\mathbf{u}_{i,k}^r(\mathbf{z}_{i,k,1}^r)-\mathbf{u}_{i,k-1}^r(\mathbf{z}_{i,k,1}^r)\|^2\Big]$$

$$=\frac{1}{N}\sum_{i}\frac{1}{|\mathcal{S}_i|}\sum_{\mathbf{z}\in\mathcal{S}_1^i}\mathbb{E}\Big[\frac{1}{2}\|\mathbf{u}_{i,k-1}^r(\mathbf{z})-g(\bar{\mathbf{w}}_k^r,\mathbf{z},\bar{\mathbf{w}}_k^r,\mathcal{S}_2)\|^2$$

$$+\frac{1}{|\mathcal{S}_1^i|}\mathbb{E}\langle\mathbf{u}_{i,k-1}^r(\mathbf{z}_{i,k,1}^r)-g(\mathbf{w}_{i,k}^r,\mathbf{z}_{i,k,1}^r,\mathbf{w}_{j,t}^{r-1},\mathbf{z}_{j,t,2}^{r-1}),\mathbf{u}_{i,k}^r(\mathbf{z}_{i,k,1}^r)-\mathbf{u}_{i,k-1}^r(\mathbf{z}_{i,k,1}^r)\rangle$$

$$+\frac{1}{|\mathcal{S}_1^i|}\mathbb{E}\langle g(\mathbf{w}_{i,k}^r,\mathbf{z}_{i,k,1}^r,\mathbf{w}_{j,t}^{r-1},\mathbf{z}_{j,t,2}^{r-1})-g(\bar{\mathbf{w}}_k^r,\mathbf{z}_{i,k,1}^r,\bar{\mathbf{w}}_k^r,\mathcal{S}_2),\mathbf{u}_{i,k}^r(\mathbf{z}_{i,k,1}^r)-\mathbf{u}_{i,k-1}^r(\mathbf{z}_{i,k,1}^r)\rangle$$

$$+\frac{1}{2|\mathcal{S}_i|}\mathbb{E}\|\mathbf{u}_{i,k}^r(\mathbf{z}_{i,k,1}^r)-\mathbf{u}_{i,k-1}^r(\mathbf{z}_{i,k,1}^r)\|^2\Big],$$

$$(38)$$

where
$$\langle \mathbf{u}_{i,k-1}^r(\mathbf{z}_{i,k,1}^r) - g(\mathbf{w}_{i,k}^r, \mathbf{z}_{i,k,1}^r, \mathbf{w}_{j,t}^{r-1}, \mathbf{z}_{j,t,2}^{r-1}), \mathbf{u}_{i,k}^r(\mathbf{z}_{i,k,1}^r) - \mathbf{u}_{i,k-1}^r(\mathbf{z}_{i,k,1}^r) \rangle$$

$$= \langle \mathbf{u}_{i,k-1}^r(\mathbf{z}_{i,k,1}^r) - g(\mathbf{w}_{i,k}^r, \mathbf{z}_{i,k,1}^r, \mathbf{w}_{j,t}^{r-1}, \mathbf{z}_{j,t,2}^{r-1}), g(\bar{\mathbf{w}}_k^r, \mathbf{z}_{i,k,1}^r, \bar{\mathbf{w}}_k^r, \mathcal{S}_2) - \mathbf{u}_{i,k-1}^r(\mathbf{z}_{i,k,1}^r) \rangle$$

$$+ \langle \mathbf{u}_{i,k-1}^r(\mathbf{z}_{i,k,1}^r) - g(\mathbf{w}_{i,k}^r, \mathbf{z}_{i,k,1}^r, \mathbf{w}_{j,t}^{r-1}, \mathbf{z}_{j,t,2}^{r-1}), \mathbf{u}_{i,k}^r(\mathbf{z}_{i,k,1}^r) - g(\bar{\mathbf{w}}_k^r, \mathbf{z}_{i,k,1}^r, \bar{\mathbf{w}}_k^r, \mathcal{S}_2) \rangle$$

$$= \langle \mathbf{u}_{i,k-1}^r(\mathbf{z}_{i,k,1}^r) - g(\mathbf{w}_{i,k}^r, \mathbf{z}_{i,k,1}^r, \mathbf{w}_{j,t}^{r-1}, \mathbf{z}_{j,t,2}^{r-1}), g(\bar{\mathbf{w}}_k^r, \mathbf{z}_{i,k,1}^r, \bar{\mathbf{w}}_k^r, \mathcal{S}_2) - \mathbf{u}_{i,k-1}^r(\mathbf{z}_{i,k,1}^r) \rangle$$

$$+ \frac{1}{\gamma} \langle \mathbf{u}_{i,k-1}^r(\mathbf{z}_{i,k,1}^r) - \mathbf{u}_{i,k}^r(\mathbf{z}_{i,k,1}^r), \mathbf{u}_{i,k}^r(\mathbf{z}_{i,k,1}^r) - g(\bar{\mathbf{w}}_k^r, \mathbf{z}_{i,k,1}^r, \bar{\mathbf{w}}_k^r, \mathcal{S}_2) \rangle$$

$$\leq \langle \mathbf{u}_{i,k-1}^r(\mathbf{z}_{i,k,1}^r) - g(\mathbf{w}_{i,k}^r, \mathbf{z}_{i,k,1}^r, \mathbf{w}_{j,t}^{r-1}, \mathbf{z}_{j,t,2}^{r-1}), g(\bar{\mathbf{w}}_k^r, \mathbf{z}_{i,k,1}^r, \bar{\mathbf{w}}_{i,k}^r, \mathcal{S}_2) - \mathbf{u}_{i,k-1}^r(\mathbf{z}_{i,k,1}^r) \rangle$$

$$+ \frac{1}{2\gamma} (\|\mathbf{u}_{i,k-1}^r(\mathbf{z}_{i,k,1}^r) - g(\bar{\mathbf{w}}_k^r, \mathbf{z}_{i,k,1}^r, \bar{\mathbf{w}}_k^r, \mathcal{S}_2)\|^2 - \|\mathbf{u}_{i,k}^r(\mathbf{z}_{i,k,1}^r) - \mathbf{u}_{i,k-1}^r(\mathbf{z}_{i,k,1}^r)\|^2 - \|\mathbf{u}_{i,k}^r(\mathbf{z}_{i,k,1}^r)$$

$$- g(\bar{\mathbf{w}}_k^r, \mathbf{z}_{i,k,1}^r, \bar{\mathbf{w}}_k^r, \mathcal{S}_2)\|^2)$$

$$\tag{39}$$

If $\gamma \leq \frac{1}{9}$, we have

$$- \frac{1}{2} \left( \frac{1}{\gamma} - 1 - \frac{\gamma+1}{4\gamma} \right) \|\mathbf{u}_{i,k}^r(\mathbf{z}_{i,k,1}^r) - \mathbf{u}_{i,k-1}^r(\mathbf{z}_{i,k,1}^r)\|^2$$

$$+ \langle g(\mathbf{w}_{i,k}^r, \mathbf{z}_{i,k,1}^r, \mathbf{w}_{j,t}^{r-1}, \mathbf{z}_{j,t,2}^{r-1}) - g(\bar{\mathbf{w}}_k^r, \mathbf{z}_{i,k,1}^r, \bar{\mathbf{w}}_k^r, \mathcal{S}_2), \mathbf{u}_{i,k}^r(\mathbf{z}_{i,k,1}^r) - \mathbf{u}_{i,k-1}^r(\mathbf{z}_{i,k,1}^r) \rangle$$

$$\leq - \frac{1}{4\gamma} \|\mathbf{u}_{i,k}^r(\mathbf{z}_{i,k,1}^r) - \mathbf{u}_{i,k-1}^r(\mathbf{z}_{i,k,1}^r)\|^2 + \gamma \|g(\mathbf{w}_{i,k}^r, \mathbf{z}_{i,k,1}^r, \mathbf{w}_{j,t}^{r-1}, \mathbf{z}_{j,t,2}^{r-1}) - g(\bar{\mathbf{w}}_k^r, \mathbf{z}_{i,k,1}^r, \bar{\mathbf{w}}_k^r, \mathcal{S}_2)\|^2$$

$$+ \frac{1}{4\gamma} \|\mathbf{u}_{i,k}^r(\mathbf{z}_{i,k,1}^r) - \mathbf{u}_{i,k-1}^r(\mathbf{z}_{i,k,1}^r)\|^2$$

$$\leq \gamma \|g(\mathbf{w}_{i,k}^r, \mathbf{z}_{i,k,1}^r, \mathbf{w}_{j,t}^{r-1}, \mathbf{z}_{j,t,2}^{r-1}) - g(\bar{\mathbf{w}}_k^r, \mathbf{z}_{i,k,1}^r, \bar{\mathbf{w}}_k^r, \mathcal{S}_2)\|^2$$

$$\leq 4\gamma \|g(\bar{\mathbf{w}}^{r-1}, \mathbf{z}_{i,k,1}^r, \bar{\mathbf{w}}^{r-1}, \mathbf{z}_{j,t,2}^{r-1}) - g(\bar{\mathbf{w}}^{r-1}, \mathbf{z}_{i,k,1}^r, \bar{\mathbf{w}}^{r-1}, \mathcal{S}_2)\|^2 + 4\gamma \tilde{L} \|\bar{\mathbf{w}}^r - \bar{\mathbf{w}}^{r-1}\|^2$$

$$+ 4\gamma \tilde{L} \|\mathbf{w}_{i,k}^r - \bar{\mathbf{w}}^r\|^2 + 4\gamma \tilde{L} \|\mathbf{w}_{i,k}^{r-1} - \bar{\mathbf{w}}^{r-1}\|^2$$

$$\leq 4\gamma \sigma^2 + 4\gamma \tilde{L} \|\bar{\mathbf{w}}^r - \bar{\mathbf{w}}^{r-1}\|^2 + 4\gamma \tilde{L} \|\mathbf{w}_{i,k}^r - \bar{\mathbf{w}}^r\|^2 + 4\gamma \tilde{L} \|\mathbf{w}_{i,k}^{r-1} - \bar{\mathbf{w}}^{r-1}\|^2$$

$$\tag{40}$$

Then, we have

$$\frac{1}{2N} \sum_{i=1}^N \frac{1}{|\mathcal{S}_1^i|} \sum_{\mathbf{z} \in |\mathcal{S}_1^i|} \mathbb{E} \|\mathbf{u}_{i,k}^r(\mathbf{z}) - g(\bar{\mathbf{w}}_k^r, \mathbf{z}, \bar{\mathbf{w}}_k^r, \mathcal{S}_2)\|^2 \leq \frac{1}{2N} \sum_{i=1}^N \frac{1}{|\mathcal{S}_1^i|} \sum_{\mathbf{z} \in |\mathcal{S}_1^i|} \mathbb{E} \|\mathbf{u}_{i,k-1}^r(\mathbf{z}) - g(\bar{\mathbf{w}}_k^r, \mathbf{z}, \bar{\mathbf{w}}_k^r, \mathcal{S}_2)\|^2$$

$$+ \frac{1}{N} \sum_i \frac{1}{|\mathcal{S}_1^i|} \left[ \frac{1}{2\gamma} \|\mathbf{u}_{i,k-1}^r(\mathbf{z}_{i,k,1}^r) - g(\mathbf{w}_k^r, \mathbf{z}_{i,k,1}^r, \mathbf{w}_k^r, \mathcal{S}_2)\|^2 - \frac{1}{2\gamma} \|\mathbf{u}_{i,k}^r(\mathbf{z}_{i,k,1}^r) - g(\mathbf{w}_k^r, \mathbf{z}_{i,k,1}^r, \mathbf{w}_k^r, \mathcal{S}_2)\|^2 \right.$$

$$- \frac{\gamma+1}{8\gamma} \|\mathbf{u}_{i,k}^r(\mathbf{z}_{i,k,1}^r) - \mathbf{u}_{i,k-1}^r(\mathbf{z}_{i,k,1}^r)\|^2 + \gamma \|g(\bar{\mathbf{w}}^{r-1}, \mathbf{z}_{i,k,1}^r, \bar{\mathbf{w}}^{r-1}, \mathbf{z}_{j,t,2}^{r-1}) - g(\bar{\mathbf{w}}^{r-1}, \mathbf{z}_{i,k,1}^r, \bar{\mathbf{w}}^{r-1}, \mathcal{S}_2)\|^2$$

$$+ 4\gamma \tilde{L} \|\bar{\mathbf{w}}^r - \bar{\mathbf{w}}^{r-1}\|^2 + 4\gamma \tilde{L} \|\mathbf{w}_{i,k}^r - \bar{\mathbf{w}}^r\|^2 + 4\gamma \tilde{L}^2 \|\mathbf{w}_{i,k}^{r-1} - \bar{\mathbf{w}}^{r-1}\|^2$$

$$\left. + \langle \mathbf{u}_{i,k-1}^r(\mathbf{z}_{i,k,1}^r) - g(\mathbf{w}_{i,k}^r, \mathbf{z}_{i,k,1}^r, \mathbf{w}_{j,t}^{r-1}, \mathbf{z}_{j,t,2}^{r-1}), g(\bar{\mathbf{w}}_k^r, \mathbf{z}_{i,k,1}^r, \bar{\mathbf{w}}_k^r, \mathcal{S}_2) - \mathbf{u}_{i,k-1}^r(\mathbf{z}_{i,k,1}^r) \rangle \right].$$

$$\tag{41}$$

Note that $\sum_{\mathbf{z} \neq \mathbf{z}_{i,k,1}^r} \|\mathbf{u}_{i,k}^r(\mathbf{z}) - g(\bar{\mathbf{w}}_{k+1}^r, \mathbf{z}, \bar{\mathbf{w}}_{k+1}^r, \mathcal{S}_2)\|^2 = \sum_{\mathbf{z} \neq \mathbf{z}_{i,k,1}^r} \|\mathbf{u}_{i,k+1}^r(\mathbf{z}) - g(\mathbf{w}_{k+1}^r, \mathbf{z}, \bar{\mathbf{w}}_{k+1}^r, \mathcal{S}_2)\|^2$, which implies

$$\frac{1}{2\gamma} \left( \|\mathbf{u}_{i,k-1}^r(\mathbf{z}_{i,k,1}^r) - g(\bar{\mathbf{w}}_k^r, \mathbf{z}, \bar{\mathbf{w}}_k^r, \mathcal{S}_2)\|^2 - \|\mathbf{u}_{i,k}^r(\mathbf{z}_{i,k,1}^r) - g(\bar{\mathbf{w}}_k^r, \mathbf{z}, \bar{\mathbf{w}}_k^r, \mathcal{S}_2)\|^2 \right)$$

$$= \frac{1}{2\gamma} \sum_{\mathbf{z} \in \mathcal{S}_1^i} \left( \|\mathbf{u}_{i,k-1}^r(\mathbf{z}) - g(\bar{\mathbf{w}}_k^r, \mathbf{z}, \bar{\mathbf{w}}_k^r, \mathcal{S}_2)\|^2 - \|\mathbf{u}_{i,k}^r(\mathbf{z}) - g(\bar{\mathbf{w}}_k^r, \mathbf{z}, \bar{\mathbf{w}}_k^r, \mathcal{S}_2)\|^2 \right).$$

$$\tag{42}$$

Besides, we have

$$\mathbb{E}\langle \mathbf{u}_{i,k-1}^r(\mathbf{z}_{i,k,1}^r) - g(\mathbf{w}_{i,k}^r, \mathbf{z}_{i,k,1}^r, \mathbf{w}_{j,t}^{r-1}, \mathbf{z}_{j,t,2}^{r-1}), g(\bar{\mathbf{w}}_k^r, \mathbf{z}_{i,k,1}^r, \bar{\mathbf{w}}_k^r, \mathcal{S}_2) - \mathbf{u}_{i,k-1}^r(\mathbf{z}_{i,k,1}^r)\rangle$$

$$= \mathbb{E}\langle \mathbf{u}_{i,k-1}^r(\mathbf{z}_{i,k,1}^r) - g(\bar{\mathbf{w}}^{r-1}, \mathbf{z}_{i,k,1}^r, \bar{\mathbf{w}}^{r-1}, \mathbf{z}_{j,t,2}^{r-1}), g(\bar{\mathbf{w}}_k^r, \mathbf{z}_{i,k,1}^r, \bar{\mathbf{w}}_k^r, \mathcal{S}_2) - \mathbf{u}_{i,k-1}^r(\mathbf{z}_{i,k,1}^r)\rangle$$

$$+ \mathbb{E}\langle g(\bar{\mathbf{w}}^{r-1}, \mathbf{z}_{i,k,1}^r, \bar{\mathbf{w}}^{r-1}, \mathbf{z}_{j,t,2}^{r-1}) - g(\mathbf{w}_{i,k}^r, \mathbf{z}_{i,k,1}^r, \mathbf{w}_{j,t}^{r-1}, \mathbf{z}_{j,t,2}^{r-1}), g(\bar{\mathbf{w}}_k^r, \mathbf{z}_{i,k,1}^r, \bar{\mathbf{w}}_k^r, \mathcal{S}_2) - \mathbf{u}_{i,k-1}^r(\mathbf{z}_{i,k,1}^r)\rangle$$

$$\leq \mathbb{E}\langle \mathbf{u}_{i,k-1}^r(\mathbf{z}_{i,k,1}^r) - g(\bar{\mathbf{w}}^{r-1}, \mathbf{z}_{i,k,1}^r, \bar{\mathbf{w}}^{r-1}, \mathbf{z}_{j,t,2}^{r-1}), g(\bar{\mathbf{w}}_k^r, \mathbf{z}_{i,k,1}^r, \bar{\mathbf{w}}_k^r, \mathcal{S}_2) - g(\bar{\mathbf{w}}^{r-1}, \mathbf{z}_{i,k,1}^r, \bar{\mathbf{w}}^{r-1}, \mathcal{S}_2)\rangle$$

$$+ \mathbb{E}\langle \mathbf{u}_{i,k-1}^r(\mathbf{z}_{i,k,1}^r) - g(\bar{\mathbf{w}}^{r-1}, \mathbf{z}_{i,k,1}^r, \bar{\mathbf{w}}^{r-1}, \mathbf{z}_{j,t,2}^{r-1}), g(\bar{\mathbf{w}}^{r-1}, \mathbf{z}_{i,k,1}^r, \bar{\mathbf{w}}^{r-1}, \mathcal{S}_2) - \mathbf{u}_{i,k-1}^r(\mathbf{z}_{i,k,1}^r)\rangle$$

$$+ \|\bar{\mathbf{w}}^{r-1} - \mathbf{w}_{i,k}^r\|^2 + \frac{1}{4}\|g(\bar{\mathbf{w}}^r, \mathbf{z}_{i,k,1}^r, \bar{\mathbf{w}}_k^r, \mathcal{S}_2) - \mathbf{u}_{i,k-1}^r(\mathbf{z}_{i,k,1}^r)\|^2$$

$$\leq \gamma C_0^2 + \frac{1}{\gamma}\|\bar{\mathbf{w}}_k^r - \bar{\mathbf{w}}^{r-1}\|^2$$

$$+ \mathbb{E}\langle \mathbf{u}_{i,k-1}^r(\mathbf{z}_{i,k,1}^r) - g(\bar{\mathbf{w}}^{r-1}, \mathbf{z}_{i,k,1}^r, \bar{\mathbf{w}}^{r-1}, \mathbf{z}_{j,t,2}^{r-1}), g(\bar{\mathbf{w}}^{r-1}, \mathbf{z}_{i,k,1}^r, \bar{\mathbf{w}}^{r-1}, \mathcal{S}_2) - \mathbf{u}_{i,k-1}^r(\mathbf{z}_{i,k,1}^r)\rangle$$

$$+ \|\bar{\mathbf{w}}^{r-1} - \mathbf{w}_{i,k}^r\|^2 + \frac{1}{4}\|g(\bar{\mathbf{w}}^r, \mathbf{z}_{i,k,1}^r, \bar{\mathbf{w}}_k^r, \mathcal{S}_2) - \mathbf{u}_{i,k-1}^r(\mathbf{z}_{i,k,1}^r)\|^2, \tag{43}$$

where

$$\mathbb{E}\langle \mathbf{u}_{i,k-1}^r(\mathbf{z}_{i,k,1}^r) - g(\bar{\mathbf{w}}^{r-1}, \mathbf{z}_{i,k,1}^r, \bar{\mathbf{w}}^{r-1}, \mathbf{z}_{j,t,2}^{r-1}), g(\bar{\mathbf{w}}^{r-1}, \mathbf{z}_{i,k,1}^r, \bar{\mathbf{w}}^{r-1}, \mathcal{S}_2) - \mathbf{u}_{i,k-1}^r(\mathbf{z}_{i,k,1}^r)\rangle$$

$$= \mathbb{E}\langle \mathbf{u}_{i,k-1}^r(\mathbf{z}_{i,k,1}^r) - \mathbf{u}_{i,0}^{r-1}(\mathbf{z}_{i,k,1}^r) + \mathbf{u}_{i,0}^{r-1}(\mathbf{z}_{i,k,1}^r) - g(\bar{\mathbf{w}}^{r-1}, \mathbf{z}_{i,k,1}^r, \bar{\mathbf{w}}^{r-1}, \mathbf{z}_{j,t,2}^{r-1}),$$

$$g(\bar{\mathbf{w}}^{r-1}, \mathbf{z}_{i,k,1}^r, \bar{\mathbf{w}}^{r-1}, \mathcal{S}_2) - \mathbf{u}_{i,0}^{r-1}(\mathbf{z}_{i,k,1}^r) + \mathbf{u}_{i,0}^{r-1}(\mathbf{z}_{i,k,1}^r) - \mathbf{u}_{i,k-1}^r(\mathbf{z}_{i,k,1}^r)\rangle$$

$$\leq \mathbb{E}\langle \mathbf{u}_{i,k-1}^r(\mathbf{z}_{i,k,1}^r) - \mathbf{u}_{i,0}^{r-1}(\mathbf{z}_{i,k,1}^r), g(\bar{\mathbf{w}}^{r-1}, \mathbf{z}_{i,k,1}^r, \bar{\mathbf{w}}^{r-1}, \mathcal{S}_2) - \mathbf{u}_{i,0}^{r-1}(\mathbf{z}_{i,k,1}^r)\rangle$$

$$+ \mathbb{E}\langle \mathbf{u}_{i,k-1}^r(\mathbf{z}_{i,k,1}^r) - \mathbf{u}_{i,0}^{r-1}(\mathbf{z}_{i,k,1}^r), \mathbf{u}_{i,0}^{r-1}(\mathbf{z}_{i,k,1}^r) - \mathbf{u}_{i,k-1}^r(\mathbf{z}_{i,k,1}^r)\rangle$$

$$+ \mathbb{E}\langle \mathbf{u}_{i,0}^{r-1}(\mathbf{z}_{i,k,1}^r) - g(\bar{\mathbf{w}}^{r-1}, \mathbf{z}_{i,k,1}^r, \bar{\mathbf{w}}^{r-1}, \mathbf{z}_{j,t,2}^{r-1}), g(\bar{\mathbf{w}}^{r-1}, \mathbf{z}_{i,k,1}^r, \bar{\mathbf{w}}^{r-1}, \mathcal{S}_2) - \mathbf{u}_{i,0}^{r-1}(\mathbf{z}_{i,k,1}^r)\rangle$$

$$+ \mathbb{E}\langle \mathbf{u}_{i,0}^{r-1}(\mathbf{z}_{i,k,1}^r) - g(\bar{\mathbf{w}}^{r-1}, \mathbf{z}_{i,k,1}^r, \bar{\mathbf{w}}^{r-1}, \mathbf{z}_{j,t,2}^{r-1}), \mathbf{u}_{i,0}^{r-1}(\mathbf{z}_{i,k,1}^r) - \mathbf{u}_{i,k-1}^r(\mathbf{z}_{i,k,1}^r)\rangle$$

$$\leq 4\mathbb{E}\|\mathbf{u}_{i,k-1}^r(\mathbf{z}_{i,k,1}^r) - \mathbf{u}_{i,0}^{r-1}(\mathbf{z}_{i,k,1}^r)\|^2 + \frac{1}{4}\mathbb{E}\|g(\bar{\mathbf{w}}^{r-1}, \mathbf{z}_{i,k,1}^r, \bar{\mathbf{w}}^{r-1}, \mathcal{S}_2) - \mathbf{u}_{i,0}^{r-1}(\mathbf{z}_{i,k,1}^r)\|^2$$

$$- \mathbb{E}\|g(\bar{\mathbf{w}}^{r-1}, \mathbf{z}_{i,k,1}^r, \bar{\mathbf{w}}^{r-1}, \mathcal{S}_2) - \mathbf{u}_{i,0}^{r-1}(\mathbf{z}_{i,k,1}^r)\|^2$$

$$+ \frac{1}{4}\mathbb{E}\|g(\bar{\mathbf{w}}^{r-1}, \mathbf{z}_{i,k,1}^r, \bar{\mathbf{w}}^{r-1}, \mathcal{S}_2) - \mathbf{u}_{i,0}^{r-1}(\mathbf{z}_{i,k,1}^r)\|^2 + 4\mathbb{E}\|\mathbf{u}_{i,k-1}^r(\mathbf{z}_{i,k,1}^r) - \mathbf{u}_{i,0}^{r-1}(\mathbf{z}_{i,k,1}^r)\|^2. \tag{44}$$

Noting

$$- \mathbb{E}\|g(\bar{\mathbf{w}}^{r-1}, \mathbf{z}_{i,k,1}^r, \bar{\mathbf{w}}^{r-1}, \mathcal{S}_2) - \mathbf{u}_{i,0}^{r-1}(\mathbf{z}_{i,k,1}^r)\|^2$$

$$= -\mathbb{E}\|g(\bar{\mathbf{w}}^{r-1}, \mathbf{z}_{i,k,1}^r, \bar{\mathbf{w}}^{r-1}, \mathcal{S}_2) - \mathbf{u}_{i,k}^r(\mathbf{z}_{i,k,1}^r) + \mathbf{u}_{i,k}^r(\mathbf{z}_{i,k,1}^r) - \mathbf{u}_{i,0}^{r-1}(\mathbf{z}_{i,k,1}^r)\|^2$$

$$= -\mathbb{E}\|g(\bar{\mathbf{w}}^{r-1}, \mathbf{z}_{i,k,1}^r, \bar{\mathbf{w}}^{r-1}, \mathcal{S}_2) - \mathbf{u}_{i,k}^r(\mathbf{z}_{i,k,1}^r)\|^2 - \mathbb{E}\|\mathbf{u}_{i,k}^r(\mathbf{z}_{i,k,1}^r) - \mathbf{u}_{i,0}^{r-1}(\mathbf{z}_{i,k,1}^r)\|^2$$

$$+ 2\mathbb{E}\langle g(\bar{\mathbf{w}}^{r-1}, \mathbf{z}_{i,k,1}^r, \bar{\mathbf{w}}^{r-1}, \mathcal{S}_2) - \mathbf{u}_{i,k}^r(\mathbf{z}_{i,k,1}^r), \mathbf{u}_{i,k}^r(\mathbf{z}_{i,k,1}^r) - \mathbf{u}_{i,0}^{r-1}(\mathbf{z}_{i,k,1}^r)\rangle \tag{45}$$

$$\leq -\frac{1}{2}\mathbb{E}\|g(\bar{\mathbf{w}}^{r-1}, \mathbf{z}_{i,k,1}^r, \bar{\mathbf{w}}^{r-1}, \mathcal{S}_2) - \mathbf{u}_{i,k}^r(\mathbf{z}_{i,k,1}^r)\|^2 + 8\|\mathbf{u}_{i,k}^r(\mathbf{z}_{i,k,1}^r) - \mathbf{u}_{i,0}^{r-1}(\mathbf{z}_{i,k,1}^r)\|^2$$

$$\leq -\frac{1}{2}\mathbb{E}\|g(\bar{\mathbf{w}}^{r-1}, \mathbf{z}_{i,k,1}^r, \bar{\mathbf{w}}^{r-1}, \mathcal{S}_2) - \mathbf{u}_{i,k}^r(\mathbf{z}_{i,k,1}^r)\|^2 + 8\beta^2 K^2 C_0^2.$$

Then, we can obtain

$$\frac{\gamma+1}{2}\frac{1}{N}\sum_{i=1}^N \frac{1}{|\mathcal{S}_1^i|}\sum_{\mathbf{z}\in|\mathcal{S}_1^i|} \mathbb{E}\|\mathbf{u}_{i,k}^r(\mathbf{z}) - g(\bar{\mathbf{w}}_k^r, \mathbf{z}, \bar{\mathbf{w}}_k^r, \mathcal{S}_2)\|^2$$

$$\leq \frac{\gamma(1-\frac{1}{|\mathcal{S}_1^i|})+1}{2}\frac{1}{N}\sum_{i=1}^N \frac{1}{|\mathcal{S}_1^i|}\sum_{\mathbf{z}\in|\mathcal{S}_1^i|} \mathbb{E}\|\mathbf{u}_{i,k-1}^r(\mathbf{z}) - g(\bar{\mathbf{w}}_k^r, \mathbf{z}, \bar{\mathbf{w}}_k^r, \mathcal{S}_2)\|^2 + \frac{\gamma^2}{|\mathcal{S}_1^i|}(\sigma^2 + C_0^2) \tag{46}$$

$$+ \frac{\gamma\beta^2 K^2 C_0^2}{|\mathcal{S}_1^i|} + \gamma^2\|\bar{\mathbf{w}}^r - \bar{\mathbf{w}}^{r-1}\|^2 + \gamma^2\frac{1}{N}\sum_i \|\bar{\mathbf{w}}^r - \mathbf{w}_{i,k}^r\|^2 + \|\bar{\mathbf{w}}^r - \bar{\mathbf{w}}_k^r\|^2.$$

Dividing $\frac{\gamma+1}{2}$ on both sides gives

$$\frac{1}{N}\sum_{i=1}^{N}\frac{1}{|\mathcal{S}_1^i|}\sum_{\mathbf{z}\in|\mathcal{S}_1^i|}\mathbb{E}\|\mathbf{u}_{i,k}^r(\mathbf{z})-g(\bar{\mathbf{w}}_k^r,\mathbf{z},\bar{\mathbf{w}}_k^r,\mathcal{S}_2)\|^2$$

$$\leq\frac{\gamma(1-\frac{1}{|\mathcal{S}_1^i|})+1}{\gamma+1}\frac{1}{N}\sum_{i=1}^{N}\frac{1}{|\mathcal{S}_1^i|}\sum_{\mathbf{z}\in|\mathcal{S}_1^i|}\mathbb{E}\|\mathbf{u}_{i,k-1}^r(\mathbf{z})-g(\bar{\mathbf{w}}_k^r,\mathbf{z},\bar{\mathbf{w}}_k^r,\mathcal{S}_2)\|^2$$

$$+\frac{\gamma\beta^2 K^2 C_0^2}{|\mathcal{S}_1^i|}+2\frac{\gamma^2}{|\mathcal{S}_1^i|}(\sigma^2+C_0^2)+2\gamma^2\|\bar{\mathbf{w}}^r-\bar{\mathbf{w}}^{r-1}\|^2+2\gamma^2\|\bar{\mathbf{w}}^r-\mathbf{w}_{i,k}^r\|^2+2\|\bar{\mathbf{w}}^r-\bar{\mathbf{w}}_k^r\|^2 \tag{47}$$

Using Young's inequality,

$$\frac{1}{N}\sum_{i=1}^{N}\frac{1}{|\mathcal{S}_1^i|}\sum_{\mathbf{z}\in|\mathcal{S}_1^i|}\mathbb{E}\|\mathbf{u}_{i,k}^r(\mathbf{z})-g(\bar{\mathbf{w}}_k^r,\mathbf{z},\bar{\mathbf{w}}_k^r,\mathcal{S}_2)\|^2$$

$$\leq(1-\frac{\gamma}{2|\mathcal{S}_1^i|})\frac{1}{N}\sum_{i=1}^{N}\frac{1}{|\mathcal{S}_1^i|}\sum_{\mathbf{z}\in|\mathcal{S}_1^i|}[(1+\frac{\gamma}{4|\mathcal{S}_1^i|})\mathbb{E}\|\mathbf{u}_{i,k-1}^r(\mathbf{z})-g(\bar{\mathbf{w}}_{k-1}^r,\mathbf{z},\bar{\mathbf{w}}_{k-1}^r,\mathcal{S}_2)\|^2$$

$$+(1+\frac{4|\mathcal{S}_1^i|}{\gamma})\tilde{L}^2\|\bar{\mathbf{w}}_{k-1}^r-\bar{\mathbf{w}}_k^r\|^2]+2\gamma^2\|\bar{\mathbf{w}}^r-\bar{\mathbf{w}}^{r-1}\|^2+2\gamma^2\|\bar{\mathbf{w}}^r-\mathbf{w}_{i,k}^r\|^2+2\|\bar{\mathbf{w}}^r-\bar{\mathbf{w}}_k^r\|^2$$

$$\leq(1-\frac{\gamma}{4|\mathcal{S}_1^i|})\frac{1}{N}\sum_{i=1}^{N}\frac{1}{|\mathcal{S}_1^i|}\sum_{\mathbf{z}\in|\mathcal{S}_1^i|}[\mathbb{E}\|\mathbf{u}_{i,k-1}^r(\mathbf{z})-g(\bar{\mathbf{w}}_{k-1}^r,\mathbf{z},\bar{\mathbf{w}}_{k-1}^r,\mathcal{S}_2)\|^2$$

$$+\frac{\gamma\beta^2 K^2 C_0}{|\mathcal{S}_1^i|}+2\frac{\gamma^2}{|\mathcal{S}_1^i|}(\sigma^2+C_0^2)$$

$$+(1+\frac{4|\mathcal{S}_1^i|}{\gamma})\tilde{L}^2\|\bar{\mathbf{w}}_{k-1}^r-\bar{\mathbf{w}}_k^r\|^2]+2\gamma^2\|\bar{\mathbf{w}}^r-\bar{\mathbf{w}}^{r-1}\|^2+2\gamma^2\|\bar{\mathbf{w}}^r-\mathbf{w}_{i,k}^r\|^2+2\|\bar{\mathbf{w}}^r-\bar{\mathbf{w}}_k^r\|^2. \tag{48}$$

$\square$

## D.2 ANALYSIS OF THE ESTIMATOR OF GRADIENT

With update $G_{i,k}^r = (1-\beta)G_{i,k-1}^r + \beta(G_{i,k,1}^r + G_{i,k,2}^r)$. we define $\bar{G}_k^r := \frac{1}{N}\sum_{i=1}^{N}G_{i,k}^r$, and $\Delta_k^r := \|\bar{G}_k^r-\nabla F(\bar{\mathbf{w}}_k^r)\|^2$. Then it follows that $\bar{G}_k^r = (1-\beta)\bar{G}_{k-1}^r + \beta\frac{1}{N}\sum_i(G_{i,k,1}^r + G_{i,k,2}^r)$.

**Lemma 2.** *Under Assumption 2, Algorithm 3 ensures that*

$$\Delta_k^r \leq (1-\beta)\|\bar{G}_{k-1}^r-\nabla F(\bar{\mathbf{w}}_{k-1}^r)\|^2 + 2\frac{\beta^2\sigma^2}{N} + 5\beta\|\bar{\mathbf{w}}^r-\mathbf{w}_{i,k}^r\|^2$$

$$+5\beta\|\bar{\mathbf{w}}^{r-1}-\bar{\mathbf{w}}^r\|^2 + 5\beta\|\mathbf{u}_{i,k}^r(\mathbf{z}_{i,k,1}^r)-g_{\mathbf{z}_{i,k,1}^r}(\bar{\mathbf{w}}^r))\|^2. \tag{49}$$

*Proof.*

$$\Delta_k^r = \|\bar{G}_k^r - \nabla F(\bar{\mathbf{w}}_k^r)\|^2$$

$$= \left\| (1-\beta)\bar{G}_{k-1}^r + \beta\frac{1}{N}\sum_i (G_{i,k,1}^r + G_{i,k,2}^r) - \nabla F(\bar{\mathbf{w}}_k^r) \right\|^2$$

$$= \left\| (1-\beta)(\bar{G}_{k-1}^r - \nabla F(\bar{\mathbf{w}}_{k-1}^r)) + (1-\beta)(\nabla F(\bar{\mathbf{w}}_{k-1}^r) - \nabla F(\bar{\mathbf{w}}_k^r)) \right.$$

$$+ \beta\left( \frac{1}{N}\sum_i (G_1(\mathbf{w}_{i,k}^r, \mathbf{z}_{i,k,1}^r, \mathbf{u}_{i,k}^r(\mathbf{z}_{i,k,1}^r), \mathbf{w}_{j,t}^{r-1}, \mathbf{z}_{j,t,2}^{r-1}) + G_2(\mathbf{w}_{j',t'}^{r-1}, \mathbf{z}_{j',t',1}^{r-1}, \mathbf{u}_{j',t'}^{r-1}(\mathbf{z}_{j',k',1}^{r-1}), \mathbf{w}_{i,k}^r, \mathbf{z}_{i,k,2}^r)) \right.$$

$$\left. - \frac{1}{N}\sum_i (G_1(\bar{\mathbf{w}}^{r-1}, \mathbf{z}_{i,k,1}^r, \mathbf{u}_{i,k}^r(\mathbf{z}_{i,k,1}^r), \mathbf{w}_{j,t}^{r-1}, \mathbf{z}_{j,t,2}^{r-1}) + G_2(\bar{\mathbf{w}}^{r-1}, \mathbf{z}_{j',t',1}^{r-1}, \mathbf{u}_{j',t'}^{r-1}(\mathbf{z}_{j',t',1}^{r-1}), \mathbf{w}_{i,k}^r, \mathbf{z}_{i,k,2}^r)) \right)$$

$$+ \beta\left( \frac{1}{N}\sum_i (G_1(\bar{\mathbf{w}}^{r-1}, \mathbf{z}_{i,k,1}^r, \mathbf{u}_{i,k}^r(\mathbf{z}_{i,k,1}^r), \bar{\mathbf{w}}^{r-1}, \mathbf{z}_{j,t,2}^{r-1}) + G_2(\bar{\mathbf{w}}^{r-1}, \mathbf{z}_{j',t',1}^{r-1}, \mathbf{u}_{j',t'}^{r-1}(\mathbf{z}_{j',t',1}^{r-1}), \bar{\mathbf{w}}^{r-1}, \mathbf{z}_{i,k,2}^r)) \right)$$

$$- \frac{1}{N}\sum_i (G_1(\bar{\mathbf{w}}^{r-1}, \mathbf{z}_{i,k,1}^r, g(\bar{\mathbf{w}}^r, \mathbf{z}_{i,k,1}^r, \bar{\mathbf{w}}^r, \mathcal{S}_2), \bar{\mathbf{w}}^{r-1}, \mathbf{z}_{j,t,2}^{r-1})$$

$$\left. + G_2(\bar{\mathbf{w}}^{r-1}, \mathbf{z}_{j',t',1}^{r-1}, g(\bar{\mathbf{w}}^{r-1}, \mathbf{z}_{j',t',1}^{r-1}, \bar{\mathbf{w}}^{r-1}, \mathcal{S}_2), \bar{\mathbf{w}}^{r-1}, \mathbf{z}_{i,k,2}^r)) \right)$$

$$+ \beta\left( \frac{1}{N}\sum_i (G_1(\bar{\mathbf{w}}^{r-1}, \mathbf{z}_{i,k,1}^r, g(\bar{\mathbf{w}}^r, \mathbf{z}_{i,k,1}^r, \bar{\mathbf{w}}^r, \mathcal{S}_2), \bar{\mathbf{w}}^{r-1}, \mathbf{z}_{j,t,2}^{r-1}) \right.$$

$$+ G_2(\bar{\mathbf{w}}^{r-1}, \mathbf{z}_{j',t',1}^{r-1}, g(\bar{\mathbf{w}}^{r-1}, \mathbf{z}_{j',t',1}^{r-1}, \bar{\mathbf{w}}^{r-1}, \mathcal{S}_2), \bar{\mathbf{w}}^{r-1}, \mathbf{z}_{i,k,2}^r))$$

$$- \frac{1}{N}\sum_i (G_1(\bar{\mathbf{w}}^{r-1}, \mathbf{z}_{i,k,1}^r, g(\bar{\mathbf{w}}^{r-1}, \mathbf{z}_{i,k,1}^r, \bar{\mathbf{w}}^{r-1}, \mathcal{S}_2), \bar{\mathbf{w}}^{r-1}, \mathbf{z}_{j,t,2}^{r-1})$$

$$\left. + G_2(\bar{\mathbf{w}}^{r-1}, \mathbf{z}_{j',t',1}^{r-1}, g(\bar{\mathbf{w}}^{r-1}, \mathbf{z}_{j',t',1}^{r-1}, \bar{\mathbf{w}}^{r-1}, \mathcal{S}_2), \bar{\mathbf{w}}^{r-1}, \mathbf{z}_{i,k,2}^r)) \right)$$

$$+ \beta\left( \frac{1}{N}\sum_i (G_1(\bar{\mathbf{w}}^{r-1}, \mathbf{z}_{i,k,1}^r, g(\bar{\mathbf{w}}^{r-1}, \mathbf{z}_{i,k,1}^r, \bar{\mathbf{w}}^{r-1}, \mathcal{S}_2), \bar{\mathbf{w}}^{r-1}, \mathbf{z}_{j,t,2}^{r-1}) \right.$$

$$\left. + G_2(\bar{\mathbf{w}}^{r-1}, \mathbf{z}_{j',t',1}^{r-1}, g(\bar{\mathbf{w}}^{r-1}, \mathbf{z}_{j',t',1}^{r-1}, \bar{\mathbf{w}}^{r-1}, \mathcal{S}_2), \bar{\mathbf{w}}^{r-1}, \mathbf{z}_{i,k,2}^r)) - \nabla F(\bar{\mathbf{w}}_k^r) \right) \right\|^2$$

$$(50)$$

Denoting $g_{\mathbf{z}}(\mathbf{w}) = g(\mathbf{w}, \mathbf{z}, \mathbf{w}, \mathcal{S}_2)$. Using Young's inequality, we can then derive

$$
\begin{aligned}
\Delta_k^r \leq {} & (1+\beta) \Big\| (1-\beta)(\bar{G}_{k-1}^r - \nabla F(\bar{\mathbf{w}}_{k-1}^r)) \\
& + \beta \Big( \frac{1}{N} \sum_i (G_1(\bar{\mathbf{w}}^{r-1}, \mathbf{z}_{i,k,1}^r, g(\bar{\mathbf{w}}^{r-1}, \mathbf{z}_{i,k,1}^r, \bar{\mathbf{w}}^{r-1}, \mathcal{S}_2), \bar{\mathbf{w}}^{r-1}, \mathbf{z}_{j,t,2}^{r-1}) \\
& + G_2(\bar{\mathbf{w}}^{r-1}, \mathbf{z}_{j',t',1}^{r-1}, g(\bar{\mathbf{w}}^{r-1}, \mathbf{z}_{j',t',1}^{r-1}, \bar{\mathbf{w}}^{r-1}, \mathcal{S}_2), \bar{\mathbf{w}}^{r-1}, \mathbf{z}_{i,k,2}^r)) - \nabla F(\bar{\mathbf{w}}^{r-1}) \Big\|^2 \\
& + \Big(1 + \frac{1}{\beta}\Big) \Big\| \beta \Big( \frac{1}{N} \sum_i (G_1(\mathbf{w}_{i,k}^r, \mathbf{z}_{i,k,1}^r, \mathbf{u}_{i,k}^r(\mathbf{z}_{i,k,1}^r), \mathbf{w}_{j,t}^{r-1}, \mathbf{z}_{j,t,2}^{r-1}) + G_2(\mathbf{w}_{j',t'}^{r-1}, \mathbf{z}_{j',t',1}^{r-1}, \mathbf{u}_{j',t'}^{r-1}(\mathbf{z}_{j',k',1}^{r-1}), \mathbf{w}_{i,k}^r, \mathbf{z}_{i,k,2}^r)) \\
& - \frac{1}{N} \sum_i (G_1(\bar{\mathbf{w}}^{r-1}, \mathbf{z}_{i,k,1}^r, \mathbf{u}_{i,k}^r(\mathbf{z}_{i,k,1}^r), \mathbf{w}_{j,t}^{r-1}, \mathbf{z}_{j,t,2}^{r-1}) + G_2(\bar{\mathbf{w}}^{r-1}, \mathbf{z}_{j',t',1}^{r-1}, \mathbf{u}_{j',t'}^{r-1}(\mathbf{z}_{j',t',1}^{r-1}), \mathbf{w}_{i,k}^r, \mathbf{z}_{i,k,2}^r)) \Big) \\
& + \beta \Big( \frac{1}{N} \sum_i (G_1(\bar{\mathbf{w}}^{r-1}, \mathbf{z}_{i,k,1}^r, \mathbf{u}_{i,k}^r(\mathbf{z}_{i,k,1}^r), \bar{\mathbf{w}}^{r-1}, \mathbf{z}_{j,t,2}^{r-1}) + G_2(\bar{\mathbf{w}}^{r-1}, \mathbf{z}_{j',t',1}^{r-1}, \mathbf{u}_{j',t'}^{r-1}(\mathbf{z}_{j',t',1}^{r-1}), \bar{\mathbf{w}}^{r-1}, \mathbf{z}_{i,k,2}^r)) \Big) \\
& - \frac{1}{N} \sum_i (G_1(\bar{\mathbf{w}}^{r-1}, \mathbf{z}_{i,k,1}^r, g(\bar{\mathbf{w}}^r, \mathbf{z}_{i,k,1}^r, \bar{\mathbf{w}}^r, \mathcal{S}_2), \bar{\mathbf{w}}^{r-1}, \mathbf{z}_{j,t,2}^{r-1}) \\
& + G_2(\bar{\mathbf{w}}^{r-1}, \mathbf{z}_{j',t',1}^{r-1}, g(\bar{\mathbf{w}}^{r-1}, \mathbf{z}_{j',t',1}^{r-1}, \bar{\mathbf{w}}^{r-1}, \mathcal{S}_2), \bar{\mathbf{w}}^{r-1}, \mathbf{z}_{i,k,2}^r)) \Big) \\
& + \beta \Big( \frac{1}{N} \sum_i (G_1(\bar{\mathbf{w}}^{r-1}, \mathbf{z}_{i,k,1}^r, g(\bar{\mathbf{w}}^r, \mathbf{z}_{i,k,1}^r, \bar{\mathbf{w}}^r, \mathcal{S}_2), \bar{\mathbf{w}}^{r-1}, \mathbf{z}_{j,t,2}^{r-1}) \\
& + G_2(\bar{\mathbf{w}}^{r-1}, \mathbf{z}_{j',t',1}^{r-1}, g(\bar{\mathbf{w}}^{r-1}, \mathbf{z}_{j',t',1}^{r-1}, \bar{\mathbf{w}}^{r-1}, \mathcal{S}_2), \bar{\mathbf{w}}^{r-1}, \mathbf{z}_{i,k,2}^r)) \\
& - \frac{1}{N} \sum_i (G_1(\bar{\mathbf{w}}^{r-1}, \mathbf{z}_{i,k,1}^r, g(\bar{\mathbf{w}}^r, \mathbf{z}_{i,k,1}^r, \bar{\mathbf{w}}^r, \mathcal{S}_2), \bar{\mathbf{w}}^{r-1}, \mathbf{z}_{j,t,2}^{r-1}) \\
& + G_2(\bar{\mathbf{w}}^{r-1}, \mathbf{z}_{j',t',1}^{r-1}, g(\bar{\mathbf{w}}^{r-1}, \mathbf{z}_{j',t',1}^{r-1}, \bar{\mathbf{w}}^{r-1}, \mathcal{S}_2), \bar{\mathbf{w}}^{r-1}, \mathbf{z}_{i,k,2}^r)) \Big\|^2 .
\end{aligned}
\tag{51}
$$

By the fact that

$$
\begin{aligned}
\mathbb{E}[\frac{1}{N} \sum_i (G_1(\bar{\mathbf{w}}^{r-1}, \mathbf{z}_{i,k,1}^r, g(\bar{\mathbf{w}}^{r-1}, \mathbf{z}_{i,k,1}^r, \bar{\mathbf{w}}^{r-1}, \mathcal{S}_2), \bar{\mathbf{w}}^{r-1}, \mathbf{z}_{j,t,2}^{r-1}) \\
+ G_2(\bar{\mathbf{w}}^{r-1}, \mathbf{z}_{j',t',1}^{r-1}, g(\bar{\mathbf{w}}^{r-1}, \mathbf{z}_{j',t',1}^{r-1}, \bar{\mathbf{w}}^{r-1}, \mathcal{S}_2), \bar{\mathbf{w}}^{r-1}, \mathbf{z}_{i,k,2}^r)) - \nabla F(\bar{\mathbf{w}}^{r-1})] = 0,
\end{aligned}
\tag{52}
$$

and

$$
\begin{aligned}
\mathbb{E}\|\frac{1}{N} \sum_i (G_1(\bar{\mathbf{w}}^{r-1}, \mathbf{z}_{i,k,1}^r, g(\bar{\mathbf{w}}^{r-1}, \mathbf{z}_{i,k,1}^r, \bar{\mathbf{w}}^{r-1}, \mathcal{S}_2), \bar{\mathbf{w}}^{r-1}, \mathbf{z}_{j,t,2}^{r-1}) \\
+ G_2(\bar{\mathbf{w}}^{r-1}, \mathbf{z}_{j',t',1}^{r-1}, g(\bar{\mathbf{w}}^{r-1}, \mathbf{z}_{j',t',1}^{r-1}, \bar{\mathbf{w}}^{r-1}, \mathcal{S}_2), \bar{\mathbf{w}}^{r-1}, \mathbf{z}_{i,k,2}^r)) - \nabla F(\bar{\mathbf{w}}^{r-1})\|^2 \leq \frac{\sigma^2}{N}
\end{aligned}
\tag{53}
$$

we obtain

$$
\begin{aligned}
\Delta_k^r \leq {} & (1+\beta)(1-\beta)^2 \|\bar{G}_{k-1}^r - \nabla F(\bar{\mathbf{w}}_{k-1}^r)\|^2 + 2\beta^2 \frac{\sigma^2 + C_\ell^2 C_g^2}{N} \\
& + 5\beta \|\bar{\mathbf{w}}^r - \mathbf{w}_{i,k}^r\|^2 + 5\beta \|\bar{\mathbf{w}}^{r-1} - \bar{\mathbf{w}}^r\|^2 + 5\beta \|\mathbf{u}_{i,k}^r(\mathbf{z}_{i,k,1}^r) - g(\bar{\mathbf{w}}^r, \mathbf{z}_{i,k,1}^r, \bar{\mathbf{w}}^r, \mathcal{S}_2))\|^2 \\
\leq {} & (1-\beta) \|\bar{G}_{k-1}^r - \nabla F(\bar{\mathbf{w}}_{k-1}^r)\|^2 + 2\frac{\beta^2 \sigma^2}{N} + 5\beta \|\bar{\mathbf{w}}^r - \mathbf{w}_{i,k}^r\|^2 + 5\beta \|\bar{\mathbf{w}}^{r-1} - \bar{\mathbf{w}}^r\|^2 \\
& + 5\beta \frac{1}{N} \sum_i \|\mathbf{u}_{i,k}^r(\mathbf{z}_{i,k,1}^r) - g(\bar{\mathbf{w}}^r, \mathbf{z}_{i,k,1}^r, \bar{\mathbf{w}}^r, \mathcal{S}_2))\|^2 \\
& + 5\beta \frac{1}{N} \sum_i \|\mathbf{u}_{j',t'}^{r-1}(\mathbf{z}_{j',t',1}^{r-1}) - g(\bar{\mathbf{w}}^{r-1}, \mathbf{z}_{j',t',1}^{r-1}, \bar{\mathbf{w}}^{r-1}, \mathcal{S}_2))\|^2 .
\end{aligned}
\tag{54}
$$

$\square$

### D.3 THEOREM 2

We re-present Theorem 2 as below.

**Theorem 4.** *Suppose Assumption 2 holds, denoting $M = \max_i |\mathcal{S}_i^1|$ as the largest number of data on a single machine, by setting $\gamma = O(\frac{M^{1/3}}{R^{2/3}})$, $\beta = O(\frac{1}{M^{1/6}R^{2/3}})$, $\eta = O(\frac{1}{M^{2/3}R^{2/3}})$ and $K = O(M^{1/3}R^{1/3})$, Algorithm 2 ensures that $\mathbb{E}\left[\frac{1}{R}\sum_{r=1}^R \|\nabla F(\bar{\mathbf{w}}^r)\|^2\right] \leq O(\frac{1}{R^{2/3}})$.*

*Proof.* By updating rules,
$$\|\bar{\mathbf{w}}^r - \mathbf{w}_{i,k}^r\|^2 \leq \eta^2 K^2 C_\ell^2 C_g^2, \tag{55}$$
and
$$\|\bar{\mathbf{w}}_k^r - \bar{\mathbf{w}}^r\|^2 = \tilde{\eta}^2 \|\frac{1}{NK}\sum_i \sum_{m=1}^k \bar{G}_k^r\|^2 \leq \tilde{\eta}^2 \frac{1}{K}\sum_k \|\bar{G}_k^r - \nabla F(\bar{\mathbf{w}}_k^r) + \nabla F(\bar{\mathbf{w}}_k^r)\|^2. \tag{56}$$

By updating rule, we also have
$$\|\bar{\mathbf{w}}^{r-1} - \bar{\mathbf{w}}^r\|^2 = \tilde{\eta}^2 \|\frac{1}{NK}\sum_i \sum_k \bar{G}_k^{r-1}\|^2 \leq \tilde{\eta}^2 \frac{1}{K}\sum_k \|\bar{G}_k^{r-1} - \nabla F(\bar{\mathbf{w}}_k^{r-1}) + \nabla F(\bar{\mathbf{w}}_k^{r-1})\|^2 \tag{57}$$

Lemma 2 gives that
$$\frac{1}{RK}\sum_{r,k} \mathbb{E}\|\bar{G}_k^r - \nabla F(\bar{\mathbf{w}}_k^r)\|^2 \leq \frac{\Delta_0^0}{\beta RK} + \frac{2\beta\sigma^2}{N} + 5\beta\frac{1}{RK}\sum_{r,k}\|\bar{\mathbf{w}}^r - \mathbf{w}_{i,k}^r\|^2 + 5\frac{1}{R}\sum_r \|\bar{\mathbf{w}}^{r-1} - \bar{\mathbf{w}}^r\|^2$$
$$+ 5\frac{1}{R}\sum_r \frac{1}{NK}\sum_{i,k}\frac{1}{|\mathcal{S}_1^i|}\sum_{\mathbf{z}\in\mathcal{S}_1^i}\mathbb{E}\|\mathbf{u}_{i,k}^r(\mathbf{z}) - g(\bar{\mathbf{w}}^r;\mathbf{z},\mathcal{S}_2)\|^2$$
$$+ 5\frac{1}{R}\sum_r \frac{1}{NK}\sum_{j',t'}\frac{1}{|\mathcal{S}_1^i|}\sum_{\mathbf{z}\in\mathcal{S}_1^i}\|\mathbf{u}_{j',t'}^{r-1}(\mathbf{z}_{j',t',1}^{r-1}) - g(\bar{\mathbf{w}}^{r-1};\mathbf{z}_{j',t',1}^{r-1},\mathcal{S}_2))\|^2 \tag{58}$$

which by setting of $\eta$ and $\beta$ leads to
$$\frac{1}{RK}\sum_{r,k} \mathbb{E}\|\bar{G}_k^r - \nabla F(\bar{\mathbf{w}}_k^r)\|^2 \leq \frac{2\Delta_0^0}{\beta RK} + \frac{4\beta\sigma^2}{N} + 10\beta\tilde{\eta}^2 C_\ell^2 C_g^2 + 2\tilde{\eta}^2 \frac{1}{R}\sum_r \|\nabla F(\bar{\mathbf{w}}^{r-1})\|^2$$
$$+ 5\frac{1}{R}\sum_r \frac{1}{NK}\sum_{i,k}\frac{1}{|\mathcal{S}_1^i|}\sum_{\mathbf{z}\in\mathcal{S}_1^i}\mathbb{E}\|\mathbf{u}_{i,k}^r(\mathbf{z}) - g(\bar{\mathbf{w}}^r;\mathbf{z},\mathcal{S}_2)\|^2$$
$$+ 5\frac{1}{R}\sum_r \frac{1}{NK}\sum_{j',t'}\frac{1}{|\mathcal{S}_1^i|}\sum_{\mathbf{z}\in\mathcal{S}_1^i}\|\mathbf{u}_{j',t'}^{r-1}(\mathbf{z}_{j',t',1}^{r-1}) - g(\bar{\mathbf{w}}^{r-1};\mathbf{z}_{j',t',1}^{r-1},\mathcal{S}_2))\|^2, \tag{59}$$

Using Lemma 1 yields
$$\frac{1}{R}\sum_r \frac{1}{NK}\sum_{i=1}^N\sum_{k=1}^K\frac{1}{|\mathcal{S}_1^i|}\sum_{\mathbf{z}\in\mathcal{S}_1^i}\mathbb{E}\|\mathbf{u}_{i,k}^r(\mathbf{z}) - g(\bar{\mathbf{w}}_k^r,\mathbf{z},\bar{\mathbf{w}}_k^r,\mathcal{S}_2)\|^2$$
$$\leq \frac{4M}{\gamma}\frac{1}{R}\sum_r \frac{1}{NK}\sum_{i=1}^N\frac{1}{|\mathcal{S}_1^i|}\sum_{\mathbf{z}\in\mathcal{S}_1^i}\mathbb{E}\|\mathbf{u}_{i,0}^0(\mathbf{z}) - g(\bar{\mathbf{w}}_0^0,\mathbf{z},\bar{\mathbf{w}}_0^0,\mathcal{S}_2)\|^2 + \frac{18M^2}{\gamma^2}\frac{1}{RK}\sum_{r,k}\tilde{L}^2\|\bar{\mathbf{w}}_{k-1}^r - \bar{\mathbf{w}}_k^r\|^2$$
$$+ 4\gamma\beta^2 K^2 C_0^2 + 8\gamma(\sigma^2 + C_0^2)$$
$$+ 8\gamma M\frac{1}{R}\sum_r \|\bar{\mathbf{w}}^r - \bar{\mathbf{w}}^{r-1}\|^2 + 8\gamma|\mathcal{S}_1^i|\frac{1}{RNK}\sum_{r,i,k}\|\bar{\mathbf{w}}^r - \mathbf{w}_{i,k}^r\|^2 + 8\frac{|\mathcal{S}_1^i|}{\gamma}\frac{1}{RK}\sum_{r,k}\cdot\|\bar{\mathbf{w}}^r - \bar{\mathbf{w}}_k^r\|^2. \tag{60}$$

Combining this with previous five inequalities, we obtain

$$\frac{1}{R}\sum_r \frac{1}{NK}\sum_{i=1}^N \sum_{k=1}^K \frac{1}{M}\sum_{\mathbf{z}\in\mathcal{S}_1^i}\mathbb{E}\|\mathbf{u}_{i,k}^r(\mathbf{z}) - g(\bar{\mathbf{w}}_k^r,\mathbf{z},\bar{\mathbf{w}}_k^r,\mathcal{S}_2)\|^2$$

$$\leq O\left(\frac{M}{\gamma RK} + \gamma\beta^2 K^2 + \gamma + \eta^2\frac{M^2}{\gamma^2} + 8\gamma M\tilde{\eta}^2 + \frac{M}{\gamma}\tilde{\eta}^2(\frac{1}{\beta RK} + \frac{\beta}{N}) + \frac{1}{R}\sum_r \tilde{\eta}^2\|\nabla F(\bar{\mathbf{w}}^{r-1})\|^2\right)$$

(61)

and

$$\frac{1}{RK}\sum_{r,k}\mathbb{E}\|\bar{G}_k^r - \nabla F(\bar{\mathbf{w}}_k^r)\|^2$$

$$\leq O\left(\frac{M}{\gamma RK} + \gamma\beta^2 K^2 + \gamma + \eta^2\frac{M^2}{\gamma^2} + 8\gamma|\mathcal{S}_1^i|\tilde{\eta}^2 + \frac{M}{\gamma}\tilde{\eta}^2(\frac{1}{\beta RK} + \frac{\beta}{N}) + \frac{1}{R}\sum_r \tilde{\eta}^2\|\nabla F(\bar{\mathbf{w}}^{r-1})\|^2\right).$$

(62)

Then using the standard analysis of smooth function, we derive

$$F(\bar{\mathbf{w}}^{r+1}) - F(\bar{\mathbf{w}}^r) \leq \nabla F(\bar{\mathbf{w}}^r)^\top (\bar{\mathbf{w}}^{r+1} - \bar{\mathbf{w}}^r) + \frac{\tilde{L}}{2}\|\bar{\mathbf{w}}^{r+1} - \bar{\mathbf{w}}^r\|^2$$

$$= -\tilde{\eta}\nabla F(\bar{\mathbf{w}}^r)^\top\left(\frac{1}{NK}\sum_i\sum_k G_{i,k}^r - \nabla F(\bar{\mathbf{w}}^r) + \nabla F(\bar{\mathbf{w}}^r)\right) + \frac{\tilde{L}}{2}\|\bar{\mathbf{w}}^{r+1} - \bar{\mathbf{w}}^r\|^2$$

$$= -\tilde{\eta}\|\nabla F(\bar{\mathbf{w}}^r)\|^2 + \frac{\tilde{\eta}}{2}\|\nabla F(\bar{\mathbf{w}}^r)\|^2 + \frac{\tilde{\eta}}{2}\|\frac{1}{NK}\sum_i\sum_k G_{i,k}^r - \nabla F(\bar{\mathbf{w}}^r)\|^2 + \frac{\tilde{L}}{2}\|\bar{\mathbf{w}}^{r+1} - \bar{\mathbf{w}}^r\|^2$$

$$\leq -\frac{\tilde{\eta}}{2}\|\nabla F(\bar{\mathbf{w}}^r)\|^2 + \tilde{\eta}\|\frac{1}{NK}\sum_i\sum_k(G_{i,k}^r - \nabla F(\bar{\mathbf{w}}_k^r))\|^2 + \tilde{\eta}\|\frac{1}{K}\sum_k(\nabla F(\bar{\mathbf{w}}_k^r) - \nabla F(\bar{\mathbf{w}}^r))\|^2$$

$$+ \frac{\tilde{L}}{2}\|\bar{\mathbf{w}}^{r+1} - \bar{\mathbf{w}}^r\|^2$$

$$\leq -\frac{\tilde{\eta}}{2}\|\nabla F(\bar{\mathbf{w}}^r)\|^2 + \tilde{\eta}\frac{1}{K}\sum_k\|\frac{1}{N}\sum_i(G_{i,k}^r - \nabla F(\bar{\mathbf{w}}_k^r))\|^2 + \tilde{\eta}\frac{\tilde{L}^2}{K}\sum_k\|\bar{\mathbf{w}}_k^r - \bar{\mathbf{w}}^r\|^2 + \frac{\tilde{L}}{2}\|\bar{\mathbf{w}}^{r+1} - \bar{\mathbf{w}}^r\|^2.$$

(63)

Combining with Lemma 1 and Lemma 2, we derive

$$\frac{1}{R}\sum_r\mathbb{E}\|\nabla F(\bar{\mathbf{w}}^r)\|^2 \leq O\left(\frac{M}{\gamma RK} + \gamma\beta^2 K^2 + \gamma + \eta^2\frac{M^2}{\gamma^2} + 8\gamma M\tilde{\eta}^2 + \frac{M}{\gamma}\tilde{\eta}^2(\frac{1}{\beta RK} + \frac{\beta}{N})\right).$$

(64)

By setting parameters as in the theorem, we can conclude the proof. Further, to get $\frac{1}{R}\sum_r\mathbb{E}\|\nabla F(\bar{\mathbf{w}}^r)\|^2 \leq \epsilon^2$, we just need to set $\gamma = \epsilon^2$, $\beta = \frac{\epsilon^2}{\sqrt{M}}$, $K = \frac{\sqrt{M}}{\epsilon}$, $\eta = \frac{\epsilon^2}{M}$, $R = \frac{\sqrt{M}}{\epsilon^3}$. □

# E ANALYSIS OF FEDX1 FOR OPTIMIZING CPR WITH LINEAR $f$ WITH A LARGER BUFFER

In this section, we present the analysis of FedX1 with a larger buffer, i.e,. $\mathcal{B}_{i,1}, \mathcal{B}_{i,2}$ keeps the history of previous $\tau > 1$ rounds instead of only keep the history of the one previous round. For $\mathbf{z} \in \mathcal{S}_i$ and $\mathbf{z}' \in \mathcal{S}_j$, we define

$$G_1(\mathbf{w},\mathbf{z},\mathbf{w}',\mathbf{z}') = \nabla\ell_{ij}(h(\mathbf{w};\mathbf{z}) - h(\mathbf{w}';\mathbf{z}'))^\top\nabla h(\mathbf{w};\mathbf{z})$$

$$G_2(\mathbf{w},\mathbf{z},\mathbf{w}',\mathbf{z}') = -\nabla\ell_{ij}(h(\mathbf{w},\mathbf{z}) - h(\mathbf{w}';\mathbf{z}'))^\top\nabla h(\mathbf{w}';\mathbf{z}'),$$

(65)

We use superscript $\{r - \tau, r - 1\}$ to denote that a historical statistics sampled from the buffer is computed at some round in $(r - \tau, r - 1)$ randomly. Therefore, the

$$G_{i,k,1}^r = \nabla_1\ell_{ij}(h(\mathbf{w}_{i,k}^r;\mathbf{z}_{i,k,1}^r), h_{2,\xi}^{r-\tau,r-1})\nabla h(\mathbf{w}_{i,k}^r;\mathbf{z}_{i,k,1}^r),$$

defined similarly as (3) is equivalent to $G_1(\mathbf{w}_{i,k}^r, \mathbf{z}_{i,k,1}^r, \mathbf{w}_{j,t}^{r-\tau,r-1}, \mathbf{z}_{j,t,2}^r)$, and the

$$G_{i,k,2}^r = \nabla_2 \ell_{j'i}(h_{1,\zeta}^{r-\tau,r-1}, h(\mathbf{w}_{i,k}^r; \mathbf{z}_{i,k,2}^r),)\nabla h(\mathbf{w}_{i,k}^r; \mathbf{z}_{i,k,2}^r),$$

defined in (5) is equivalent to $G_2(\mathbf{w}_{j',t'}^{r-\tau,r-1}, \mathbf{z}_{j',t',1}^{r-\tau,r-1}, \mathbf{w}_{i,k}^r, \mathbf{z}_{i,k,2}^r)$.

We use same assumption and notations as in Appendix C Under Assumption 1, it follows that $F(\cdot)$ is $L_F$-smooth, with $L_F := 2(L_\ell C_h + C_\ell L_h)$. Similarly, $G_1, G_2$ are also Lipschtz in $\mathbf{w}$ with some constant modulus $\tilde{L}$ that depend on $C_h, C_\ell, L_\ell, L_h$.

We present the analysis in the theorem below.

**Theorem 5.** *Under Assumption 1, by setting $\eta = O(\frac{N}{R^{2/3}})$ and $K = O(\frac{1}{NR^{1/3}})$ and $\tau = O(1)$, Algorithm 2 with a larger buffer that keeps the history of last $\tau$ rounds ensures that*

$$\mathbb{E}[\frac{1}{R}\sum_{r=1}^R \|\nabla F(\bar{\mathbf{w}}^{r-\tau})\|^2] \leq O(\frac{1}{R^{2/3}}). \tag{66}$$

*Proof.* Denote $\tilde{\eta} = \eta K$. Using the $L$-smoothness of $F(\mathbf{w})$, we have

$$F(\bar{\mathbf{w}}^{r+1}) - F(\bar{\mathbf{w}}^r) \leq \nabla F(\bar{\mathbf{w}}^r)^\top (\bar{\mathbf{w}}^{r+1} - \bar{\mathbf{w}}^r) + \frac{\tilde{L}}{2}\|\bar{\mathbf{w}}^{r+1} - \bar{\mathbf{w}}^r\|^2$$

$$= -\tilde{\eta}(\nabla F(\bar{\mathbf{w}}^r) - \nabla F(\bar{\mathbf{w}}^{r-\tau}) + \nabla F(\bar{\mathbf{w}}^{r-\tau}))^\top \left(\frac{1}{NK}\sum_i\sum_k(G_{i,k,1}^r + G_{i,k,2}^r)\right) + \frac{\tilde{L}}{2}\|\bar{\mathbf{w}}^{r+1} - \bar{\mathbf{w}}^r\|^2$$

$$\leq \frac{1}{2\tilde{L}}\|\nabla F(\bar{\mathbf{w}}^r) - \nabla F(\bar{\mathbf{w}}^{r-\tau})\|^2 + 2\tilde{\eta}^2\tilde{L}\|\frac{1}{NK}\sum_i\sum_k(G_{i,k,1}^r + G_{i,k,2}^r)\|^2$$

$$- \tilde{\eta}\nabla F(\bar{\mathbf{w}}^{r-\tau})^\top \left(\frac{1}{NK}\sum_i\sum_k(G_{i,k,1}^r + G_{i,k,2}^r)\right) + \frac{\tilde{L}}{2}\|\bar{\mathbf{w}}^{r+1} - \bar{\mathbf{w}}^r\|^2$$

$$\leq 2\tilde{\eta}^2\tilde{L}\|\frac{1}{NK}\sum_i\sum_k(G_{i,k,1}^r + G_{i,k,2}^r)\|^2 + \tilde{L}\|\bar{\mathbf{w}}^{r+1} - \bar{\mathbf{w}}^r\|^2$$

$$- \tilde{\eta}\nabla F(\bar{\mathbf{w}}^{r-\tau})^\top \left(\frac{1}{NK}\sum_i\sum_k(G_{i,k,1}^r + G_{i,k,2}^r)\right), \tag{67}$$

where

$$-\mathbb{E}\left[\tilde{\eta}\nabla F(\bar{\mathbf{w}}^{r-\tau})^\top \left(\frac{1}{NK}\sum_i\sum_k(G_{i,k,1}^r + G_{i,k,2}^r)\right)\right]$$

$$= -\mathbb{E}\left[\tilde{\eta}\nabla F(\bar{\mathbf{w}}^{r-\tau})^\top \left(\frac{1}{NK}\sum_i\sum_k(G_1(\mathbf{w}_{i,k}^r, \mathbf{z}_{i,k,1}^r, \mathbf{w}_{j,t}^{r-\tau,r-1}, \mathbf{z}_{j,t,2}^{r-1}) + G_2(\mathbf{w}_{j',t'}^{r-\tau,r-1}, \mathbf{z}_{j',t',1}^{r-1}, \mathbf{w}_{i,k}^r, \mathbf{z}_{i,k,2}^r)\right.\right.$$

$$- (G_1(\bar{\mathbf{w}}^{r-\tau}, \mathbf{z}_{i,k,1}^r, \bar{\mathbf{w}}^{r-\tau}, \mathbf{z}_{j,t,2}^{r-\tau,r-1}) + G_2(\bar{\mathbf{w}}^{r-\tau}, \mathbf{z}_{j',t',1}^{r-\tau,r-1}, \bar{\mathbf{w}}^{r-\tau}, \mathbf{z}_{i,k,2}^r))$$

$$\left.\left.+ G_1(\bar{\mathbf{w}}^{r-\tau}, \mathbf{z}_{i,k,1}^r, \bar{\mathbf{w}}^{r-\tau}, \mathbf{z}_{j,t,2}^{r-1}) + G_2(\bar{\mathbf{w}}^{r-\tau}, \mathbf{z}_{j',t',1}^{r-1}, \bar{\mathbf{w}}^{r-\tau}, \mathbf{z}_{i,k,2}^r)))\right)\right]$$

$$= \frac{\tilde{\eta}}{4}\mathbb{E}\|\nabla F(\bar{\mathbf{w}}^{r-\tau})\|^2 + 8\tilde{\eta}\tilde{L}^2\mathbb{E}\|\bar{\mathbf{w}}^r - \bar{\mathbf{w}}^{r-\tau}\|^2 + 8\tilde{\eta}\tilde{L}^2\frac{1}{NK}\sum_i\sum_k\mathbb{E}\|\bar{\mathbf{w}}^r - \mathbf{w}_{i,k}^r\|^2$$

$$- \mathbb{E}\left[\tilde{\eta}\nabla F(\bar{\mathbf{w}}^{r-\tau})^\top \left(\frac{1}{NK}\sum_i\sum_k(G_1(\bar{\mathbf{w}}^{r-\tau}, \mathbf{z}_{i,k,1}^r, \bar{\mathbf{w}}^{r-\tau}, \mathbf{z}_{j,t,2}^{r-\tau,r-1}) + G_2(\bar{\mathbf{w}}^{r-\tau}, \mathbf{z}_{j',t',1}^{r-\tau,r-1}, \bar{\mathbf{w}}^{r-\tau}, \mathbf{z}_{i,k,2}^r)\right.\right.$$

$$\left.\left.- \nabla F(\bar{\mathbf{w}}^{r-\tau}) + \nabla F(\bar{\mathbf{w}}^{r-\tau}))\right)\right]$$

$$\leq \frac{\tilde{\eta}}{4}\mathbb{E}\|\nabla F(\bar{\mathbf{w}}^{r-\tau})\|^2 + 8\tilde{\eta}\tilde{L}^2\mathbb{E}\|\bar{\mathbf{w}}^r - \bar{\mathbf{w}}^{r-\tau}\|^2 + 8\tilde{\eta}\tilde{L}^2\frac{1}{NK}\sum_i\sum_k\mathbb{E}\|\bar{\mathbf{w}}^{r-\tau} - \mathbf{w}_{i,k}^r\|^2 - \tilde{\eta}\mathbb{E}\|\nabla F(\bar{\mathbf{w}}^{r-\tau})\|^2. \tag{68}$$

By the updating rule,

$$\mathbb{E}\|\bar{\mathbf{w}}_{r+1} - \bar{\mathbf{w}}_r\|^2 = \tilde{\eta}^2 \mathbb{E}\|\frac{1}{NK}\sum_i\sum_k(G_{i,k,1}^r + G_{i,k,2}^r)\|^2$$

$$= \tilde{\eta}^2 \mathbb{E}\|\frac{1}{NK}\sum_i\sum_k(G_1(\mathbf{w}_{i,k}^r, \mathbf{z}_{i,k,1}^r, \mathbf{w}_{j,t}^{r-\tau,r-1}, \mathbf{z}_{j,t,2}^{r-\tau,r-1}) + G_2(\mathbf{w}_{j',t'}^{r-\tau,r-1}, \mathbf{z}_{j',t',1}^{r-\tau,r-1}, \mathbf{w}_{i,k}^r, \mathbf{z}_{i,k,2}^r))\|^2$$

$$\leq 5\tilde{\eta}^2 \mathbb{E}\left\|\frac{1}{NK}\sum_i\sum_k[G_1(\mathbf{w}_{i,k}^r, \mathbf{z}_{i,k,1}^r, \mathbf{w}_{j,t}^{r-\tau,r-1}, \mathbf{z}_{j,t,2}^{r-\tau,r-1}) + G_2(\mathbf{w}_{j',t'}^{r-\tau,r-1}, \mathbf{z}_{j',t',1}^{r-\tau,r-1}, \mathbf{w}_{i,k}^r, \mathbf{z}_{i,k,2}^r)]\right.$$

$$\left. - \frac{1}{NK}\sum_i\sum_k[G_1(\bar{\mathbf{w}}^{r-\tau}, \mathbf{z}_{i,k,1}^r, \bar{\mathbf{w}}^{r-\tau}, \mathbf{z}_{j,t,2}^{r-1}) + G_2(\bar{\mathbf{w}}^{r-\tau}, \mathbf{z}_{j',t',1}^{r-1}, \bar{\mathbf{w}}^{r-\tau}, \mathbf{z}_{i,k,2}^r)]\right\|^2$$

$$+ 5\tilde{\eta}^2 \mathbb{E}\left\|\frac{1}{NK}\sum_i\sum_k[G_1(\bar{\mathbf{w}}^{r-\tau}, \mathbf{z}_{i,k,1}^r, \bar{\mathbf{w}}^{r-\tau}, \mathbf{z}_{j,t,2}^{r-1}) + G_2(\bar{\mathbf{w}}^{r-\tau}, \mathbf{z}_{j',t',1}^{r-1}, \bar{\mathbf{w}}^{r-\tau}, \mathbf{z}_{i,k,2}^r) - \nabla F_i(\bar{\mathbf{w}}^{r-\tau})]\right\|^2$$

$$+ 5\tilde{\eta}^2 \mathbb{E}\left\|\nabla F(\bar{\mathbf{w}}^{r-\tau})\right\|^2$$

$$\leq 10\tilde{\eta}^2 \frac{\tilde{L}^2}{NK}\sum_i\sum_k \mathbb{E}\|\mathbf{w}_{i,k}^r - \bar{\mathbf{w}}^r\|^2 + 10\tilde{\eta}^2\frac{\tilde{L}^2}{NK}\sum_i\sum_k\mathbb{E}\|\mathbf{w}_{i,k}^{r-1} - \bar{\mathbf{w}}^{r-1}\|^2 + 10\tilde{\eta}^2\mathbb{E}\|\bar{\mathbf{w}}^r - \bar{\mathbf{w}}^{r-1}\|^2$$

$$+ 10\tilde{\eta}^2\frac{\sigma^2}{NK} + 10\tilde{\eta}^2\mathbb{E}\|F(\bar{\mathbf{w}}^{r-\tau})\|^2. \tag{69}$$

Thus,

$$\frac{1}{R}\sum_r \mathbb{E}\|\bar{\mathbf{w}}^{r+1} - \bar{\mathbf{w}}^r\|^2$$
$$\leq \frac{1}{R}\sum_r\left[40\tilde{\eta}^2\frac{1}{NK}\sum_i\sum_k\mathbb{E}\|\mathbf{w}_{i,k}^r - \bar{\mathbf{w}}^r\|^2 + 20\tilde{\eta}^2\frac{\sigma^2}{NK} + 20\tilde{\eta}^2\mathbb{E}\|F(\bar{\mathbf{w}}^{r-1})\|^2\right]. \tag{70}$$

Since $\ell_{ij}(\cdot)$ is $C_\ell$ Lipschitz, we have

$$\mathbb{E}\|\bar{\mathbf{w}}^r - \mathbf{w}_{i,k}^r\|^2 \leq \eta^2 K^2 C_\ell^2. \tag{71}$$

and

$$\mathbb{E}\|\bar{\mathbf{w}}^r - \mathbf{w}^{r-\tau}\|^2 \leq \eta^2 K^2 \tau^2 C_\ell^2. \tag{72}$$

Thus,

$$\frac{1}{R}\sum_{r=1}^R \mathbb{E}\|F(\bar{\mathbf{w}}_{r-\tau})\|^2 \leq O\left(\frac{2(F(\bar{\mathbf{w}}_0) - F_*)}{\tilde{\eta}R} + \tilde{\eta}^2(D^2 + \sigma^2) + \tilde{\eta}^2\tau^2 C_\ell^2 + 40\tilde{\eta}\frac{\sigma^2}{NK}\right). \tag{73}$$

Set $\eta$ and $K$ as in theorem, we conclude the proof. Further, to ensure $\frac{1}{R}\sum_{r=1}^R \mathbb{E}\|F(\bar{\mathbf{w}}^{r-\tau})\|^2 \leq \epsilon^2$, we just need to set $\eta = N\epsilon^2$, $K = 1/N\epsilon$, $\tilde{\eta} = \eta K = \epsilon$ and $\tau = O(1)$, then number of communication rounds is $R = O(\frac{1}{\epsilon^3})$, sample complexity on each machine is $O(\frac{1}{N\epsilon^4})$. $\qquad\square$

## F  FEDX2 FOR OPTIMIZING CPR WITH NON-LINEAR $f$ WITH MEMORY BANK

In this section, we present the analysis of FedX2 with a larger buffer, i.e., $\mathcal{B}_{i,1}, \mathcal{B}_{i,2}$ keeps the history of previous $\tau > 1$ rounds instead of only keep the history of the one previous round. We use the same notations and assumptions as in Appendix D. The framework of the proof is similar as in Appendix D except that we need to handle the extra error caused by the large buffer.

**Theorem 6.** *Suppose Assumption 2 holds, denoting $M = \max_i |\mathcal{S}_i^1|$ as the largest number of data on a single machine, by setting $\gamma = O(\frac{M^{1/3}}{R^{2/3}})$, $\beta = O(\frac{1}{M^{1/6}R^{2/3}})$, $\eta = O(\frac{1}{M^{2/3}R^{2/3}})$, $\tau = O(M^{1/4})$ and $K = O(M^{1/3}R^{1/3})$, Algorithm 2 ensures that $\mathbb{E}\left[\frac{1}{R}\sum_{r=1}^R \|\nabla F(\bar{\mathbf{w}}^r)\|^2\right] \leq O(\frac{1}{R^{2/3}})$.*

*Proof.* First, we need to handle the $\mathbf{u}$ estimator. Denote $g(\mathbf{w}_1, p, \mathbf{w}_2, q) = \ell(h(\mathbf{w}_1; p), h(\mathbf{w}_2, q))$.

$$\mathbf{u}_{i,k}^r(\mathbf{z}) = \begin{cases} \mathbf{u}_{i,k-1}^r(\mathbf{z}) - \gamma(\mathbf{u}_{i,k-1}^r(\mathbf{z}) - \ell(h(\mathbf{w}_{i,k}^r; \mathbf{z}_{i,k,1}^r), h(\mathbf{w}_{j,t}^{r-\tau,r-1}; \mathbf{z}_{j,t,2}^{r-\tau,r-1}))) & \mathbf{z} = \mathbf{z}_{i,k,1}^r \\ \mathbf{u}_{i,k-1}^r(\mathbf{z}) & \mathbf{z} \neq \mathbf{z}_{i,k,1}^r \end{cases}$$
(74)

Or equivalently,

$$\mathbf{u}_{i,k}^r(\mathbf{z}) = \begin{cases} \mathbf{u}_{i,k-1}^r(\mathbf{z}) - \gamma(\mathbf{u}_{i,k-1}^r(\mathbf{z}) - g(\mathbf{w}_{i,k}^r, \mathbf{z}_{i,k,1}^r, \mathbf{w}_{j,t}^{r-\tau,r-1}, \mathbf{z}_{j,t,2}^{r-\tau,r-1})) & \mathbf{z} = \mathbf{z}_{i,k,1}^r \\ \mathbf{u}_{i,k-1}^r(\mathbf{z}) & \mathbf{z} \neq \mathbf{z}_{i,k,1}^r \end{cases}$$
(75)

Define $\bar{\mathbf{u}}_k^r = (\mathbf{u}_{1,k}^r, \mathbf{u}_{2,k}^r, ..., \mathbf{u}_{N,k}^r)$ and $\bar{\mathbf{w}}_k^r = \frac{1}{N}\sum_{i=1}^N \mathbf{w}_{i,k}^r$. We have

$$\frac{1}{2N}\sum_{i=1}^N \frac{1}{|\mathcal{S}_1^i|}\sum_{\mathbf{z}\in\mathcal{S}_1^i}\mathbb{E}\|\mathbf{u}_{i,k}^r(\mathbf{z}) - g(\bar{\mathbf{w}}_k^r, \mathbf{z}, \bar{\mathbf{w}}_k^r, \mathcal{S}_2)\|^2$$

$$= \frac{1}{N}\sum_i \frac{1}{|\mathcal{S}_1^i|}\sum_{\mathbf{z}\in\mathcal{S}_1^i}\mathbb{E}\Big[\frac{1}{2}\|\mathbf{u}_{i,k-1}^r(\mathbf{z}) - g(\bar{\mathbf{w}}_k^r, \mathbf{z}, \bar{\mathbf{w}}_k^r, \mathcal{S}_2)\|^2 + \langle\mathbf{u}_{i,k-1}^r(\mathbf{z}) - g(\bar{\mathbf{w}}_k^r, \mathbf{z}, \bar{\mathbf{w}}_k^r, \mathcal{S}_2), \mathbf{u}_{i,k}^r(\mathbf{z}) - \mathbf{u}_{i,k-1}^r(\mathbf{z})\rangle$$

$$+ \frac{1}{2}\|\mathbf{u}_{i,k}^r(\mathbf{z}) - \mathbf{u}_{i,k-1}^r(\mathbf{z})\|^2\Big]$$

$$= \frac{1}{N}\sum_i \frac{1}{|\mathcal{S}_1^i|}\sum_{\mathbf{z}\in\mathcal{S}_1^i}\mathbb{E}\Big[\frac{1}{2}\|\mathbf{u}_{i,k-1}^r(\mathbf{z}) - g(\bar{\mathbf{w}}_k^r, \mathbf{z}, \bar{\mathbf{w}}_k^r, \mathcal{S}_2)\|^2$$

$$+ \frac{1}{|\mathcal{S}_1^i|}\langle\mathbf{u}_{i,k-1}^r(\mathbf{z}_{i,k,1}^r) - g(\bar{\mathbf{w}}_k^r, \mathbf{z}, \bar{\mathbf{w}}_k^r, \mathcal{S}_2), \mathbf{u}_{i,k}^r(\mathbf{z}_{i,k,1}^r) - \mathbf{u}_{i,k-1}^r(\mathbf{z}_{i,k,1}^r)\rangle$$

$$+ \frac{1}{2|\mathcal{S}_1^i|}\|\mathbf{u}_{i,k}^r(\mathbf{z}_{i,k,1}^r) - \mathbf{u}_{i,k-1}^r(\mathbf{z}_{i,k,1}^r)\|^2\Big]$$

$$= \frac{1}{N}\sum_i \frac{1}{|\mathcal{S}_1^i|}\sum_{\mathbf{z}\in\mathcal{S}_1^i}\mathbb{E}\Big[\frac{1}{2}\|\mathbf{u}_{i,k-1}^r(\mathbf{z}) - g(\bar{\mathbf{w}}_k^r, \mathbf{z}, \bar{\mathbf{w}}_k^r, \mathcal{S}_2)\|^2$$

$$+ \frac{1}{|\mathcal{S}_1^i|}\langle\mathbf{u}_{i,k-1}^r(\mathbf{z}_{i,k,1}^r) - g(\mathbf{w}_{i,k}^r, \mathbf{z}_{i,k,1}^r, \mathbf{w}_{j,t}^{r-\tau,r-1}, \mathbf{z}_{j,t,2}^{r-\tau,r-1}), \mathbf{u}_{i,k}^r(\mathbf{z}_{i,k,1}^r) - \mathbf{u}_{i,k-1}^r(\mathbf{z}_{i,k,1}^r)\rangle$$

$$+ \frac{1}{|\mathcal{S}_1^i|}\langle g(\mathbf{w}_{i,k}^r, \mathbf{z}_{i,k,1}^r, \mathbf{w}_{j,t}^{r-\tau,r-1}, \mathbf{z}_{j,t,2}^{r-\tau,r-1}) - g(\bar{\mathbf{w}}_k^r, \mathbf{z}_{i,k,1}^r, \bar{\mathbf{w}}_k^r, \mathcal{S}_2), \mathbf{u}_{i,k}^r(\mathbf{z}_{i,k,1}^r) - \mathbf{u}_{i,k-1}^r(\mathbf{z}_{i,k,1}^r)\rangle$$

$$+ \frac{1}{2|\mathcal{S}_1^i|}\|\mathbf{u}_{i,k}^r(\mathbf{z}_{i,k,1}^r) - \mathbf{u}_{i,k-1}^r(\mathbf{z}_{i,k,1}^r)\|^2\Big],$$
(76)

where
$$\langle\mathbf{u}_{i,k-1}^r(\mathbf{z}_{i,k,1}^r) - g(\mathbf{w}_{i,k}^r, \mathbf{z}_{i,k,1}^r, \mathbf{w}_{j,t}^{r-\tau,r-1}, \mathbf{z}_{j,t,2}^{r-\tau,r-1}), \mathbf{u}_{i,k}^r(\mathbf{z}_{i,k,1}^r) - \mathbf{u}_{i,k-1}^r(\mathbf{z}_{i,k,1}^r)\rangle$$

$$= \langle\mathbf{u}_{i,k-1}^r(\mathbf{z}_{i,k,1}^r) - g(\mathbf{w}_{i,k}^r, \mathbf{z}_{i,k,1}^r, \mathbf{w}_{j,t}^{r-\tau,r-1}, \mathbf{z}_{j,t,2}^{r-\tau,r-1}), g(\bar{\mathbf{w}}_k^r, \mathbf{z}_{i,k,1}^r, \bar{\mathbf{w}}_k^r, \mathcal{S}_2) - \mathbf{u}_{i,k-1}^r(\mathbf{z}_{i,k,1}^r)\rangle$$

$$+ \langle\mathbf{u}_{i,k-1}^r(\mathbf{z}_{i,k,1}^r) - g(\mathbf{w}_{i,k}^r, \mathbf{z}_{i,k,1}^r, \mathbf{w}_{j,t}^{r-\tau,r-1}, \mathbf{z}_{j,t,2}^{r-\tau,r-1}), \mathbf{u}_{i,k}^r(\mathbf{z}_{i,k,1}^r) - g(\bar{\mathbf{w}}_k^r, \mathbf{z}_{i,k,1}^r, \bar{\mathbf{w}}_k^r, \mathcal{S}_2)\rangle$$

$$= \langle\mathbf{u}_{i,k-1}^r(\mathbf{z}_{i,k,1}^r) - g(\mathbf{w}_{i,k}^r, \mathbf{z}_{i,k,1}^r, \mathbf{w}_{j,t}^{r-\tau,r-1}, \mathbf{z}_{j,t,2}^{r-\tau,r-1}), g(\bar{\mathbf{w}}_k^r, \mathbf{z}_{i,k,1}^r, \bar{\mathbf{w}}_k^r, \mathcal{S}_2) - \mathbf{u}_{i,k-1}^r(\mathbf{z}_{i,k,1}^r)\rangle$$

$$+ \frac{1}{\gamma}\langle\mathbf{u}_{i,k-1}^r(\mathbf{z}_{i,k,1}^r) - \mathbf{u}_{i,k}^r(\mathbf{z}_{i,k,1}^r), \mathbf{u}_{i,k}^r(\mathbf{z}_{i,k,1}^r) - g(\bar{\mathbf{w}}_k^r, \mathbf{z}_{i,k,1}^r, \bar{\mathbf{w}}_k^r, \mathcal{S}_2)\rangle$$

$$\leq \langle\mathbf{u}_{i,k-1}^r(\mathbf{z}_{i,k,1}^r) - g(\mathbf{w}_{i,k}^r, \mathbf{z}_{i,k,1}^r, \mathbf{w}_{j,t}^{r-\tau,r-1}, \mathbf{z}_{j,t,2}^{r-\tau,r-1}), g(\bar{\mathbf{w}}_k^r, \mathbf{z}_{i,k,1}^r, \bar{\mathbf{w}}_{i,k}^r, \mathcal{S}_2) - \mathbf{u}_{i,k-1}^r(\mathbf{z}_{i,k,1}^r)\rangle$$

$$+ \frac{1}{2\gamma}(\|\mathbf{u}_{i,k-1}^r(\mathbf{z}_{i,k,1}^r) - g(\bar{\mathbf{w}}_k^r, \mathbf{z}_{i,k,1}^r, \bar{\mathbf{w}}_k^r, \mathcal{S}_2)\|^2 - \|\mathbf{u}_{i,k}^r(\mathbf{z}_{i,k,1}^r) - \mathbf{u}_{i,k-1}^r(\mathbf{z}_{i,k,1}^r)\|^2$$

$$- \|\mathbf{u}_{i,k}^r(\mathbf{z}_{i,k,1}^r) - g(\bar{\mathbf{w}}_k^r, \mathbf{z}_{i,k,1}^r, \bar{\mathbf{w}}_k^r, \mathcal{S}_2)\|^2).$$
(77)

If $\gamma \leq \frac{1}{9}$, we have
$$
-\frac{1}{2}\left(\frac{1}{\gamma} - 1 - \frac{\gamma+1}{4\gamma}\right)\|\mathbf{u}_{i,k}^r(\mathbf{z}_{i,k,1}^r) - \mathbf{u}_{i,k-1}^r(\mathbf{z}_{i,k,1}^r)\|^2
$$
$$
+ \langle g(\mathbf{w}_{i,k}^r, \mathbf{z}_{i,k,1}^r, \mathbf{w}_{j,t}^{r-\tau,r-1}, \mathbf{z}_{j,t,2}^{r-\tau,r-1}) - g(\bar{\mathbf{w}}_k^r, \mathbf{z}_{i,k,1}^r, \bar{\mathbf{w}}_k^r, \mathcal{S}_2), \mathbf{u}_{i,k}^r(\mathbf{z}_{i,k,1}^r) - \mathbf{u}_{i,k-1}^r(\mathbf{z}_{i,k,1}^r)\rangle
$$
$$
\leq -\frac{1}{4\gamma}\|\mathbf{u}_{i,k}^r(\mathbf{z}_{i,k,1}^r) - \mathbf{u}_{i,k-1}^r(\mathbf{z}_{i,k,1}^r)\|^2 + \gamma\|g(\mathbf{w}_{i,k}^r, \mathbf{z}_{i,k,1}^r, \mathbf{w}_{j,t}^{r-\tau,r-1}, \mathbf{z}_{j,t,2}^{r-\tau,r-1}) - g(\bar{\mathbf{w}}_k^r, \mathbf{z}_{i,k,1}^r, \bar{\mathbf{w}}_k^r, \mathcal{S}_2)\|^2
$$
$$
+ \frac{1}{4\gamma}\|\mathbf{u}_{i,k}^r(\mathbf{z}_{i,k,1}^r) - \mathbf{u}_{i,k-1}^r(\mathbf{z}_{i,k,1}^r)\|^2
$$
$$
\leq \gamma\|g(\mathbf{w}_{i,k}^r, \mathbf{z}_{i,k,1}^r, \mathbf{w}_{j,t}^{r-\tau,r-1}, \mathbf{z}_{j,t,2}^{r-\tau,r-1}) - g(\bar{\mathbf{w}}_k^r, \mathbf{z}_{i,k,1}^r, \bar{\mathbf{w}}_k^r, \mathcal{S}_2)\|^2
$$
$$
\leq 4\gamma\|g(\bar{\mathbf{w}}^{r-\tau}, \mathbf{z}_{i,k,1}^r, \bar{\mathbf{w}}^{r-\tau}, \mathbf{z}_{j,t,2}^{r-\tau,r-1}) - g(\bar{\mathbf{w}}^{r-\tau}, \mathbf{z}_{i,k,1}^r, \bar{\mathbf{w}}^{r-\tau}, \mathcal{S}_2)\|^2 + 4\gamma\tilde{L}\|\bar{\mathbf{w}}^r - \bar{\mathbf{w}}^{r-\tau}\|^2
$$
$$
+ 4\gamma\tilde{L}\|\mathbf{w}_{i,k}^r - \bar{\mathbf{w}}^r\|^2 + 4\gamma\tilde{L}\|\mathbf{w}_{i,k}^{r-1} - \bar{\mathbf{w}}^{r-1}\|^2
$$
$$
4\gamma\sigma^2 + 4\gamma\tilde{L}\|\bar{\mathbf{w}}^r - \bar{\mathbf{w}}^{r-\tau}\|^2 + 4\gamma\tilde{L}\|\mathbf{w}_{i,k}^r - \bar{\mathbf{w}}^r\|^2 + 4\gamma\tilde{L}\|\mathbf{w}_{i,k}^{r-1} - \bar{\mathbf{w}}^{r-1}\|^2
$$
$$(78)$$

Then, we have
$$
\frac{1}{2}\|\bar{\mathbf{u}}_k^r - g(\bar{\mathbf{w}}_k^r)\|^2 \leq \frac{1}{2}\|\bar{\mathbf{u}}_{k-1}^r - g(\bar{\mathbf{w}}_{i,k}^r)\|^2
$$
$$
+ \frac{1}{N}\sum_i \frac{1}{|\mathcal{S}_1^i|}\left[\frac{1}{2\gamma}\|\mathbf{u}_{i,k-1}^r(\mathbf{z}_{i,k,1}^r) - g(\mathbf{w}_k^r, \mathbf{z}_{i,k,1}^r, \mathbf{w}_k^r, \mathcal{S}_2)\|^2 - \frac{1}{2\gamma}\|\mathbf{u}_{i,k}^r(\mathbf{z}_{i,k,1}^r) - g(\mathbf{w}_k^r, \mathbf{z}_{i,k,1}^r, \mathbf{w}_k^r, \mathcal{S}_2)\|^2\right.
$$
$$
- \frac{\gamma+1}{8\gamma}\|\mathbf{u}_{i,k}^r(\mathbf{z}_{i,k,1}^r) - \mathbf{u}_{i,k-1}^r(\mathbf{z}_{i,k,1}^r)\|^2 + \gamma\|g(\bar{\mathbf{w}}^{r-\tau}, \mathbf{z}_{i,k,1}^r, \bar{\mathbf{w}}^{r-\tau}, \mathbf{z}_{j,t,2}^{r-1}) - g(\bar{\mathbf{w}}^{r-\tau}, \mathbf{z}_{i,k,1}^r, \bar{\mathbf{w}}^{r-\tau}, \mathcal{S}_2)\|^2
$$
$$
+ 4\gamma\tilde{L}\|\bar{\mathbf{w}}^r - \bar{\mathbf{w}}^{r-1}\|^2 + 4\gamma\tilde{L}\|\mathbf{w}_{i,k}^r - \mathbf{w}^r\|^2 + 4\gamma\tilde{L}^2\|\mathbf{w}_{i,k}^{r-1} - \mathbf{w}^{r-1}\|^2
$$
$$
\left. + \langle \mathbf{u}_{i,k-1}^r(\mathbf{z}_{i,k,1}^r) - g(\mathbf{w}_{i,k}^r, \mathbf{z}_{i,k,1}^r, \mathbf{w}_{j,t}^{r-1}, \mathbf{z}_{j,t,2}^{r-1}), g(\bar{\mathbf{w}}_k^r, \mathbf{z}_{i,k,1}^r, \bar{\mathbf{w}}_k^r, \mathcal{S}_2) - \mathbf{u}_{i,k-1}^r(\mathbf{z}_{i,k,1}^r)\rangle\right].
$$
$$(79)$$

Note that $\sum_{\mathbf{z}\neq\mathbf{z}_{i,k,1}^r}\|\mathbf{u}_{i,k}^r(\mathbf{z}) - g(\bar{\mathbf{w}}_{k+1}^r, \mathbf{z}, \bar{\mathbf{w}}_{k+1}^r, \mathcal{S}_2)\|^2 = \sum_{\mathbf{z}\neq\mathbf{z}_{i,k,1}^r}\|\mathbf{u}_{i,k+1}^r(\mathbf{z}) - g(\mathbf{w}_{k+1}^r, \mathbf{z}, \bar{\mathbf{w}}_{k+1}^r, \mathcal{S}_2)\|^2$, which implies
$$
\frac{1}{2\gamma}\left(\|\mathbf{u}_{i,k-1}^r(\mathbf{z}_{i,k,1}^r) - g(\bar{\mathbf{w}}_k^r, \mathbf{z}_{i,k,1}^r, \bar{\mathbf{w}}_k^r, \mathcal{S}_2)\|^2 - \|\mathbf{u}_{i,k}^r(\mathbf{z}_{i,k,1}^r) - g(\bar{\mathbf{w}}_k^r, \mathbf{z}_{i,k,1}^r, \bar{\mathbf{w}}_k^r, \mathcal{S}_2)\|^2\right)
$$
$$
= \frac{1}{2\gamma}\sum_{\mathbf{z}\in\mathcal{S}_1^i}\left(\|\mathbf{u}_{i,k-1}^r(\mathbf{z}) - g(\bar{\mathbf{w}}_k^r, \mathbf{z}, \bar{\mathbf{w}}_k^r, \mathcal{S}_2)\|^2 - \|\mathbf{u}_{i,k}^r(\mathbf{z}) - g(\bar{\mathbf{w}}_k^r, \mathbf{z}, \bar{\mathbf{w}}_k^r, \mathcal{S}_2)\|^2\right).
$$
$$(80)$$

Besides, we have
$$
\mathbb{E}\langle \mathbf{u}_{i,k-1}^r(\mathbf{z}_{i,k,1}^r) - g(\mathbf{w}_{i,k}^r, \mathbf{z}_{i,k,1}^r, \mathbf{w}_{j,t}^{r-\tau,r-1}, \mathbf{z}_{j,t,2}^{r-\tau,r-1}), g(\bar{\mathbf{w}}_k^r, \mathbf{z}_{i,k,1}^r, \bar{\mathbf{w}}_k^r, \mathcal{S}_2) - \mathbf{u}_{i,k-1}^r(\mathbf{z}_{i,k,1}^r)\rangle
$$
$$
= \mathbb{E}\langle \mathbf{u}_{i,k-1}^r(\mathbf{z}_{i,k,1}^r) - g(\bar{\mathbf{w}}^{r-\tau}, \mathbf{z}_{i,k,1}^r, \bar{\mathbf{w}}^{r-\tau}, \mathbf{z}_{j,t,2}^{r-\tau,r-1}), g(\bar{\mathbf{w}}_k^r, \mathbf{z}_{i,k,1}^r, \bar{\mathbf{w}}_k^r, \mathcal{S}_2) - \mathbf{u}_{i,k-1}^r(\mathbf{z}_{i,k,1}^r)\rangle
$$
$$
+ \mathbb{E}\langle g(\bar{\mathbf{w}}^{r-\tau}, \mathbf{z}_{i,k,1}^r, \bar{\mathbf{w}}^{r-\tau}, \mathbf{z}_{j,t,2}^{r-\tau,r-1}) - g(\mathbf{w}_{i,k}^r, \mathbf{z}_{i,k,1}^r, \mathbf{w}_{j,t}^{r-1}, \mathbf{z}_{j,t,2}^{r-1}), g(\bar{\mathbf{w}}_k^r, \mathbf{z}_{i,k,1}^r, \bar{\mathbf{w}}_k^r, \mathcal{S}_2) - \mathbf{u}_{i,k-1}^r(\mathbf{z}_{i,k,1}^r)\rangle
$$
$$
\leq \mathbb{E}\langle \mathbf{u}_{i,k-1}^r(\mathbf{z}_{i,k,1}^r) - g(\bar{\mathbf{w}}^{r-\tau}, \mathbf{z}_{i,k,1}^r, \bar{\mathbf{w}}^{r-\tau}, \mathbf{z}_{j,t,2}^{r-1}), g(\bar{\mathbf{w}}_k^r, \mathbf{z}_{i,k,1}^r, \bar{\mathbf{w}}_k^r, \mathcal{S}_2) - g(\bar{\mathbf{w}}^{r-\tau}, \mathbf{z}_{i,k,1}^r, \bar{\mathbf{w}}^{r-\tau}, \mathcal{S}_2)\rangle
$$
$$
+ \mathbb{E}\langle \mathbf{u}_{i,k-1}^r(\mathbf{z}_{i,k,1}^r) - g(\bar{\mathbf{w}}^{r-\tau}, \mathbf{z}_{i,k,1}^r, \bar{\mathbf{w}}^{r-\tau}, \mathbf{z}_{j,t,2}^{r-\tau,r-1}), g(\bar{\mathbf{w}}^{r-\tau}, \mathbf{z}_{i,k,1}^r, \bar{\mathbf{w}}^{r-1}, \mathcal{S}_2) - \mathbf{u}_{i,k-1}^r(\mathbf{z}_{i,k,1}^r)\rangle
$$
$$
+ \|\bar{\mathbf{w}}^{r-\tau} - \mathbf{w}_{i,k}^r\|^2 + \frac{1}{4}\|g(\bar{\mathbf{w}}^r, \mathbf{z}_{i,k,1}^r, \bar{\mathbf{w}}_k^r, \mathcal{S}_2) - \mathbf{u}_{i,k-1}^r(\mathbf{z}_{i,k,1}^r)\|^2
$$
$$
\leq \gamma\|\mathbf{u}_{i,k-1}^r(\mathbf{z}_{i,k,1}^r) - g(\bar{\mathbf{w}}^{r-1}, \mathbf{z}_{i,k,1}^r, \bar{\mathbf{w}}^{r-\tau}, \mathbf{z}_{j,t,2}^{r-\tau,r-1})\|^2 + \frac{1}{\gamma}\|\bar{\mathbf{w}}_k^r - \bar{\mathbf{w}}^{r-\tau}\|^2
$$
$$
+ \mathbb{E}\langle \mathbf{u}_{i,k-1}^r(\mathbf{z}_{i,k,1}^r) - g(\bar{\mathbf{w}}^{r-1}, \mathbf{z}_{i,k,1}^r, \bar{\mathbf{w}}^{r-\tau}, \mathbf{z}_{j,t,2}^{r-\tau,r-1}), g(\bar{\mathbf{w}}^{r-\tau}, \mathbf{z}_{i,k,1}^r, \bar{\mathbf{w}}^{r-\tau,r-1}, \mathcal{S}_2) - \mathbf{u}_{i,k-1}^r(\mathbf{z}_{i,k,1}^r)\rangle
$$
$$
+ \|\bar{\mathbf{w}}^{r-\tau} - \mathbf{w}_{i,k}^r\|^2 + \frac{1}{4}\|g(\bar{\mathbf{w}}^r, \mathbf{z}_{i,k,1}^r, \bar{\mathbf{w}}_k^r, \mathcal{S}_2) - \mathbf{u}_{i,k-1}^r(\mathbf{z}_{i,k,1}^r)\|^2,
$$
$$(81)$$

where
$$\mathbb{E}\langle \mathbf{u}_{i,k-1}^r(\mathbf{z}_{i,k,1}^r) - g(\bar{\mathbf{w}}^{r-\tau,r-1}, \mathbf{z}_{i,k,1}^r, \bar{\mathbf{w}}^{r-\tau,r-1}, \mathbf{z}_{j,t,2}^{r-1}), g(\bar{\mathbf{w}}^{r-\tau,r-1}, \mathbf{z}_{i,k,1}^r, \bar{\mathbf{w}}^{r-\tau}, \mathcal{S}_2) - \mathbf{u}_{i,k-1}^r(\mathbf{z}_{i,k,1}^r)\rangle$$

$$= \mathbb{E}\langle \mathbf{u}_{i,k-1}^r(\mathbf{z}_{i,k,1}^r) - \mathbf{u}_{i,0}^{r-tau}(\mathbf{z}_{i,k,1}^r) + \mathbf{u}_{i,0}^{r-\tau}(\mathbf{z}_{i,k,1}^r) - g(\bar{\mathbf{w}}^{r-\tau}, \mathbf{z}_{i,k,1}^r, \bar{\mathbf{w}}^{r-\tau}, \mathbf{z}_{j,t,2}^{r-\tau,r-1}),$$
$$g(\bar{\mathbf{w}}^{r-\tau}, \mathbf{z}_{i,k,1}^r, \bar{\mathbf{w}}^{r-\tau}, \mathcal{S}_2) - \mathbf{u}_{i,0}^{r-\tau}(\mathbf{z}_{i,k,1}^r) + \mathbf{u}_{i,0}^{r-\tau}(\mathbf{z}_{i,k,1}^r) - \mathbf{u}_{i,k-1}^r(\mathbf{z}_{i,k,1}^r)\rangle$$

$$\leq \mathbb{E}\langle \mathbf{u}_{i,k-1}^r(\mathbf{z}_{i,k,1}^r) - \mathbf{u}_{i,0}^{r-\tau}(\mathbf{z}_{i,k,1}^r), g(\bar{\mathbf{w}}^{r-\tau}, \mathbf{z}_{i,k,1}^r, \bar{\mathbf{w}}^{r-\tau}, \mathcal{S}_2) - \mathbf{u}_{i,0}^{r-\tau}(\mathbf{z}_{i,k,1}^r)\rangle$$

$$+ \mathbb{E}\langle \mathbf{u}_{i,k-1}^r(\mathbf{z}_{i,k,1}^r) - \mathbf{u}_{i,0}^{r-\tau}(\mathbf{z}_{i,k,1}^r), \mathbf{u}_{i,0}^{r-\tau}(\mathbf{z}_{i,k,1}^r) - \mathbf{u}_{i,k-1}^r(\mathbf{z}_{i,k,1}^r)\rangle$$

$$+ \mathbb{E}\langle \mathbf{u}_{i,0}^{r-\tau}(\mathbf{z}_{i,k,1}^r) - g(\bar{\mathbf{w}}^{r-\tau}, \mathbf{z}_{i,k,1}^r, \bar{\mathbf{w}}^{r-\tau}, \mathbf{z}_{j,t,2}^{r-\tau,r-1}), g(\bar{\mathbf{w}}^{r-1}, \mathbf{z}_{i,k,1}^r, \bar{\mathbf{w}}^{r-1}, \mathcal{S}_2) - \mathbf{u}_{i,0}^{r-\tau}(\mathbf{z}_{i,k,1}^r)\rangle$$

$$+ \mathbb{E}\langle \mathbf{u}_{i,0}^{r-\tau}(\mathbf{z}_{i,k,1}^r) - g(\bar{\mathbf{w}}^{r-\tau}, \mathbf{z}_{i,k,1}^r, \bar{\mathbf{w}}^{r-\tau}, \mathbf{z}_{j,t,2}^{r-\tau,r-1}), \mathbf{u}_{i,0}^{r-\tau}(\mathbf{z}_{i,k,1}^r) - \mathbf{u}_{i,k-1}^r(\mathbf{z}_{i,k,1}^r)\rangle$$

$$\leq 4\mathbb{E}\|\mathbf{u}_{i,k-1}^r(\mathbf{z}_{i,k,1}^r) - \mathbf{u}_{i,0}^{r-\tau}(\mathbf{z}_{i,k,1}^r)\|^2 + \frac{1}{4}\mathbb{E}\|g(\bar{\mathbf{w}}^{r-\tau}, \mathbf{z}_{i,k,1}^r, \bar{\mathbf{w}}^{r-\tau}, \mathcal{S}_2) - \mathbf{u}_{i,0}^{r-\tau}(\mathbf{z}_{i,k,1}^r)\|^2$$

$$- \mathbb{E}\|g(\bar{\mathbf{w}}^{r-\tau}, \mathbf{z}_{i,k,1}^r, \bar{\mathbf{w}}^{r-\tau}, S_2) - \mathbf{u}_{i,0}^{r-\tau}(\mathbf{z}_{i,k,1}^r)\|^2$$

$$+ \frac{1}{4}\mathbb{E}\|g(\bar{\mathbf{w}}^{r-\tau}, \mathbf{z}_{i,k,1}^r, \bar{\mathbf{w}}^{r-\tau}, \mathcal{S}_2) - \mathbf{u}_{i,0}^{r-\tau}(\mathbf{z}_{i,k,1}^r)\|^2 + 4\mathbb{E}\|\mathbf{u}_{i,k-1}^r(\mathbf{z}_{i,k,1}^r) - \mathbf{u}_{i,0}^{r-\tau}(\mathbf{z}_{i,k,1}^r)\|^2.$$
(82)

Noting
$$- \mathbb{E}\|g(\bar{\mathbf{w}}^{r-\tau}, \mathbf{z}_{i,k,1}^r, \bar{\mathbf{w}}^{r-\tau}, \mathcal{S}_2) - \mathbf{u}_{i,0}^{r-\tau}(\mathbf{z}_{i,k,1}^r)\|^2$$

$$= -\mathbb{E}\|g(\bar{\mathbf{w}}^{r-\tau}, \mathbf{z}_{i,k,1}^r, \bar{\mathbf{w}}^{r-\tau}, \mathcal{S}_2) - \mathbf{u}_{i,k}^r(\mathbf{z}_{i,k,1}^r) + \mathbf{u}_{i,k}^r(\mathbf{z}_{i,k,1}^r) - \mathbf{u}_{i,0}^{r-\tau}(\mathbf{z}_{i,k,1}^r)\|^2$$

$$= -\mathbb{E}\|g(\bar{\mathbf{w}}^{r-\tau}, \mathbf{z}_{i,k,1}^r, \bar{\mathbf{w}}^{r-\tau}, \mathcal{S}_2) - \mathbf{u}_{i,k}^r(\mathbf{z}_{i,k,1}^r)\|^2 - \mathbb{E}\|\mathbf{u}_{i,k}^r(\mathbf{z}_{i,k,1}^r) - \mathbf{u}_{i,0}^{r-\tau}(\mathbf{z}_{i,k,1}^r)\|^2$$

$$+ 2\mathbb{E}\langle g(\bar{\mathbf{w}}^{r-\tau}, \mathbf{z}_{i,k,1}^r, \bar{\mathbf{w}}^{r-\tau}, \mathcal{S}_2) - \mathbf{u}_{i,k}^r(\mathbf{z}_{i,k,1}^r), \mathbf{u}_{i,k}^r(\mathbf{z}_{i,k,1}^r) - \mathbf{u}_{i,0}^{r-\tau}(\mathbf{z}_{i,k,1}^r)\rangle$$
(83)

$$\leq -\frac{1}{2}\mathbb{E}\|g(\bar{\mathbf{w}}^{r-\tau}, \mathbf{z}_{i,k,1}^r, \bar{\mathbf{w}}^{r-\tau}, \mathcal{S}_2) - \mathbf{u}_{i,k}^r(\mathbf{z}_{i,k,1}^r)\|^2 + 8\|\mathbf{u}_{i,k}^r(\mathbf{z}_{i,k,1}^r) - \mathbf{u}_{i,0}^{r-\tau}(\mathbf{z}_{i,k,1}^r)\|^2$$

$$\leq -\frac{1}{2}\mathbb{E}\|g(\bar{\mathbf{w}}^{r-\tau}, \mathbf{z}_{i,k,1}^r, \bar{\mathbf{w}}^{r-\tau}, \mathcal{S}_2) - \mathbf{u}_{i,k}^r(\mathbf{z}_{i,k,1}^r)\|^2 + 8\beta^2 K^2 \tau^2 C_0^2.$$

Then, we can obtain
$$\frac{\gamma+1}{2}\mathbb{E}\|\mathbf{u}_k^r - g(\bar{\mathbf{w}}_k^r)\|^2 \leq \frac{\gamma(1 - \frac{1}{|\mathcal{S}_1^i|}) + 1}{2}\mathbb{E}\|\mathbf{u}_{k-1}^r - g(\bar{\mathbf{w}}_k^r)\|^2 + \frac{\gamma^2\sigma^2}{|\mathcal{S}_1^i|} + \frac{8\beta^2 K^2 \tau^2 C_0^2}{|\mathcal{S}_1^i|}$$
$$+ \gamma^2\|\bar{\mathbf{w}}^r - \bar{\mathbf{w}}^{r-1}\|^2 + \gamma^2\|\bar{\mathbf{w}}^r - \mathbf{w}_{i,k}^r\|^2.$$
(84)

Dividing $\frac{\gamma+1}{2}$ on both sides gives
$$\mathbb{E}\|\mathbf{u}_k^r - g(\bar{\mathbf{w}}_{i,k}^r)\|^2 = (1 - \frac{\gamma}{4P_i})\mathbb{E}\|\mathbf{u}_{k-1}^r - g(\bar{\mathbf{w}}_{k-1}^r)\|^2 + \gamma^2\|\bar{\mathbf{w}}^r - \bar{\mathbf{w}}^{r-1}\|^2 + \gamma^2\|\bar{\mathbf{w}}^{r-1} - \mathbf{w}_k^r\|^2$$
$$+ \gamma\eta^2 K^2 \tau^2 C_0^2 + \frac{\gamma^2\sigma^2}{|\mathcal{S}_1^i|}.$$
(85)

Next, we deal with moving average of gradients, i.e., $G_{i,k}^r$. With update $G_{i,k}^r = (1 - \beta)G_{i,k-1}^r + \beta(G_{i,k,1}^r + G_{i,k,2}^r)$. we define $\bar{G}_k^r := \frac{1}{N}\sum_{i=1}^N G_{i,k}^r$, and $\Delta_k^r := \|\bar{G}_k^r - \nabla F(\bar{\mathbf{w}}_k^r)\|^2$. Then it follows that $\bar{G}_k^r = (1 - \beta)\bar{G}_{k-1}^r + \beta\frac{1}{N}\sum_i(G_{i,k,1}^r + G_{i,k,2}^r)$.

We get
$$\Delta_k^r = \|\bar{G}_k^r - \nabla F(\bar{\mathbf{w}}_k^r)\|^2$$

$$= \|(1-\beta)\bar{G}_{k-1}^r + \beta\frac{1}{N}\sum_i(G_{i,k,1}^r + G_{i,k,2}^r) - \nabla F(\bar{\mathbf{w}}_k^r)\|^2$$

$$= \Bigg\|(1-\beta)(\bar{G}_{k-1}^r - \nabla F(\bar{\mathbf{w}}_{k-1}^r)) + (1-\beta)(\nabla F(\bar{\mathbf{w}}_{k-1}^r) - \nabla F(\bar{\mathbf{w}}_k^r))$$

$$+ \beta\Bigg(\frac{1}{N}\sum_i(G_1(\mathbf{w}_{i,k}^r, \mathbf{z}_{i,k,1}^r, \mathbf{u}_{i,k}^r(\mathbf{z}_{i,k,1}^r), \mathbf{w}_{j,t}^{r-1}, \mathbf{z}_{j,t,2}^{r-1})$$

$$+ G_2(\mathbf{w}_{j',t'}^{r-\tau,r-1}, \mathbf{z}_{j',t',1}^{r-\tau,r-1}, \mathbf{u}_{j',t'}^{r-\tau,r-1}(\mathbf{z}_{j',k',1}^{r-1}), \mathbf{w}_{i,k}^r, \mathbf{z}_{i,k,2}^r))$$

$$- \frac{1}{N}\sum_i(G_1(\bar{\mathbf{w}}^{r-\tau}, \mathbf{z}_{i,k,1}^r, \mathbf{u}_{i,k}^r(\mathbf{z}_{i,k,1}^r), \mathbf{w}_{j,t}^{r-\tau,r-1}, \mathbf{z}_{j,t,2}^{r-1})$$

$$+ G_2(\bar{\mathbf{w}}^{r-\tau}, \mathbf{z}_{j',t',1}^{r-\tau,r-1}, \mathbf{u}_{j',t'}^{r-\tau,r-1}(\mathbf{z}_{j',t',1}^{r-1}), \mathbf{w}_{i,k}^r, \mathbf{z}_{i,k,2}^r))\Bigg)$$

$$+ \beta\Bigg(\frac{1}{N}\sum_i(G_1(\bar{\mathbf{w}}^{r-1}, \mathbf{z}_{i,k,1}^r, \mathbf{u}_{i,k}^r(\mathbf{z}_{i,k,1}^r), \bar{\mathbf{w}}^{r-1}, \mathbf{z}_{j,t,2}^{r-1})$$

$$+ G_2(\bar{\mathbf{w}}^{r-\tau}, \mathbf{z}_{j',t',1}^{r-1}, \mathbf{u}_{j',t'}^{r-\tau,r-1}(\mathbf{z}_{j',t',1}^{r-\tau,r-1}), \bar{\mathbf{w}}^{r-\tau}, \mathbf{z}_{i,k,2}^r))\Bigg)$$

$$- \frac{1}{N}\sum_i(G_1(\bar{\mathbf{w}}^{r-\tau}, \mathbf{z}_{i,k,1}^r, g(\bar{\mathbf{w}}^r, \mathbf{z}_{i,k,1}^r, \bar{\mathbf{w}}^r, \mathcal{S}_2), \bar{\mathbf{w}}^{r-\tau}, \mathbf{z}_{j,t,2}^{r-\tau,r-1})$$

$$+ G_2(\bar{\mathbf{w}}^{r-\tau}, \mathbf{z}_{j',t',1}^{r-\tau,r-1}, g(\bar{\mathbf{w}}^{r-\tau}, \mathbf{z}_{j',t',1}^{r-\tau,r-1}, \bar{\mathbf{w}}^{r-\tau}, \mathcal{S}_2), \bar{\mathbf{w}}^{r-\tau}, \mathbf{z}_{i,k,2}^r))\Bigg)$$

$$+ \beta\Bigg(\frac{1}{N}\sum_i(G_1(\bar{\mathbf{w}}^{r-\tau}, \mathbf{z}_{i,k,1}^r, g(\bar{\mathbf{w}}^r, \mathbf{z}_{i,k,1}^r, \bar{\mathbf{w}}^r, \mathcal{S}_2), \bar{\mathbf{w}}^{r-\tau}, \mathbf{z}_{j,t,2}^{r-\tau,r-1})$$

$$+ G_2(\bar{\mathbf{w}}^{r-\tau,r-1}, \mathbf{z}_{j',t',1}^{r-\tau,r-1}, g(\bar{\mathbf{w}}^{r-\tau}, \mathbf{z}_{j',t',1}^{r-\tau,r-1}, \bar{\mathbf{w}}^{r-\tau}, \mathcal{S}_2), \bar{\mathbf{w}}^{r-\tau}, \mathbf{z}_{i,k,2}^r))$$

$$- \frac{1}{N}\sum_i(G_1(\bar{\mathbf{w}}^{r-\tau}, \mathbf{z}_{i,k,1}^r, g(\bar{\mathbf{w}}^{r-\tau}, \mathbf{z}_{i,k,1}^r, \bar{\mathbf{w}}^{r-\tau}, \mathcal{S}_2), \bar{\mathbf{w}}^{r-\tau}, \mathbf{z}_{j,t,2}^{r-\tau,r-1})$$

$$+ G_2(\bar{\mathbf{w}}^{r-\tau}, \mathbf{z}_{j',t',1}^{r-\tau,r-1}, g(\bar{\mathbf{w}}^{r-\tau}, \mathbf{z}_{j',t',1}^{r-\tau,r-1}, \bar{\mathbf{w}}^{r-\tau}, \mathcal{S}_2), \bar{\mathbf{w}}^{r-\tau}, \mathbf{z}_{i,k,2}^r))\Bigg)$$

$$+ \beta\Bigg(\frac{1}{N}\sum_i(G_1(\bar{\mathbf{w}}^{r-\tau}, \mathbf{z}_{i,k,1}^r, g_{\mathbf{z}_{i,k,1}^r}(\bar{\mathbf{w}}^{r-\tau}), \bar{\mathbf{w}}^{r-\tau}, \mathbf{z}_{j,t,2}^{r-\tau,r-1})$$

$$+ G_2(\bar{\mathbf{w}}^{r-\tau}, \mathbf{z}_{j',t',1}^{r-\tau,r-1}, g(\bar{\mathbf{w}}^{r-\tau}, \mathbf{z}_{j',t',1}^{r-\tau,r-1}, \bar{\mathbf{w}}^{r-\tau}, \mathcal{S}_2), \bar{\mathbf{w}}^{r-\tau}, \mathbf{z}_{i,k,2}^r)) - \nabla F(\bar{\mathbf{w}}_k^r)\Bigg)\Bigg\|^2$$

$$\tag{86}$$

Using Young's inequality, we can then derive

$$
\begin{aligned}
\Delta_k^r \leq (1+\beta) \Big\| &(1-\beta)(\bar{G}_{k-1}^r - \nabla F(\bar{\mathbf{w}}_{k-1}^r)) \\
&+ \beta \Big( \frac{1}{N} \sum_i (G_1(\bar{\mathbf{w}}^{r-1}, \mathbf{z}_{i,k,1}^r, g(\bar{\mathbf{w}}^{r-\tau}, \mathbf{z}_{i,k,1}^r, \bar{\mathbf{w}}^{r-\tau}, \mathcal{S}_2), \bar{\mathbf{w}}^{r-\tau}, \mathbf{z}_{j,t,2}^{r-1}) \\
&+ G_2(\bar{\mathbf{w}}^{r-1}, \mathbf{z}_{j',t',1}^{r-\tau,r-1}, g(\bar{\mathbf{w}}^{r-\tau}, p_{j',t'}^{r-\tau}, \bar{\mathbf{w}}^{r-\tau}, \mathcal{S}_2), \bar{\mathbf{w}}^{r-\tau}, \mathbf{z}_{i,k,2}^r)) - \nabla F(\bar{\mathbf{w}}_k^r) \Big) \Big\|^2 \\
+ (1+\frac{1}{\beta}) \Big\| &\beta \Big( \frac{1}{N} \sum_i (G_1(\mathbf{w}_{i,k}^r, \mathbf{z}_{i,k,1}^r, \mathbf{u}_{i,k}^r(\mathbf{z}_{i,k,1}^r), \mathbf{w}_{j,t}^{r-\tau,r-1}, \mathbf{z}_{j,t,2}^{r-\tau,r-1}) \\
&\qquad + G_2(\mathbf{w}_{j',t'}^{r-\tau,r-1}, \mathbf{z}_{j',t',1}^{r-\tau,r-1}, \mathbf{u}_{j',t'}^{r-\tau,r-1}(\mathbf{z}_{j',t',1}^{r-\tau,r-1}), \mathbf{w}_{i,k}^r, \mathbf{z}_{i,k,2}^r)) \\
&\quad - \frac{1}{N} \sum_i (G_1(\bar{\mathbf{w}}^{r-\tau}, \mathbf{z}_{i,k,1}^r, \mathbf{u}_{i,k}^r(\mathbf{z}_{i,k,1}^r), \mathbf{w}_{j,t}^{r-\tau,r-1}, \mathbf{z}_{j,t,2}^{r-\tau,r-1}) \\
&\qquad + G_2(\bar{\mathbf{w}}^{r-\tau,r-1}, \mathbf{z}_{j',t',1}^{r-\tau,r-1}, \mathbf{u}_{j',t'}^{r-\tau}(\mathbf{z}_{j',t',1}^{r-\tau,r-1}), \mathbf{w}_{i,k}^r, \mathbf{z}_{i,k,2}^r)) \Big) \\
+ \beta \Big( \frac{1}{N} \sum_i &(G_1(\bar{\mathbf{w}}^{r-\tau}, \mathbf{z}_{i,k,1}^r, \mathbf{u}_{i,k}^r(\mathbf{z}_{i,k,1}^r), \bar{\mathbf{w}}^{r-\tau}, \mathbf{z}_{j,t,2}^{r-\tau,r-1}) + G_2(\bar{\mathbf{w}}^{r-\tau}, \mathbf{z}_{j',t',1}^{r-\tau,r-1}, \mathbf{u}_{j',t'}^{r-\tau}(\mathbf{z}_{j',t',1}^{r-1}), \bar{\mathbf{w}}^{r-\tau}, \mathbf{z}_{i,k,2}^r)) \Big) \\
&\quad - \frac{1}{N} \sum_i (G_1(\bar{\mathbf{w}}^{r-\tau}, \mathbf{z}_{i,k,1}^r, g(\bar{\mathbf{w}}^r, \mathbf{z}_{i,k,1}^r, \bar{\mathbf{w}}^r, \mathcal{S}_2), \bar{\mathbf{w}}^{r-\tau}, \mathbf{z}_{j,t,2}^{r-\tau,r-1}) \\
&\qquad + G_2(\bar{\mathbf{w}}^{r-\tau}, \mathbf{z}_{j',t',1}^{r-\tau,r-1}, g(\bar{\mathbf{w}}^{r-\tau}, \mathbf{z}_{j',t',1}^{r-\tau,r-1}, \bar{\mathbf{w}}^{r-\tau}, \mathcal{S}_2), \bar{\mathbf{w}}^{r-\tau}, \mathbf{z}_{i,k,2}^r)) \Big) \\
+ \beta \Big( \frac{1}{N} \sum_i &(G_1(\bar{\mathbf{w}}^{r-\tau}, \mathbf{z}_{i,k,1}^r, g(\bar{\mathbf{w}}^r, \mathbf{z}_{i,k,1}^r, \bar{\mathbf{w}}^r, \mathcal{S}_2), \bar{\mathbf{w}}^{r-\tau}, \mathbf{z}_{j,t,2}^{r-1}) \\
&\qquad + G_2(\bar{\mathbf{w}}^{r-1}, \mathbf{z}_{j',t',1}^{r-1}, g(\bar{\mathbf{w}}^{r-\tau}, \mathbf{z}_{j',t',1}^{r-\tau}, \bar{\mathbf{w}}^{r-\tau}, \mathcal{S}_2), \bar{\mathbf{w}}^{r-\tau}, \mathbf{z}_{i,k,2}^r)) \\
&\quad - \frac{1}{N} \sum_i (G_1(\bar{\mathbf{w}}^{r-\tau}, \mathbf{z}_{i,k,1}^r, g(\bar{\mathbf{w}}^{r-\tau}, \mathbf{z}_{i,k,1}^r, \bar{\mathbf{w}}^{r-\tau}, \mathcal{S}_2), \bar{\mathbf{w}}^{r-\tau}, \mathbf{z}_{j,t,2}^{r-\tau,r-1}) \\
&\qquad + G_2(\bar{\mathbf{w}}^{r-\tau}, p_{j',t'}^{r-\tau}, g(\bar{\mathbf{w}}^{r-\tau}, \mathbf{z}_{j',t',1}^{r-\tau,r-1}, \bar{\mathbf{w}}^{r-\tau}, \mathcal{S}_2), \bar{\mathbf{w}}^{r-\tau}, \mathbf{z}_{i,k,2}^r)) \Big\|^2 .
\end{aligned}
$$
(87)

By the fact that

$$
\begin{aligned}
\mathbb{E}[\frac{1}{N} \sum_i &(G_1(\bar{\mathbf{w}}^{r-\tau}, \mathbf{z}_{i,k,1}^r, g(\bar{\mathbf{w}}^{r-\tau}, \mathbf{z}_{i,k,1}^r, \bar{\mathbf{w}}^{r-\tau}, \mathcal{S}_2), \bar{\mathbf{w}}^{r-\tau}, \mathbf{z}_{j,t,2}^{r-1}) \\
&+ G_2(\bar{\mathbf{w}}^{r-\tau}, \mathbf{z}_{j',t',1}^{r-\tau,r-1}, g(\bar{\mathbf{w}}^{r-\tau}, \mathbf{z}_{j',t',1}^{r-\tau,r-1}, \bar{\mathbf{w}}^{r-\tau}, \mathcal{S}_2), \bar{\mathbf{w}}^{r-\tau}, \mathbf{z}_{i,k,2}^r)) - \nabla F(\bar{\mathbf{w}}^{r-\tau})] = 0,
\end{aligned}
$$
(88)

and

$$
\begin{aligned}
\mathbb{E}\| \frac{1}{N} \sum_i &(G_1(\bar{\mathbf{w}}^{r-\tau}, \mathbf{z}_{i,k,1}^r, g(\bar{\mathbf{w}}^{r-\tau}, \mathbf{z}_{i,k,1}^r, \bar{\mathbf{w}}^{r-\tau}, \mathcal{S}_2), \bar{\mathbf{w}}^{r-\tau}, \mathbf{z}_{j,t,2}^{r-\tau,r-1}) \\
&+ G_2(\bar{\mathbf{w}}^{r-1}, \mathbf{z}_{j',t',1}^{r-\tau,r-1}, g(\bar{\mathbf{w}}^{r-\tau}, \mathbf{z}_{j',t',1}^{r-1}, \bar{\mathbf{w}}^{r-\tau}, \mathcal{S}_2), \bar{\mathbf{w}}^{r-\tau}, \mathbf{z}_{i,k,2}^r)) - \nabla F(\bar{\mathbf{w}}_k^r)\|^2 \\
&\leq \frac{\sigma^2}{N}
\end{aligned}
$$
(89)

we obtain

$$
\begin{aligned}
\Delta_k^r \leq{}& (1+\beta)(1-\beta)^2 \|\bar{G}_{k-1}^r - \nabla F(\bar{\mathbf{w}}_{k-1}^r)\|^2 + 2\beta^2 \frac{\sigma^2}{N} \\
&+ 5\beta \|\bar{\mathbf{w}}^{r-\tau} - \mathbf{w}_{i,k}^r\|^2 + 5\beta \|\bar{\mathbf{w}}^{r-\tau} - \bar{\mathbf{w}}^r\|^2 + 5\beta \|\mathbf{u}_{i,k}^r(\mathbf{z}_{i,k,1}^r) - g(\bar{\mathbf{w}}^r, \mathbf{z}_{i,k,1}^r, , \bar{\mathbf{w}}^r, \mathcal{S}_2))\|^2 \\
\leq{}& (1-\beta) \|\bar{G}_{k-1}^r - \nabla F(\bar{\mathbf{w}}_{k-1}^r)\|^2 + 2\frac{\beta^2 \sigma^2}{N} + 5\beta \|\bar{\mathbf{w}}^r - \mathbf{w}_{i,k}^r\|^2 + 5\beta \|\bar{\mathbf{w}}^{r-\tau} - \bar{\mathbf{w}}^r\|^2 \\
&+ 5\beta \|\mathbf{u}_{i,k}^r(\mathbf{z}_{i,k,1}^r) - g_{\mathbf{z}_{i,k,1}^r}(\bar{\mathbf{w}}^r))\|^2 .
\end{aligned}
$$
(90)

$$\|\bar{\mathbf{w}}^{r-1} - \bar{\mathbf{w}}^r\|^2 = \tilde{\eta}^2 \|\frac{1}{NK}\sum_i\sum_k \bar{G}_k^r\|^2$$

$$\leq \tilde{\eta}^2 \frac{1}{K}\sum_k \|\bar{G}_k^r - \nabla F(\bar{\mathbf{w}}_k^r) + \nabla F(\bar{\mathbf{w}}_k^r)\|^2 \tag{91}$$

Then,

$$\frac{1}{K}\sum_k \Delta_k^r$$

$$\leq (1 - \frac{\beta}{2})\frac{1}{K}\sum_K \|\bar{G}_{k-1}^r - \nabla F(\bar{\mathbf{w}}_{k-1}^r)\|^2 + 4\frac{\beta^2\sigma^2}{N} + 5\beta\frac{1}{NK}\sum_i\sum_k \|\bar{\mathbf{w}}^{r-\tau} - \mathbf{w}_{i,k}^r\|^2 \tag{92}$$

$$+ 10\beta\|\mathbf{u}_{i,k}^r(\mathbf{z}_{i,k,1}^r) - g(\bar{\mathbf{w}}^r, \mathbf{z}_{i,k,1}^r, \bar{\mathbf{w}}^r, \mathcal{S}_2)\|^2.$$

Finally, we can analyze the convergence of the $\nabla F(\mathbf{w})$,

$$F(\bar{\mathbf{w}}^{r+1}) - F(\bar{\mathbf{w}}^r) \leq \nabla F(\bar{\mathbf{w}}^r)^\top (\bar{\mathbf{w}}^{r+1} - \bar{\mathbf{w}}^r) + \frac{L}{2}\|\bar{\mathbf{w}}^{r+1} - \bar{\mathbf{w}}^r\|^2$$

$$= -\tilde{\eta}\nabla F(\bar{\mathbf{w}}^r)^\top \left(\frac{1}{NK}\sum_i\sum_k G_{i,k}^r\right) + \frac{L}{2}\|\bar{\mathbf{w}}^{r+1} - \bar{\mathbf{w}}^r\|^2$$

$$= -\tilde{\eta}\nabla F(\bar{\mathbf{w}}^r)^\top \left(\frac{1}{NK}\sum_i\sum_k G_{i,k}^r - \nabla F(\bar{\mathbf{w}}^r) + \nabla F(\bar{\mathbf{w}}^r)\right) + \frac{L}{2}\|\bar{\mathbf{w}}^{r+1} - \bar{\mathbf{w}}^r\|^2$$

$$= -\tilde{\eta}\|\nabla F(\bar{\mathbf{w}}^r)\|^2 + \frac{\tilde{\eta}}{2}\|\nabla F(\bar{\mathbf{w}}^r)\|^2 + \frac{\tilde{\eta}}{2}\|\frac{1}{NK}\sum_i\sum_k G_{i,k}^r - \nabla F(\bar{\mathbf{w}}^r)\|^2 + \frac{L}{2}\|\bar{\mathbf{w}}^{r+1} - \bar{\mathbf{w}}^r\|^2$$

$$\leq -\frac{\tilde{\eta}}{2}\|\nabla F(\bar{\mathbf{w}}^r)\|^2 + \tilde{\eta}\|\frac{1}{NK}\sum_i\sum_k (G_{i,k}^r - \nabla F(\bar{\mathbf{w}}_k^r))\|^2 + \tilde{\eta}\|\frac{1}{K}\sum_k (\nabla F(\bar{\mathbf{w}}_k^r) - \nabla F(\bar{\mathbf{w}}^r))\|^2$$

$$+ \frac{L}{2}\|\bar{\mathbf{w}}^{r+1} - \bar{\mathbf{w}}^r\|^2$$

$$\leq -\frac{\tilde{\eta}}{2}\|\nabla F(\bar{\mathbf{w}}^r)\|^2 + \tilde{\eta}\|\frac{1}{NK}\sum_i\sum_k (G_{i,k}^r - \nabla F(\bar{\mathbf{w}}_k^r))\|^2 + \tilde{\eta}\frac{\tilde{L}^2}{NK}\sum_i\sum_k \|\mathbf{w}_{i,k}^r - \bar{\mathbf{w}}^r\|^2$$

$$+ \frac{L}{2}\|\bar{\mathbf{w}}^{r+1} - \bar{\mathbf{w}}^r\|^2. \tag{93}$$

Noting

$$\|\bar{\mathbf{w}}^r - \mathbf{w}_{i,k}^r\|^2 \leq \eta^2 K^2 C_\ell^2 C_g^2 \tag{94}$$

With similar technique to Appendix D except that we have an extra error term caused by the larger buffer, we obtain

$$\frac{1}{R}\sum_r \mathbb{E}\|\nabla F(\bar{\mathbf{w}}^r)\|^2$$

$$\leq O\left(\frac{M}{\gamma RK} + \gamma\beta^2 K^2 + \gamma + \eta^2\frac{M^2}{\gamma^2} + 8\gamma M\tilde{\eta}^2 + \gamma\eta^2 K^2\tau^2 C_0^2 + \frac{M}{\gamma}\tilde{\eta}^2\tau^2(\frac{1}{\beta RK} + \frac{\beta}{N})\right). \tag{95}$$

By setting parameters as in the theorem, we can conclude the proof. Further, to get $\frac{1}{R}\sum_r \mathbb{E}\|\nabla F(\bar{\mathbf{w}}^r)\|^2 \leq \epsilon^2$, we just need to set $\gamma = O(\epsilon^2)$, $\beta = O(\frac{\epsilon^2}{\sqrt{M}})$, $\tau = O(M^{1/4})$, $K = O(\frac{\sqrt{M}}{\epsilon})$, $\eta = O(\frac{\epsilon^2}{M})$, $R = O(\frac{\sqrt{M}}{\epsilon^3})$. $\qquad\square$

## G  FEDX WITH PARTIAL CLIENT PARTICIPATION

Considering that not all client machines are available to work at each round, in this section, we provide an algorithm that allows partial client participation in every round. The algorithm is given in Algorithm 3. We use the same assumption as in Appendix D. The convergence results will be presented in Theorem 7.

---

**Algorithm 3** FedX2: Federated Learning for CPR with non-linear $f$

---

1: On Client $i$: **Require** parameters $\eta, K$
2: Initialize model $\mathbf{w}_{i,0}^0, \mathcal{U}_i^0 = \{u^0(\mathbf{z}) = 0, \mathbf{z} \in \mathcal{S}_1^i\}, G_{i,0}^0 = 0$, and buffer $\mathcal{B}_{i,1}, \mathcal{B}_{i,2}, \mathcal{C}_i = \emptyset$
3: Send $\mathcal{H}_{i,1}^0, \mathcal{H}_{i,2}^0, \mathcal{U}_i^0$ to the server
4: Sample $K$ points from $S_1^i$, compute their predictions using model $\mathbf{w}_{i,0}^0$ denoted by $\mathcal{H}_{i,1}^0$
5: Sample $K$ points from $S_2^i$, compute their predictions using model $\mathbf{w}_{i,0}^0$ denoted by $\mathcal{H}_{i,2}^0$
6: **for** $r = 1, ..., R$ **do**
7:     if $i \notin P^r$ then skip this round, otherwise continue
8:     Receive $\mathcal{R}_{i,1}^{r-1}, \mathcal{R}_{i,2}^{r-1}, \mathcal{P}^{r-1}$ from the server
9:     Update the buffer $\mathcal{B}_{i,1}, \mathcal{B}_{i,2}, \mathcal{C}_i$ using $\mathcal{R}_{i,1}^{r-1}, \mathcal{R}_{i,2}^{r-1}, \mathcal{P}^{r-1}$ with shuffling, respectively
10:     Set $\mathcal{H}_{i,1}^r = \emptyset, \mathcal{H}_{i,2}^r = \emptyset, \mathcal{U}_i^r = \emptyset$
11:     **for** $k = 0, .., K - 1$ **do**
12:         Sample $\mathbf{z}_{i,k,1}^r$ from $\mathcal{S}_1^i$, sample $\mathbf{z}_{i,k,2}^r$ from $\mathcal{S}_2^i$       $\diamond$ or sample two mini-batches of data
13:         Take next $h_\xi^{r-1}, h_\zeta^{r-1}$ and $u_\zeta^{r-1}$ from $\mathcal{B}_{i,1}$ and $\mathcal{B}_{i,2}$ and $\mathcal{C}_i$, respectively
14:         Compute $h(\mathbf{w}_{i,k}^r, \mathbf{z}_{i,k,1}^r)$ and $h(\mathbf{w}_{i,k}^r, \mathbf{z}_{i,k,2}^r)$
15:         Compute $h(\mathbf{w}_{i,k}^r, \hat{\mathbf{z}}_{i,k,1}^r)$ and $h(\mathbf{w}_{i,k}^r, \hat{\mathbf{z}}_{i,k,2}^r)$ and add them to $\mathcal{H}_{i,1}^r, \mathcal{H}_{i,2}^r$, respectively
16:         Compute $\mathbf{u}_{i,k}^r(\mathbf{z}_{i,k,1}^r)$ according to (8 and add it to $\mathcal{U}_i^r$
17:         Compute $G_{i,k,1}^r$ and $G_{i,k,2}^r$ according to (9)
18:         $G_{i,k}^r = (1 - \beta)G_{i,k-1}^r + \beta(G_{i,k,1}^r + G_{i,k,2}^r)$
19:         $\mathbf{w}_{i,k+1}^r = \mathbf{w}_{i,k}^r - \eta G_{i,k}^r$
20:     **end for**
21:     Sends $\mathbf{w}_{i,K}^r, G_{i,k}^r$ to the server
22:     Send $\mathcal{H}_{i,1}^r, \mathcal{H}_{i,2}^r, \mathcal{U}_i^r$ to the server
23:     Receives $\bar{\mathbf{w}}^r, \bar{G}^r$ from the server and set $\mathbf{w}_{i,0}^{r+1} = \bar{\mathbf{w}}_r, G_{i,0}^{r+1} = \bar{G}^r$
24: **end for**

---

25: On Server
26: Collects $\mathcal{H}_*^0 = \mathcal{H}_{1,*}^0 \cup \mathcal{H}_{2,*}^0 \ldots \cup \mathcal{H}_{N,*}^0$ and $\mathcal{U}^0 = \mathcal{U}_1^0 \cup \mathcal{U}_1^0 \ldots \cup \mathcal{U}_N^0$, where $* = 1, 2$
27: **for** $r = 1, ..., R$ **do**
28:     Sample a set $P^r$ of clients to participant this round
29:     Broadcast $\bar{\mathbf{w}}^r$ and $G^r$ to clients in $P^r$
30:     Set $\mathcal{R}_{i,1}^{r-1} = \mathcal{H}_1^{r-1}, \mathcal{R}_{i,2}^{r-1} = \mathcal{H}_2^r, \mathcal{P}_i^{r-1} = \mathcal{U}^{r-1}$ and send them to Client $i$ for all $i \in P^r$
31:     Receive $\mathbf{w}_{i,K}^{r+1}, G_{i,K}^{r+1}$ from client $i \in P^r$, compute $\bar{\mathbf{w}}^{r+1} = \frac{1}{|P^r|}\sum_{i \in P^r} \mathbf{w}_{i,K}^{r+1}, G^{r+1} = \frac{1}{|P^r|}\sum_{i \in P^r} G_{i,K}^{r+1}$.
32:     Collects $\mathcal{H}_*^{r+1} = \cup \mathcal{H}_{i,*}^r, \forall i \in P^r$ and $\mathcal{U}^{r+1} = \cup \mathcal{U}_i^r, \forall i \in P_i$, where $* = 1, 2$
33: **end for**

---

## G.1 Analysis of the moving average estimator $\mathbf{u}$

**Lemma 3.** *Under Assumption 2, the moving average estimator $\mathbf{u}$ satisfies*

$$\frac{1}{N}\sum_{i=1}^{N}\frac{1}{|\mathcal{S}_1^i|}\sum_{\mathbf{z}\in|\mathcal{S}_1^i|}\mathbb{E}\|\mathbf{u}_{i,k}^r(\mathbf{z})-g(\bar{\mathbf{w}}_k^r,\mathbf{z},\bar{\mathbf{w}}_k^r,\mathcal{S}_2)\|^2$$

$$\leq(1-\frac{\gamma|P^r|}{16|\mathcal{S}_1^i|N})\frac{1}{N}\sum_{i=1}^{N}\frac{1}{|\mathcal{S}_1^i|}\sum_{\mathbf{z}\in|\mathcal{S}_1^i|}[\mathbb{E}\|\mathbf{u}_{i,k-1}^r(\mathbf{z})-g(\bar{\mathbf{w}}_{k-1}^r,\mathbf{z},\bar{\mathbf{w}}_{k-1}^r,\mathcal{S}_2)\|^2$$

$$+\frac{20|\mathcal{S}_1^i|N}{\gamma|P^r|}\tilde{L}^2\|\bar{\mathbf{w}}_{k-1}^r-\bar{\mathbf{w}}_k^r\|^2]+8\frac{\gamma^2}{|\mathcal{S}_1^i|}\frac{|P^r|}{N}(\sigma^2+C_0^2)+\frac{16\gamma\beta^2K^2C_0^2|P^r|}{|\mathcal{S}_1^i|N}$$

$$+8\frac{|P^r|}{N}\tilde{L}^2\|\bar{\mathbf{w}}^r-\bar{\mathbf{w}}^{r-1}\|^2+8\tilde{L}^2\frac{|P^r|}{N}\|\bar{\mathbf{w}}^r-\bar{\mathbf{w}}_k^r\|^2$$

$$+8(\gamma^2+\frac{\gamma}{|\mathcal{S}_1^i|})\tilde{L}^2\frac{1}{N}\sum_{i\in P^r}\|\bar{\mathbf{w}}^r-\mathbf{w}_{i,k}^r\|^2+2(\gamma^2+\frac{\gamma}{|\mathcal{S}_1^i|})\tilde{L}^2\frac{1}{NK}\sum_{i\in P^r}\sum_{k=1}^{K}\mathbb{E}\|\bar{\mathbf{w}}^{r-1}-\bar{\mathbf{w}}_{i,k}^{r-1}\|^2.$$

*Proof.* Denote $P^r$ as the clients that are sampled to take participation in the $r$-th round. By update rules of $\mathbf{u}$, we have

$$\mathbf{u}_{i,k}^r(\mathbf{z})=\begin{cases}\mathbf{u}_{i,k-1}^r(\mathbf{z})-\gamma(\mathbf{u}_{i,k-1}^r(\mathbf{z})-\ell(h(\mathbf{w}_{i,k}^r,\mathbf{z}_{i,k,1}^r),h(\mathbf{w}_{j,t}^{r-1},\hat{\mathbf{z}}_{j,t,2}^{r-1}))),&i\in P^r\text{ and }\mathbf{z}=\mathbf{z}_{i,k,1}^r\\\mathbf{u}_{i,k-1}^r(\mathbf{z}),&otherwise.\end{cases}$$
$$(96)$$

Or equivalently,

$$\mathbf{u}_{i,k}^r(\mathbf{z})=\begin{cases}\mathbf{u}_{i,k-1}^r(\mathbf{z})-\gamma(\mathbf{u}_{i,k-1}^r(\mathbf{z})-g(\mathbf{w}_{i,k}^r,\mathbf{z}_{i,k,1}^r,\mathbf{w}_{j,t}^{r-1},\hat{\mathbf{z}}_{j,t,2}^{r-1})),&i\in P^r\text{ and }\mathbf{z}=\mathbf{z}_{i,k,1}^r\\\mathbf{u}_{i,k-1}^r(\mathbf{z}),&otherwise.\end{cases}$$
$$(97)$$

Define $\bar{\mathbf{u}}_k^r=(\mathbf{u}_{1,k}^r,\mathbf{u}_{2,k}^r,...,\mathbf{u}_{N,k}^r)$, $\bar{\mathbf{w}}_k^r=\frac{1}{|P^r|}\sum_{i\in P^r}\mathbf{w}_{i,k}^r$. Then it follows that

$$\frac{1}{2N}\sum_{i=1}^{N}\frac{1}{|\mathcal{S}_1^i|}\sum_{\mathbf{z}\in|\mathcal{S}_1^i|}\mathbb{E}\|\mathbf{u}_{i,k}^r(\mathbf{z})-g(\bar{\mathbf{w}}_k^r,\mathbf{z},\bar{\mathbf{w}}_k^r,\mathcal{S}_2)\|^2$$

$$=\frac{1}{N}\sum_i\frac{1}{|\mathcal{S}_1^i|}\sum_{\mathbf{z}\in|\mathcal{S}_1^i|}\mathbb{E}\bigg[\frac{1}{2}\|\mathbf{u}_{i,k-1}^r(\mathbf{z})-g(\bar{\mathbf{w}}_k^r,\mathbf{z},\bar{\mathbf{w}}_k^r,\mathcal{S}_2)\|^2$$

$$+\langle\mathbf{u}_{i,k-1}^r(\mathbf{z})-g(\bar{\mathbf{w}}_k^r,\mathbf{z},\bar{\mathbf{w}}_k^r,\mathcal{S}_2),\mathbf{u}_{i,k}^r(\mathbf{z})-\mathbf{u}_{i,k-1}^r(\mathbf{z})\rangle+\frac{1}{2}\|\mathbf{u}_{i,k}^r(\mathbf{z})-\mathbf{u}_{i,k-1}^r(\mathbf{z})\|^2\bigg]$$

$$=\frac{1}{2N}\sum_i\frac{1}{|\mathcal{S}_i|}\sum_{\mathbf{z}\in\mathcal{S}_1^i}\mathbb{E}\|\mathbf{u}_{i,k-1}^r(\mathbf{z})-g(\bar{\mathbf{w}}_k^r,\mathbf{z},\bar{\mathbf{w}}_k^r,\mathcal{S}_2)\|^2$$

$$+\mathbb{E}\frac{1}{N}\sum_{i\in P^r}\frac{1}{|\mathcal{S}_1^i|}\langle\mathbf{u}_{i,k-1}^r(\mathbf{z}_{i,k,1}^r)-g(\bar{\mathbf{w}}_k^r,\mathbf{z}_{i,k,1}^r,\bar{\mathbf{w}}_k^r,\mathcal{S}_2),\mathbf{u}_{i,k}^r(\mathbf{z}_{i,k,1}^r)-\mathbf{u}_{i,k-1}^r(\mathbf{z}_{i,k,1}^r)\rangle$$

$$+\frac{1}{N}\sum_i\frac{1}{2|\mathcal{S}_1^i|}\mathbb{E}\|\mathbf{u}_{i,k}^r(\mathbf{z}_{i,k,1}^r)-\mathbf{u}_{i,k-1}^r(\mathbf{z}_{i,k,1}^r)\|^2$$

$$=\frac{1}{2N}\sum_i\frac{1}{|\mathcal{S}_i|}\sum_{\mathbf{z}\in\mathcal{S}_1^i}\mathbb{E}\|\mathbf{u}_{i,k-1}^r(\mathbf{z})-g(\bar{\mathbf{w}}_k^r,\mathbf{z},\bar{\mathbf{w}}_k^r,\mathcal{S}_2)\|^2$$

$$+\mathbb{E}[\frac{1}{N}\sum_{i\in P^r}\frac{1}{|\mathcal{S}_1^i|}\langle\mathbf{u}_{i,k-1}^r(\mathbf{z}_{i,k,1}^r)-g(\mathbf{w}_{i,k}^r,\mathbf{z}_{i,k,1}^r,\mathbf{w}_{j,t}^{r-1},\hat{\mathbf{z}}_{j,t,2}^{r-1}),\mathbf{u}_{i,k}^r(\mathbf{z}_{i,k,1}^r)-\mathbf{u}_{i,k-1}^r(\mathbf{z}_{i,k,1}^r)\rangle]$$

$$+\mathbb{E}[\frac{1}{N}\sum_{i\in P^r}\frac{1}{|\mathcal{S}_1^i|}\langle g(\mathbf{w}_{i,k}^r,\mathbf{z}_{i,k,1}^r,\mathbf{w}_{j,t}^{r-1},\hat{\mathbf{z}}_{j,t,2}^{r-1})-g(\bar{\mathbf{w}}_k^r,\mathbf{z}_{i,k,1}^r,\bar{\mathbf{w}}_k^r,\mathcal{S}_2),\mathbf{u}_{i,k}^r(\mathbf{z}_{i,k,1}^r)-\mathbf{u}_{i,k-1}^r(\mathbf{z}_{i,k,1}^r)\rangle]$$

$$+\mathbb{E}[\frac{1}{N}\sum_{i\in P^r}\frac{1}{2|\mathcal{S}_i|}\|\mathbf{u}_{i,k}^r(\mathbf{z}_{i,k,1}^r)-\mathbf{u}_{i,k-1}^r(\mathbf{z}_{i,k,1}^r)\|^2],$$

$$(98)$$

where for $i \in P^r$ it has

$$\langle \mathbf{u}_{i,k-1}^r(\mathbf{z}_{i,k,1}^r) - g(\mathbf{w}_{i,k}^r, \mathbf{z}_{i,k,1}^r, \mathbf{w}_{j,t}^{r-1}, \hat{\mathbf{z}}_{j,t,2}^{r-1}), \mathbf{u}_{i,k}^r(\mathbf{z}_{i,k,1}^r) - \mathbf{u}_{i,k-1}^r(\mathbf{z}_{i,k,1}^r)\rangle$$

$$= \langle \mathbf{u}_{i,k-1}^r(\mathbf{z}_{i,k,1}^r) - g(\mathbf{w}_{i,k}^r, \mathbf{z}_{i,k,1}^r, \mathbf{w}_{j,t}^{r-1}, \hat{\mathbf{z}}_{j,t,2}^{r-1}), g(\bar{\mathbf{w}}_k^r, \mathbf{z}_{i,k,1}^r, \bar{\mathbf{w}}_k^r, \mathcal{S}_2) - \mathbf{u}_{i,k-1}^r(\mathbf{z}_{i,k,1}^r)\rangle$$

$$\quad + \langle \mathbf{u}_{i,k-1}^r(\mathbf{z}_{i,k,1}^r) - g(\mathbf{w}_{i,k}^r, \mathbf{z}_{i,k,1}^r, \mathbf{w}_{j,t}^{r-1}, \hat{\mathbf{z}}_{j,t,2}^{r-1}), \mathbf{u}_{i,k}^r(\mathbf{z}_{i,k,1}^r) - g(\bar{\mathbf{w}}_k^r, \mathbf{z}_{i,k,1}^r, \bar{\mathbf{w}}_k^r, \mathcal{S}_2)\rangle$$

$$= \langle \mathbf{u}_{i,k-1}^r(\mathbf{z}_{i,k,1}^r) - g(\mathbf{w}_{i,k}^r, \mathbf{z}_{i,k,1}^r, \mathbf{w}_{j,t}^{r-1}, \hat{\mathbf{z}}_{j,t,2}^{r-1}), g(\bar{\mathbf{w}}_k^r, \mathbf{z}_{i,k,1}^r, \bar{\mathbf{w}}_k^r, \mathcal{S}_2) - \mathbf{u}_{i,k-1}^r(\mathbf{z}_{i,k,1}^r)\rangle$$

$$\quad + \frac{1}{\gamma}\langle \mathbf{u}_{i,k-1}^r(\mathbf{z}_{i,k,1}^r) - \mathbf{u}_{i,k}^r(\mathbf{z}_{i,k,1}^r), \mathbf{u}_{i,k}^r(\mathbf{z}_{i,k,1}^r) - g(\bar{\mathbf{w}}_k^r, \mathbf{z}_{i,k,1}^r, \bar{\mathbf{w}}_k^r, \mathcal{S}_2)\rangle \tag{99}$$

$$= \langle \mathbf{u}_{i,k-1}^r(\mathbf{z}_{i,k,1}^r) - g(\mathbf{w}_{i,k}^r, \mathbf{z}_{i,k,1}^r, \mathbf{w}_{j,t}^{r-1}, \hat{\mathbf{z}}_{j,t,2}^{r-1}), g(\bar{\mathbf{w}}_k^r, \mathbf{z}_{i,k,1}^r, \bar{\mathbf{w}}_k^r, \mathcal{S}_2) - \mathbf{u}_{i,k-1}^r(\mathbf{z}_{i,k,1}^r)\rangle$$

$$\quad + \frac{1}{2\gamma}(\|\mathbf{u}_{i,k-1}^r(\mathbf{z}_{i,k,1}^r) - g(\bar{\mathbf{w}}_k^r, \mathbf{z}_{i,k,1}^r, \bar{\mathbf{w}}_k^r, \mathcal{S}_2)\|^2 - \|\mathbf{u}_{i,k}^r(\mathbf{z}_{i,k,1}^r) - \mathbf{u}_{i,k-1}^r(\mathbf{z}_{i,k,1}^r)\|^2$$

$$\quad - \|\mathbf{u}_{i,k}^r(\mathbf{z}_{i,k,1}^r) - g(\bar{\mathbf{w}}_k^r, \mathbf{z}_{i,k,1}^r, \bar{\mathbf{w}}_k^r, \mathcal{S}_2)\|^2)$$

If $\gamma \leq \frac{1}{5}$, we have for $i \in P^r$

$$-\frac{1}{2}\left(\frac{1}{\gamma} - 1 - \frac{\gamma+1}{4\gamma}\right)\mathbb{E}\|\mathbf{u}_{i,k}^r(\mathbf{z}_{i,k,1}^r) - \mathbf{u}_{i,k-1}^r(\mathbf{z}_{i,k,1}^r)\|^2$$

$$\quad + \mathbb{E}\langle g(\mathbf{w}_{i,k}^r, \mathbf{z}_{i,k,1}^r, \mathbf{w}_{j,t}^{r-1}, \hat{\mathbf{z}}_{j,t,2}^{r-1}) - g(\bar{\mathbf{w}}_k^r, \mathbf{z}_{i,k,1}^r, \bar{\mathbf{w}}_k^r, \mathcal{S}_2), \mathbf{u}_{i,k}^r(\mathbf{z}_{i,k,1}^r) - \mathbf{u}_{i,k-1}^r(\mathbf{z}_{i,k,1}^r)\rangle$$

$$\leq -\frac{1}{4\gamma}\mathbb{E}\|\mathbf{u}_{i,k}^r(\mathbf{z}_{i,k,1}^r) - \mathbf{u}_{i,k-1}^r(\mathbf{z}_{i,k,1}^r)\|^2 + \gamma\mathbb{E}\|g(\mathbf{w}_{i,k}^r, \mathbf{z}_{i,k,1}^r, \mathbf{w}_{j,t}^{r-1}, \hat{\mathbf{z}}_{j,t,2}^{r-1}) - g(\bar{\mathbf{w}}_k^r, \mathbf{z}_{i,k,1}^r, \bar{\mathbf{w}}_k^r, \mathcal{S}_2)\|^2$$

$$\quad + \frac{1}{4\gamma}\mathbb{E}\|\mathbf{u}_{i,k}^r(\mathbf{z}_{i,k,1}^r) - \mathbf{u}_{i,k-1}^r(\mathbf{z}_{i,k,1}^r)\|^2$$

$$\leq \gamma\mathbb{E}\|g(\mathbf{w}_{i,k}^r, \mathbf{z}_{i,k,1}^r, \mathbf{w}_{j,t}^{r-1}, \hat{\mathbf{z}}_{j,t,2}^{r-1}) - g(\bar{\mathbf{w}}_k^r, \mathbf{z}_{i,k,1}^r, \bar{\mathbf{w}}_k^r, \mathcal{S}_2)\|^2$$

$$\leq 4\gamma\mathbb{E}\|g(\bar{\mathbf{w}}^{r-1}, \mathbf{z}_{i,k,1}^r, \bar{\mathbf{w}}^{r-1}, \hat{\mathbf{z}}_{j,t,2}^{r-1}) - g(\bar{\mathbf{w}}^{r-1}, \mathbf{z}_{i,k,1}^r, \bar{\mathbf{w}}^{r-1}, \mathcal{S}_2)\|^2 + 4\gamma\tilde{L}^2\mathbb{E}\|\bar{\mathbf{w}}^r - \bar{\mathbf{w}}^{r-1}\|^2$$

$$\quad + 4\gamma\tilde{L}^2\mathbb{E}\|\mathbf{w}_{i,k}^r - \bar{\mathbf{w}}^r\|^2 + 4\gamma\tilde{L}^2\mathbb{E}\|\mathbf{w}_{j,t}^{r-1} - \bar{\mathbf{w}}^{r-1}\|^2$$

$$\leq 4\gamma\sigma^2 + 4\gamma\tilde{L}^2\mathbb{E}\|\bar{\mathbf{w}}^r - \bar{\mathbf{w}}^{r-1}\|^2 + 4\gamma\tilde{L}^2\mathbb{E}\|\mathbf{w}_{i,k}^r - \bar{\mathbf{w}}^r\|^2 + 4\gamma\tilde{L}^2\mathbb{E}\|\mathbf{w}_{j,t}^{r-1} - \bar{\mathbf{w}}^{r-1}\|^2. \tag{100}$$

Then, we have

$$\frac{1}{2N}\sum_{i=1}^N \frac{1}{|\mathcal{S}_1^i|}\sum_{\mathbf{z}\in|\mathcal{S}_1^i|}\mathbb{E}\|\mathbf{u}_{i,k}^r(\mathbf{z}) - g(\bar{\mathbf{w}}_k^r, \mathbf{z}, \bar{\mathbf{w}}_k^r, \mathcal{S}_2)\|^2$$

$$\leq \frac{1}{2N}\sum_{i=1}^N \frac{1}{|\mathcal{S}_1^i|}\sum_{\mathbf{z}\in|\mathcal{S}_1^i|}\mathbb{E}\|\mathbf{u}_{i,k-1}^r(\mathbf{z}) - g(\bar{\mathbf{w}}_k^r, \mathbf{z}, \bar{\mathbf{w}}_k^r, \mathcal{S}_2)\|^2$$

$$+ \frac{1}{N}\sum_{i\in P^r}\frac{1}{|\mathcal{S}_1^i|}\left[\frac{1}{2\gamma}\mathbb{E}\|\mathbf{u}_{i,k-1}^r(\mathbf{z}_{i,k,1}^r) - g(\bar{\mathbf{w}}_k^r, \mathbf{z}_{i,k,1}^r, \bar{\mathbf{w}}_k^r, \mathcal{S}_2)\|^2\right.$$

$$- \frac{1}{2\gamma}\mathbb{E}\|\mathbf{u}_{i,k}^r(\mathbf{z}_{i,k,1}^r) - g(\bar{\mathbf{w}}_k^r, \mathbf{z}_{i,k,1}^r, \bar{\mathbf{w}}_k^r, \mathcal{S}_2)\|^2 - \frac{\gamma+1}{8\gamma}\|\mathbf{u}_{i,k}^r(\mathbf{z}_{i,k,1}^r) - \mathbf{u}_{i,k-1}^r(\mathbf{z}_{i,k,1}^r)\|^2 + 4\gamma\sigma^2$$

$$+ 4\gamma\tilde{L}^2\mathbb{E}\|\bar{\mathbf{w}}^r - \bar{\mathbf{w}}^{r-1}\|^2 + 4\gamma\tilde{L}^2\mathbb{E}\|\mathbf{w}_{i,k}^r - \bar{\mathbf{w}}^r\|^2 + 4\gamma\tilde{L}^2\mathbb{E}\|\mathbf{w}_{j,t}^{r-1} - \bar{\mathbf{w}}^{r-1}\|^2$$

$$+ \mathbb{E}\langle \mathbf{u}_{i,k-1}^r(\mathbf{z}_{i,k,1}^r) - g(\mathbf{w}_{i,k}^r, \mathbf{z}_{i,k,1}^r, \mathbf{w}_{j,t}^{r-1}, \hat{\mathbf{z}}_{j,t,2}^{r-1}), g(\bar{\mathbf{w}}_k^r, \mathbf{z}_{i,k,1}^r, \bar{\mathbf{w}}_k^r, \mathcal{S}_2) - \mathbf{u}_{i,k-1}^r(\mathbf{z}_{i,k,1}^r)\rangle\Bigg].$$

$$\tag{101}$$

Note that for $i \in P^r$, $\sum_{\mathbf{z} \neq \mathbf{z}_{i,k,1}^r} \|\mathbf{u}_{i,k-1}^r(\mathbf{z}) - g(\bar{\mathbf{w}}_k^r, \mathbf{z}, \bar{\mathbf{w}}_k^r, \mathcal{S}_2)\|^2 = \sum_{\mathbf{z} \neq \mathbf{z}_{i,k,1}^r} \|\mathbf{u}_{i,k}^r(\mathbf{z}) - g(\mathbf{w}_k^r, \mathbf{z}, \bar{\mathbf{w}}_k^r, \mathcal{S}_2)\|^2$, which implies for $i \in P^r$

$$\frac{1}{2\gamma} \left( \|\mathbf{u}_{i,k-1}^r(\mathbf{z}_{i,k,1}^r) - g(\bar{\mathbf{w}}_k^r, \mathbf{z}_{i,k,1}^r, \bar{\mathbf{w}}_k^r, \mathcal{S}_2)\|^2 - \|\mathbf{u}_{i,k}^r(\mathbf{z}_{i,k,1}^r) - g(\bar{\mathbf{w}}_k^r, \mathbf{z}_{i,k,1}^r, \bar{\mathbf{w}}_k^r, \mathcal{S}_2)\|^2 \right)$$
$$= \frac{1}{2\gamma} \sum_{\mathbf{z} \in \mathcal{S}_1^i} \left( \|\mathbf{u}_{i,k-1}^r(\mathbf{z}) - g(\bar{\mathbf{w}}_k^r, \mathbf{z}, \bar{\mathbf{w}}_k^r, \mathcal{S}_2)\|^2 - \|\mathbf{u}_{i,k}^r(\mathbf{z}) - g(\bar{\mathbf{w}}_k^r, \mathbf{z}, \bar{\mathbf{w}}_k^r, \mathcal{S}_2)\|^2 \right). \tag{102}$$

Since $\ell(\cdot) \leq C_0$, we have that $\|g(\cdot)\|^2 \leq C_0^2$, $\|\mathbf{u}_{i,k}^r(\mathbf{z})\|^2 \leq C_0^2$ and
$$\|\mathbf{u}_{i,k}^r(\mathbf{z}) - \mathbf{u}_{i,0}^r(\mathbf{z})\|^2 \leq \beta^2 K^2 C_0^2.$$
Besides, we have for $i \in P^r$ that

$$\mathbb{E}\langle \mathbf{u}_{i,k-1}^r(\mathbf{z}_{i,k,1}^r) - g(\mathbf{w}_{i,k}^r, \mathbf{z}_{i,k,1}^r, \mathbf{w}_{j,t}^{r-1}, \hat{\mathbf{z}}_{j,t,2}^{r-1}), g(\bar{\mathbf{w}}_k^r, \mathbf{z}_{i,k,1}^r, \bar{\mathbf{w}}_k^r, \mathcal{S}_2) - \mathbf{u}_{i,k-1}^r(\mathbf{z}_{i,k,1}^r)\rangle$$
$$= \mathbb{E}\langle \mathbf{u}_{i,k-1}^r(\mathbf{z}_{i,k,1}^r) - g(\bar{\mathbf{w}}^{r-1}, \mathbf{z}_{i,k,1}^r, \bar{\mathbf{w}}^{r-1}, \hat{\mathbf{z}}_{j,t,2}^{r-1}), g(\bar{\mathbf{w}}_k^r, \mathbf{z}_{i,k,1}^r, \bar{\mathbf{w}}_k^r, \mathcal{S}_2) - \mathbf{u}_{i,k-1}^r(\mathbf{z}_{i,k,1}^r)\rangle$$
$$+ \mathbb{E}\langle g(\bar{\mathbf{w}}^{r-1}, \mathbf{z}_{i,k,1}^r, \bar{\mathbf{w}}^{r-1}, \hat{\mathbf{z}}_{j,t,2}^{r-1}) - g(\mathbf{w}_{i,k}^r, \mathbf{z}_{i,k,1}^r, \mathbf{w}_{j,t}^{r-1}, \hat{\mathbf{z}}_{j,t,2}^{r-1}), g(\bar{\mathbf{w}}_k^r, \mathbf{z}_{i,k,1}^r, \bar{\mathbf{w}}_k^r, \mathcal{S}_2) - \mathbf{u}_{i,k-1}^r(\mathbf{z}_{i,k,1}^r)\rangle$$
$$\leq \mathbb{E}\langle \mathbf{u}_{i,k-1}^r(\mathbf{z}_{i,k,1}^r) - g(\bar{\mathbf{w}}^{r-1}, \mathbf{z}_{i,k,1}^r, \bar{\mathbf{w}}^{r-1}, \hat{\mathbf{z}}_{j,t,2}^{r-1}), g(\bar{\mathbf{w}}_k^r, \mathbf{z}_{i,k,1}^r, \bar{\mathbf{w}}_k^r, \mathcal{S}_2) - g(\bar{\mathbf{w}}^{r-1}, \mathbf{z}_{i,k,1}^r, \bar{\mathbf{w}}^{r-1}, \mathcal{S}_2)\rangle$$
$$+ \mathbb{E}\langle \mathbf{u}_{i,k-1}^r(\mathbf{z}_{i,k,1}^r) - g(\bar{\mathbf{w}}^{r-1}, \mathbf{z}_{i,k,1}^r, \bar{\mathbf{w}}^{r-1}, \hat{\mathbf{z}}_{j,t,2}^{r-1}), g(\bar{\mathbf{w}}^{r-1}, \mathbf{z}_{i,k,1}^r, \bar{\mathbf{w}}^{r-1}, \mathcal{S}_2) - \mathbf{u}_{i,k-1}^r(\mathbf{z}_{i,k,1}^r)\rangle$$
$$+ 2\tilde{L}^2 \mathbb{E}\|\bar{\mathbf{w}}^r - \bar{\mathbf{w}}^{r-1}\|^2 + 2\tilde{L}^2 \mathbb{E}\|\bar{\mathbf{w}}^r - \mathbf{w}_{i,k}^r\|^2 + \tilde{L}^2 \mathbb{E}\|\bar{\mathbf{w}}^{r-1} - \mathbf{w}_{j,t}^{r-1}\|^2$$
$$+ \frac{1}{4} \mathbb{E}\|g(\bar{\mathbf{w}}^r, \mathbf{z}_{i,k,1}^r, \bar{\mathbf{w}}_k^r, \mathcal{S}_2) - \mathbf{u}_{i,k-1}^r(\mathbf{z}_{i,k,1}^r)\|^2$$
$$\leq 2\gamma C_0^2 + \frac{1}{\gamma}\|\bar{\mathbf{w}}_k^r - \bar{\mathbf{w}}^{r-1}\|^2$$
$$+ \mathbb{E}\langle \mathbf{u}_{i,k-1}^r(\mathbf{z}_{i,k,1}^r) - g(\bar{\mathbf{w}}^{r-1}, \mathbf{z}_{i,k,1}^r, \bar{\mathbf{w}}^{r-1}, \hat{\mathbf{z}}_{j,t,2}^{r-1}), g(\bar{\mathbf{w}}^{r-1}, \mathbf{z}_{i,k,1}^r, \bar{\mathbf{w}}^{r-1}, \mathcal{S}_2) - \mathbf{u}_{i,k-1}^r(\mathbf{z}_{i,k,1}^r)\rangle$$
$$+ 2\tilde{L}^2 \mathbb{E}\|\bar{\mathbf{w}}^r - \bar{\mathbf{w}}^{r-1}\|^2 + 2\tilde{L}^2 \mathbb{E}\|\bar{\mathbf{w}}^r - \mathbf{w}_{i,k}^r\|^2 + \tilde{L}^2 \mathbb{E}\|\bar{\mathbf{w}}^{r-1} - \mathbf{w}_{j,t}^{r-1}\|^2$$
$$+ \frac{1}{4} \mathbb{E}\|g(\bar{\mathbf{w}}^r, \mathbf{z}_{i,k,1}^r, \bar{\mathbf{w}}_k^r, \mathcal{S}_2) - \mathbf{u}_{i,k-1}^r(\mathbf{z}_{i,k,1}^r)\|^2, \tag{103}$$

where
$$\mathbb{E}\langle \mathbf{u}_{i,k-1}^r(\mathbf{z}_{i,k,1}^r) - g(\bar{\mathbf{w}}^{r-1}, \mathbf{z}_{i,k,1}^r, \bar{\mathbf{w}}^{r-1}, \hat{\mathbf{z}}_{j,t,2}^{r-1}), g(\bar{\mathbf{w}}^{r-1}, \mathbf{z}_{i,k,1}^r, \bar{\mathbf{w}}^{r-1}, \mathcal{S}_2) - \mathbf{u}_{i,k-1}^r(\mathbf{z}_{i,k,1}^r)\rangle$$
$$= \mathbb{E}\langle \mathbf{u}_{i,k-1}^r(\mathbf{z}_{i,k,1}^r) - \mathbf{u}_{i,0}^{r-1}(\mathbf{z}_{i,k,1}^r) + \mathbf{u}_{i,0}^{r-1}(\mathbf{z}_{i,k,1}^r) - g(\bar{\mathbf{w}}^{r-1}, \mathbf{z}_{i,k,1}^r, \bar{\mathbf{w}}^{r-1}, \hat{\mathbf{z}}_{j,t,2}^{r-1}),$$
$$g(\bar{\mathbf{w}}^{r-1}, \mathbf{z}_{i,k,1}^r, \bar{\mathbf{w}}^{r-1}, \mathcal{S}_2) - \mathbf{u}_{i,0}^{r-1}(\mathbf{z}_{i,k,1}^r) + \mathbf{u}_{i,0}^{r-1}(\mathbf{z}_{i,k,1}^r) - \mathbf{u}_{i,k-1}^r(\mathbf{z}_{i,k,1}^r)\rangle$$
$$\leq \mathbb{E}\langle \mathbf{u}_{i,k-1}^r(\mathbf{z}_{i,k,1}^r) - \mathbf{u}_{i,0}^{r-1}(\mathbf{z}_{i,k,1}^r), g(\bar{\mathbf{w}}^{r-1}, \mathbf{z}_{i,k,1}^r, \bar{\mathbf{w}}^{r-1}, \mathcal{S}_2) - \mathbf{u}_{i,0}^{r-1}(\mathbf{z}_{i,k,1}^r)\rangle$$
$$+ \mathbb{E}\langle \mathbf{u}_{i,k-1}^r(\mathbf{z}_{i,k,1}^r) - \mathbf{u}_{i,0}^{r-1}(\mathbf{z}_{i,k,1}^r), \mathbf{u}_{i,0}^{r-1}(\mathbf{z}_{i,k,1}^r) - \mathbf{u}_{i,k-1}^r(\mathbf{z}_{i,k,1}^r)\rangle$$
$$+ \mathbb{E}\langle \mathbf{u}_{i,0}^{r-1}(\mathbf{z}_{i,k,1}^r) - g(\bar{\mathbf{w}}^{r-1}, \mathbf{z}_{i,k,1}^r, \bar{\mathbf{w}}^{r-1}, \hat{\mathbf{z}}_{j,t,2}^{r-1}), g(\bar{\mathbf{w}}^{r-1}, \mathbf{z}_{i,k,1}^r, \bar{\mathbf{w}}^{r-1}, \mathcal{S}_2) - \mathbf{u}_{i,0}^{r-1}(\mathbf{z}_{i,k,1}^r)\rangle$$
$$+ \mathbb{E}\langle \mathbf{u}_{i,0}^{r-1}(\mathbf{z}_{i,k,1}^r) - g(\bar{\mathbf{w}}^{r-1}, \mathbf{z}_{i,k,1}^r, \bar{\mathbf{w}}^{r-1}, \hat{\mathbf{z}}_{j,t,2}^{r-1}), \mathbf{u}_{i,0}^{r-1}(\mathbf{z}_{i,k,1}^r) - \mathbf{u}_{i,k-1}^r(\mathbf{z}_{i,k,1}^r)\rangle$$
$$\leq 4\mathbb{E}\|\mathbf{u}_{i,k-1}^r(\mathbf{z}_{i,k,1}^r) - \mathbf{u}_{i,0}^{r-1}(\mathbf{z}_{i,k,1}^r)\|^2 + \frac{1}{4}\mathbb{E}\|g(\bar{\mathbf{w}}^{r-1}, \mathbf{z}_{i,k,1}^r, \bar{\mathbf{w}}^{r-1}, \mathcal{S}_2) - \mathbf{u}_{i,0}^{r-1}(\mathbf{z}_{i,k,1}^r)\|^2$$
$$- \mathbb{E}\|g(\bar{\mathbf{w}}^{r-1}, \mathbf{z}_{i,k,1}^r, \bar{\mathbf{w}}^{r-1}, \mathcal{S}_2) - \mathbf{u}_{i,0}^{r-1}(\mathbf{z}_{i,k,1}^r)\|^2$$
$$+ \frac{1}{4}\mathbb{E}\|g(\bar{\mathbf{w}}^{r-1}, \mathbf{z}_{i,k,1}^r, \bar{\mathbf{w}}^{r-1}, \mathcal{S}_2) - \mathbf{u}_{i,0}^{r-1}(\mathbf{z}_{i,k,1}^r)\|^2 + 4\mathbb{E}\|\mathbf{u}_{i,k-1}^r(\mathbf{z}_{i,k,1}^r) - \mathbf{u}_{i,0}^{r-1}(\mathbf{z}_{i,k,1}^r)\|^2$$
$$\leq 4\mathbb{E}\|\mathbf{u}_{i,k-1}^r(\mathbf{z}_{i,k,1}^r) - \mathbf{u}_{i,0}^{r-1}(\mathbf{z}_{i,k,1}^r)\|^2 - \frac{1}{2}\mathbb{E}\|g(\bar{\mathbf{w}}^{r-1}, \mathbf{z}_{i,k,1}^r, \bar{\mathbf{w}}^{r-1}, \mathcal{S}_2) - \mathbf{u}_{i,0}^{r-1}(\mathbf{z}_{i,k,1}^r)\|^2$$
$$+ 8\beta^2 K^2 C_0^2. \tag{104}$$

Noting for $i \in P^r$,

$$-\mathbb{E}\|g(\bar{\mathbf{w}}^{r-1}, \mathbf{z}^r_{i,k,1}, \bar{\mathbf{w}}^{r-1}, \mathcal{S}_2) - \mathbf{u}^{r-1}_{i,0}(\mathbf{z}^r_{i,k,1})\|^2$$

$$= -\mathbb{E}\|g(\bar{\mathbf{w}}^{r-1}, \mathbf{z}^r_{i,k,1}, \bar{\mathbf{w}}^{r-1}, \mathcal{S}_2) - \mathbf{u}^r_{i,k-1}(\mathbf{z}^r_{i,k,1}) + \mathbf{u}^r_{i,k-1}(\mathbf{z}^r_{i,k,1}) - \mathbf{u}^{r-1}_{i,0}(\mathbf{z}^r_{i,k,1})\|^2$$

$$= -\mathbb{E}\|g(\bar{\mathbf{w}}^{r-1}, \mathbf{z}^r_{i,k,1}, \bar{\mathbf{w}}^{r-1}, \mathcal{S}_2) - \mathbf{u}^r_{i,k-1}(\mathbf{z}^r_{i,k,1})\|^2 - \mathbb{E}\|\mathbf{u}^r_{i,k-1}(\mathbf{z}^r_{i,k,1}) - \mathbf{u}^{r-1}_{i,0}(\mathbf{z}^r_{i,k,1})\|^2$$

$$+ 2\mathbb{E}\langle g(\bar{\mathbf{w}}^{r-1}, \mathbf{z}^r_{i,k,1}, \bar{\mathbf{w}}^{r-1}, \mathcal{S}_2) - \mathbf{u}^r_{i,k-1}(\mathbf{z}^r_{i,k,1}), \mathbf{u}^r_{i,k-1}(\mathbf{z}^r_{i,k,1}) - \mathbf{u}^{r-1}_{i,0}(\mathbf{z}^r_{i,k,1})\rangle$$

$$\leq -\frac{1}{2}\mathbb{E}\|g(\bar{\mathbf{w}}^{r-1}, \mathbf{z}^r_{i,k,1}, \bar{\mathbf{w}}^{r-1}, \mathcal{S}_2) - \mathbf{u}^r_{i,k-1}(\mathbf{z}^r_{i,k,1})\|^2 + 8\|\mathbf{u}^r_{i,k-1}(\mathbf{z}^r_{i,k,1}) - \mathbf{u}^{r-1}_{i,0}(\mathbf{z}^r_{i,k,1})\|^2$$

$$\leq -\frac{1}{2}\mathbb{E}\|g(\bar{\mathbf{w}}^{r-1}, \mathbf{z}^r_{i,k,1}, \bar{\mathbf{w}}^{r-1}, \mathcal{S}_2) - \mathbf{u}^r_{i,k-1}(\mathbf{z}^r_{i,k,1})\|^2 + 8\beta^2 K^2 C_0^2$$

$$\leq -\frac{1}{4}\mathbb{E}\|g(\bar{\mathbf{w}}^r_k, \mathbf{z}^r_{i,k,1}, \bar{\mathbf{w}}^r_k, \mathcal{S}_2) - \mathbf{u}^r_{i,k-1}(\mathbf{z}^r_{i,k,1})\|^2 + \frac{1}{2}\tilde{L}^2\|\bar{\mathbf{w}}^{r-1} - \bar{\mathbf{w}}^r_k\|^2 + 8\beta^2 K^2 C_0^2. \tag{105}$$

With the client sampling and data sampling, we observe that

$$-\mathbb{E}\left[\frac{1}{N}\sum_{i \in P^r}\frac{1}{|\mathcal{S}^i_1|}\|g(\bar{\mathbf{w}}^r_k, \mathbf{z}^r_{i,k,1}, \bar{\mathbf{w}}^r_k, \mathcal{S}_2) - \mathbf{u}^r_{i,k-1}(\mathbf{z}^r_{i,k,1})\|^2\right]$$

$$= -\frac{1}{N}\frac{|P^r|}{N}\sum_{i=1}^{N}\mathbb{E}_{\mathbf{z}^r_{i,k,1} \in \mathcal{S}^i_1}\left[\frac{1}{|\mathcal{S}^i_1|}\|g(\bar{\mathbf{w}}^r_k, \mathbf{z}^r_{i,k,1}, \bar{\mathbf{w}}^r_k, \mathcal{S}_2) - \mathbf{u}^r_{i,k-1}(\mathbf{z}^r_{i,k,1})\|^2\right]. \tag{106}$$

Then by multiplying $\gamma$ to every term and rearranging terms using the setting of $\gamma \leq O(1)$, we can obtain

$$\frac{\gamma+1}{2}\frac{1}{N}\sum_{i=1}^{N}\frac{1}{|\mathcal{S}^i_1|}\sum_{\mathbf{z} \in |\mathcal{S}^i_1|}\mathbb{E}\|\mathbf{u}^r_{i,k}(\mathbf{z}) - g(\bar{\mathbf{w}}^r_k, \mathbf{z}, \bar{\mathbf{w}}^r_k, \mathcal{S}_2)\|^2$$

$$\leq \frac{\gamma(1 - \frac{|P^r|}{8|\mathcal{S}^i_1|N}) + 1}{2}\frac{1}{N}\sum_{i=1}^{N}\frac{1}{|\mathcal{S}^i_1|}\sum_{\mathbf{z} \in |\mathcal{S}^i_1|}\mathbb{E}\|\mathbf{u}^r_{i,k-1}(\mathbf{z}) - g(\bar{\mathbf{w}}^r_k, \mathbf{z}, \bar{\mathbf{w}}^r_k, \mathcal{S}_2)\|^2$$

$$+ \frac{4\gamma^2|P^r|}{|\mathcal{S}^i_1|N}(\sigma^2 + C_0^2) + \frac{8\gamma\beta^2 K^2 C_0^2|P^r|}{|\mathcal{S}^i_1|N} + 4\tilde{L}^2\frac{|P^r|}{N}\mathbb{E}\|\bar{\mathbf{w}}^r - \bar{\mathbf{w}}^{r-1}\|^2 + 4\tilde{L}^2\frac{|P^r|}{N}\mathbb{E}\|\bar{\mathbf{w}}^r - \bar{\mathbf{w}}^r_k\|^2$$

$$+ 4(\gamma^2 + \frac{\gamma}{|\mathcal{S}^i_1|})\tilde{L}^2\frac{1}{N}\sum_{i \in P^r}\mathbb{E}\|\bar{\mathbf{w}}^r - \mathbf{w}^r_{i,k}\|^2 + (\gamma^2 + \frac{\gamma}{|\mathcal{S}^i_1|})\tilde{L}^2\frac{1}{NK}\sum_{i \in P^r}\sum_{k=1}^{K}\mathbb{E}\|\bar{\mathbf{w}}^{r-1} - \mathbf{w}^{r-1}_{i,k}\|^2. \tag{107}$$

Dividing $\frac{\gamma+1}{2}$ on both sides gives

$$\frac{1}{N}\sum_{i=1}^{N}\frac{1}{|\mathcal{S}^i_1|}\sum_{\mathbf{z} \in |\mathcal{S}^i_1|}\mathbb{E}\|\mathbf{u}^r_{i,k}(\mathbf{z}) - g(\bar{\mathbf{w}}^r_k, \mathbf{z}, \bar{\mathbf{w}}^r_k, \mathcal{S}_2)\|^2$$

$$\leq \frac{\gamma(1 - \frac{|P^r|}{8|\mathcal{S}^i_1|N}) + 1}{\gamma+1}\frac{1}{N}\sum_{i=1}^{N}\frac{1}{|\mathcal{S}^i_1|}\sum_{\mathbf{z} \in |\mathcal{S}^i_1|}\mathbb{E}\|\mathbf{u}^r_{i,k-1}(\mathbf{z}) - g(\bar{\mathbf{w}}^r_k, \mathbf{z}, \bar{\mathbf{w}}^r_k, \mathcal{S}_2)\|^2$$

$$+ 8\frac{\gamma^2|P^r|}{|\mathcal{S}^i_1|N}(\sigma^2 + C_0^2) + \frac{16\gamma\beta^2 K^2 C_0^2|P^r|}{|\mathcal{S}^i_1|N} + 8\tilde{L}^2\frac{|P^r|}{N}\|\bar{\mathbf{w}}^r - \bar{\mathbf{w}}^{r-1}\|^2 + 8\tilde{L}^2\frac{|P^r|}{N}\|\bar{\mathbf{w}}^r - \bar{\mathbf{w}}^r_k\|^2$$

$$+ 8(\gamma^2 + \frac{\gamma}{|\mathcal{S}^i_1|})\tilde{L}^2\frac{1}{N}\sum_{i \in P^r}\|\bar{\mathbf{w}}^r - \mathbf{w}^r_{i,k}\|^2 + 2(\gamma^2 + \frac{\gamma}{|\mathcal{S}^i_1|})\tilde{L}^2\frac{1}{NK}\sum_{i \in P^r}\sum_{k=1}^{K}\mathbb{E}\|\bar{\mathbf{w}}^{r-1} - \bar{\mathbf{w}}^{r-1}_{i,k}\|^2. \tag{108}$$

Using Young's inequality,

$$\frac{1}{N}\sum_{i=1}^{N}\frac{1}{|\mathcal{S}_1^i|}\sum_{\mathbf{z}\in|\mathcal{S}_1^i|}\mathbb{E}\|\mathbf{u}_{i,k}^r(\mathbf{z})-g(\bar{\mathbf{w}}_k^r,\mathbf{z},\bar{\mathbf{w}}_k^r,\mathcal{S}_2)\|^2$$

$$\leq (1-\frac{\gamma|P^r|}{8|\mathcal{S}_1^i|N})\frac{1}{N}\sum_{i=1}^{N}\frac{1}{|\mathcal{S}_1^i|}\sum_{\mathbf{z}\in|\mathcal{S}_1^i|}\left[(1+\frac{\gamma|P^r|}{16|\mathcal{S}_1^i|N})\mathbb{E}\|\mathbf{u}_{i,k-1}^r(\mathbf{z})-g(\bar{\mathbf{w}}_{k-1}^r,\mathbf{z},\bar{\mathbf{w}}_{k-1}^r,\mathcal{S}_2)\|^2\right.$$

$$\left.+(1+\frac{16|\mathcal{S}_1^i|N}{\gamma|P^r|})\tilde{L}^2\|\bar{\mathbf{w}}_{k-1}^r-\bar{\mathbf{w}}_k^r\|^2\right]$$

$$+8\frac{\gamma^2|P^r|}{|\mathcal{S}_1^i|N}(\sigma^2+C_0^2)+\frac{16\gamma\beta^2K^2C_0^2|P^r|}{|\mathcal{S}_1^i|N}+8\tilde{L}^2\frac{|P^r|}{N}\|\bar{\mathbf{w}}^r-\bar{\mathbf{w}}^{r-1}\|^2+8\tilde{L}^2\frac{|P^r|}{N}\|\bar{\mathbf{w}}^r-\bar{\mathbf{w}}_k^r\|^2$$

$$+8(\gamma^2+\frac{\gamma}{|\mathcal{S}_1^i|})\tilde{L}^2\frac{1}{N}\sum_{i\in P^r}\|\bar{\mathbf{w}}^r-\mathbf{w}_{i,k}^r\|^2+2(\gamma^2+\frac{\gamma}{|\mathcal{S}_1^i|})\tilde{L}^2\frac{1}{NK}\sum_{i\in P^r}\sum_{k=1}^{K}\mathbb{E}\|\bar{\mathbf{w}}^{r-1}-\bar{\mathbf{w}}_{i,k}^{r-1}\|^2$$

$$\leq (1-\frac{\gamma|P^r|}{16|\mathcal{S}_1^i|N})\frac{1}{N}\sum_{i=1}^{N}\frac{1}{|\mathcal{S}_1^i|}\sum_{\mathbf{z}\in|\mathcal{S}_1^i|}[\mathbb{E}\|\mathbf{u}_{i,k-1}^r(\mathbf{z})-g(\bar{\mathbf{w}}_{k-1}^r,\mathbf{z},\bar{\mathbf{w}}_{k-1}^r,\mathcal{S}_2)\|^2$$

$$+\frac{20|\mathcal{S}_1^i|N}{\gamma|P^r|}\tilde{L}^2\|\bar{\mathbf{w}}_{k-1}^r-\bar{\mathbf{w}}_k^r\|^2]+8\frac{\gamma^2}{|\mathcal{S}_1^i|}\frac{|P^r|}{N}(\sigma^2+C_0^2)+\frac{16\gamma\beta^2K^2C_0^2|P^r|}{|\mathcal{S}_1^i|N}$$

$$+8\frac{|P^r|}{N}\tilde{L}^2\|\bar{\mathbf{w}}^r-\bar{\mathbf{w}}^{r-1}\|^2+8\tilde{L}^2\frac{|P^r|}{N}\|\bar{\mathbf{w}}^r-\bar{\mathbf{w}}_k^r\|^2$$

$$+8(\gamma^2+\frac{\gamma}{|\mathcal{S}_1^i|})\tilde{L}^2\frac{1}{N}\sum_{i\in P^r}\|\bar{\mathbf{w}}^r-\mathbf{w}_{i,k}^r\|^2+2(\gamma^2+\frac{\gamma}{|\mathcal{S}_1^i|})\tilde{L}^2\frac{1}{NK}\sum_{i\in P^r}\sum_{k=1}^{K}\mathbb{E}\|\bar{\mathbf{w}}^{r-1}-\bar{\mathbf{w}}_{i,k}^{r-1}\|^2.$$

$$\square$$

## G.2 ANALYSIS OF THE ESTIMATOR OF GRADIENT

With update $G_{i,k}^r=(1-\beta)G_{i,k-1}^r+\beta(G_{i,k,1}^r+G_{i,k,2}^r)$, we define $\bar{G}_k^r:=\frac{1}{|P^r|}\sum_{i\in P^r}G_{i,k}^r$, and $\Delta_k^r:=\|\bar{G}_k^r-\nabla F(\bar{\mathbf{w}}_k^r)\|^2$. Then it follows that $\bar{G}_k^r=(1-\beta)\bar{G}_{k-1}^r+\beta\frac{1}{|P^r|}\sum_{i\in P^r}(G_{i,k,1}^r+G_{i,k,2}^r)$.

**Lemma 4.** *Under Assumption 2, Algorithm 3 ensures that*

$$\Delta_k^r\leq (1-\beta)\|\bar{G}_{k-1}^r-\nabla F(\bar{\mathbf{w}}_{k-1}^r)\|^2+\frac{\beta^2\sigma^2}{N}$$

$$+2\beta\left(\frac{1}{N}\sum_i 4\tilde{L}^2\mathbb{E}\|\mathbf{w}_{i,k}^r-\bar{\mathbf{w}}^r\|^2+4\tilde{L}^2\mathbb{E}\|\bar{\mathbf{w}}^r-\bar{\mathbf{w}}^{r-1}\|^2+\frac{1}{N}\sum_i 4\tilde{L}^2\mathbb{E}\|\mathbf{w}_{j',t'}^{r-1}-\bar{\mathbf{w}}^{r-1}\|^2\right)$$

$$+2\beta\frac{1}{N}\sum_i\left(\tilde{L}^2\mathbb{E}\|\mathbf{u}_{i,k}^r(\mathbf{z}_{i,k,1}^r)-g(\bar{\mathbf{w}}_k^r,\mathbf{z}_{i,k,1}^r,\bar{\mathbf{w}}_k^r,\mathcal{S}_2)\|^2\right.$$

$$\left.+\tilde{L}^2\mathbb{E}\|\mathbf{u}_{j',t'}^{r-1}(\hat{\mathbf{z}}_{j',t',1}^{r-1})-g(\bar{\mathbf{w}}_{t'}^{r-1},\hat{\mathbf{z}}_{j',t',1}^{r-1},\bar{\mathbf{w}}_{t'}^{r-1},\mathcal{S}_2)\|^2\right).$$

*Proof.*

$$\Delta_k^r = \|\bar{G}_k^r - \nabla F(\bar{\mathbf{w}}_k^r)\|^2$$

$$= \|(1-\beta)\bar{G}_{k-1}^r + \beta \frac{1}{|P^r|}\sum_{i\in P^r}(G_{i,k,1}^r + G_{i,k,2}^r) - \nabla F(\bar{\mathbf{w}}_k^r)\|^2$$

$$= \left\| (1-\beta)(\bar{G}_{k-1}^r - \nabla F(\bar{\mathbf{w}}_{k-1}^r)) + (1-\beta)(\nabla F(\bar{\mathbf{w}}_{k-1}^r) - \nabla F(\bar{\mathbf{w}}_k^r)) \right.$$

$$+ \beta\left(\frac{1}{|P^r|}\sum_{i\in P^r}(G_1(\mathbf{w}_{i,k}^r, \mathbf{z}_{i,k,1}^r, \mathbf{u}_{i,k}^r(\mathbf{z}_{i,k,1}^r), \mathbf{w}_{j,t}^{r-1}, \hat{\mathbf{z}}_{j,t,2}^{r-1}) + G_2(\mathbf{w}_{j',t'}^{r-1}, \hat{\mathbf{z}}_{j',t',1}^{r-1}, \mathbf{u}_{j',t'}^{r-1}(\hat{\mathbf{z}}_{j',t',1}^{r-1}), \mathbf{w}_{i,k}^r, \mathbf{z}_{i,k,2}^r))\right.$$

$$\left. - \frac{1}{P^r}\sum_{i\in P^r}(G_1(\bar{\mathbf{w}}^{r-1}, \mathbf{z}_{i,k,1}^r, g(\bar{\mathbf{w}}^{r-1}, \mathbf{z}_{i,k,1}^r, \bar{\mathbf{w}}^{r-1}, \mathcal{S}_2), \bar{\mathbf{w}}^{r-1}, \hat{\mathbf{z}}_{j,t,2}^{r-1})\right.$$

$$\left.\left. + G_2(\bar{\mathbf{w}}^{r-1}, \hat{\mathbf{z}}_{j',t',1}^{r-1}, g(\bar{\mathbf{w}}^{r-1}, \hat{\mathbf{z}}_{j',t',1}^{r-1}, \bar{\mathbf{w}}^{r-1}, \mathcal{S}_2), \bar{\mathbf{w}}^{r-1}, \mathbf{z}_{i,k,2}^r))\right)\right.$$

$$+ \beta\left(\frac{1}{|P^r|}\sum_{i\in P^r}(G_1(\bar{\mathbf{w}}^{r-1}, \mathbf{z}_{i,k,1}^r, g(\bar{\mathbf{w}}^{r-1}, \mathbf{z}_{i,k,1}^r, \bar{\mathbf{w}}^{r-1}, \mathcal{S}_2), \bar{\mathbf{w}}^{r-1}, \hat{\mathbf{z}}_{j,t,2}^{r-1})\right.$$

$$\left.\left. + G_2(\bar{\mathbf{w}}^{r-1}, \hat{\mathbf{z}}_{j',t',1}^{r-1}, g(\bar{\mathbf{w}}^{r-1}, \hat{\mathbf{z}}_{j',t',1}^{r-1}, \bar{\mathbf{w}}^{r-1}, \mathcal{S}_2), \bar{\mathbf{w}}^{r-1}, \mathbf{z}_{i,k,2}^r)) - \nabla F(\bar{\mathbf{w}}_k^r)\right)\right\|^2. \tag{109}$$

Using Young's inequality and $\tilde{L}$-Lipschtzness of $G_1, G_2$, we can then derive

$$\Delta_k^r \le (1+\beta)\left\| (1-\beta)(\bar{G}_{k-1}^r - \nabla F(\bar{\mathbf{w}}_{k-1}^r))\right.$$

$$+ \beta\left(\frac{1}{|P^r|}\sum_{i\in P^r}(G_1(\bar{\mathbf{w}}^{r-1}, \mathbf{z}_{i,k,1}^r, g(\bar{\mathbf{w}}^{r-1}, \mathbf{z}_{i,k,1}^r, \bar{\mathbf{w}}^{r-1}, \mathcal{S}_2), \bar{\mathbf{w}}^{r-1}, \hat{\mathbf{z}}_{j,t,2}^{r-1})\right.$$

$$\left.\left. + G_2(\bar{\mathbf{w}}^{r-1}, \hat{\mathbf{z}}_{j',t',1}^{r-1}, g(\bar{\mathbf{w}}^{r-1}, \hat{\mathbf{z}}_{j',t',1}^{r-1}, \bar{\mathbf{w}}^{r-1}, \mathcal{S}_2), \bar{\mathbf{w}}^{r-1}, \mathbf{z}_{i,k,2}^r)) - \nabla F(\bar{\mathbf{w}}^{r-1})\right)\right\|^2$$

$$+ (1+\frac{1}{\beta})\beta^2\left(\frac{1}{|P^r|}\sum_{i\in P^r}4\tilde{L}^2\mathbb{E}\|\mathbf{w}_{i,k}^r - \bar{\mathbf{w}}^r\|^2 + 4\tilde{L}^2\mathbb{E}\|\bar{\mathbf{w}}^r - \bar{\mathbf{w}}^{r-1}\|^2 + \frac{1}{N}\sum_i 4\tilde{L}^2\mathbb{E}\|\mathbf{w}_{j',t'}^{r-1} - \bar{\mathbf{w}}^{r-1}\|^2\right)$$

$$+ (1+\frac{1}{\beta})\beta^2\frac{1}{|P^r|}\sum_{i\in P^r}\left(\tilde{L}^2\mathbb{E}\|\mathbf{u}_{i,k}^r(\mathbf{z}_{i,k,1}^r) - g(\bar{\mathbf{w}}_k^r, \mathbf{z}_{i,k,1}^r, \bar{\mathbf{w}}_k^r, \mathcal{S}_2)\|^2\right.$$

$$\left. + \tilde{L}^2\mathbb{E}\|\mathbf{u}_{j',t'}^{r-1}(\hat{\mathbf{z}}_{j',t',1}^{r-1}) - g(\bar{\mathbf{w}}_{t'}^{r-1}, \hat{\mathbf{z}}_{j',t',1}^{r-1}, \bar{\mathbf{w}}_{t'}^{r-1}, \mathcal{S}_2)\|^2\right). \tag{110}$$

By the fact that

$$\mathbb{E}[\frac{1}{|P^r|}\sum_{i\in P^r}(G_1(\bar{\mathbf{w}}^{r-1}, \mathbf{z}_{i,k,1}^r, g(\bar{\mathbf{w}}^{r-1}, \mathbf{z}_{i,k,1}^r, \bar{\mathbf{w}}^{r-1}, \mathcal{S}_2), \bar{\mathbf{w}}^{r-1}, \hat{\mathbf{z}}_{j,t,2}^{r-1})$$

$$+ G_2(\bar{\mathbf{w}}^{r-1}, \hat{\mathbf{z}}_{j',t',1}^{r-1}, g(\bar{\mathbf{w}}^{r-1}, \hat{\mathbf{z}}_{j',t',1}^{r-1}, \bar{\mathbf{w}}^{r-1}, \mathcal{S}_2), \bar{\mathbf{w}}^{r-1}, \mathbf{z}_{i,k,2}^r)) - \nabla F(\bar{\mathbf{w}}^{r-1})] = 0, \tag{111}$$

and

$$\mathbb{E}\|\frac{1}{|P^r|}\sum_{i\in P^r}(G_1(\bar{\mathbf{w}}^{r-1}, \mathbf{z}_{i,k,1}^r, g(\bar{\mathbf{w}}^{r-1}, \mathbf{z}_{i,k,1}^r, \bar{\mathbf{w}}^{r-1}, \mathcal{S}_2), \bar{\mathbf{w}}^{r-1}, \hat{\mathbf{z}}_{j,t,2}^{r-1})$$

$$+ G_2(\bar{\mathbf{w}}^{r-1}, \hat{\mathbf{z}}_{j',t',1}^{r-1}, g(\bar{\mathbf{w}}^{r-1}, \hat{\mathbf{z}}_{j',t',1}^{r-1}, \bar{\mathbf{w}}^{r-1}, \mathcal{S}_2), \bar{\mathbf{w}}^{r-1}, \mathbf{z}_{i,k,2}^r)) - \nabla F(\bar{\mathbf{w}}^{r-1})\|^2 \le \frac{\sigma^2}{|P^r|} \tag{112}$$

we obtain

$$\Delta_k^r \leq (1-\beta)\|\bar{G}_{k-1}^r - \nabla F(\bar{\mathbf{w}}_{k-1}^r)\|^2 + \frac{\beta^2\sigma^2}{|P^r|}$$

$$+ 2\beta\left(\frac{1}{|P^r|}\sum_{i\in P^r} 4\tilde{L}^2\mathbb{E}\|\mathbf{w}_{i,k}^r - \bar{\mathbf{w}}^r\|^2 + 4\tilde{L}^2\mathbb{E}\|\bar{\mathbf{w}}^r - \bar{\mathbf{w}}^{r-1}\|^2 + \frac{1}{|P^r|}\sum_{i\in P^r} 4\tilde{L}^2\mathbb{E}\|\mathbf{w}_{j',t'}^{r-1} - \bar{\mathbf{w}}^{r-1}\|^2\right)$$

$$+ 2\beta\frac{1}{|P^r|}\sum_{i\in P^r}\left(\tilde{L}^2\mathbb{E}\|\mathbf{u}_{i,k}^r(\mathbf{z}_{i,k,1}^r) - g(\bar{\mathbf{w}}_k^r, \mathbf{z}_{i,k,1}^r, \bar{\mathbf{w}}_k^r, \mathcal{S}_2)\|^2\right.$$

$$\left. + \tilde{L}^2\mathbb{E}\|\mathbf{u}_{j',t'}^{r-1}(\hat{\mathbf{z}}_{j',t',1}^{r-1}) - g(\bar{\mathbf{w}}_{t'}^{r-1}, \hat{\mathbf{z}}_{j',t',1}^{r-1}, \bar{\mathbf{w}}_{t'}^{r-1}, \mathcal{S}_2)\|^2\right).$$

$\square$

## G.3 CONVERGENCE RESULT

**Theorem 7.** *Suppose Assumption 2 holds, and assume there are at least $|P|$ machines take participation in each round. Denoting $M = \max_i |\mathcal{S}_i^1|$ as the largest number of data on a single machine, by setting $\gamma = O(\frac{M^{1/3}N^{2/3}}{(R|P|)^{2/3}})$, $\beta = O(\frac{N^{2/3}}{M^{1/6}(R|P|)^{2/3}})$, $\eta = O(\frac{N^{2/3}}{M^{2/3}(R|P|)^{2/3}})$ and $K = O(\frac{M^{1/3}(R|P|)^{1/3}}{N^{1/3}})$, Algorithm 2 ensures that $\mathbb{E}\left[\frac{1}{R}\sum_{r=1}^R \|\nabla F(\bar{\mathbf{w}}^r)\|^2\right] \leq O(\frac{1}{R^{2/3}})$.*

*Proof.* By updating rules, we have that for $i \in P^r$,

$$\|\bar{\mathbf{w}}^r - \mathbf{w}_{i,k}^r\|^2 \leq \eta^2 K^2 C_f^2 C_\ell^2 C_g^2, \tag{113}$$

and

$$\|\bar{\mathbf{w}}_k^r - \bar{\mathbf{w}}^r\|^2 = \tilde{\eta}^2\|\frac{1}{|P^r|K}\sum_{i\in P^r}\sum_{m=1}^k \bar{G}_m^r\|^2 \leq \tilde{\eta}^2\frac{1}{K}\sum_{m=1}^K \|\bar{G}_m^r - \nabla F(\bar{\mathbf{w}}_m^r) + \nabla F(\bar{\mathbf{w}}_m^r)\|^2. \tag{114}$$

Similarly, we also have

$$\|\bar{\mathbf{w}}^{r-1} - \bar{\mathbf{w}}^r\|^2 = \tilde{\eta}^2\|\frac{1}{|P^r|K}\sum_{i\in P^r}\sum_{k=1}^K \bar{G}_k^{r-1}\|^2$$

$$\leq \tilde{\eta}^2\frac{1}{K}\sum_{k=1}^K \|\bar{G}_k^{r-1} - \nabla F(\bar{\mathbf{w}}_k^{r-1}) + \nabla F(\bar{\mathbf{w}}_k^{r-1})\|^2 \tag{115}$$

Lemma 4 yields that

$$\frac{1}{RK}\sum_{r,k}\mathbb{E}\|\bar{G}_k^r - \nabla F(\bar{\mathbf{w}}_k^r)\|^2 \leq \frac{\Delta_0^0}{\beta RK} + \frac{\beta\sigma^2}{|P^r|}$$

$$+ 2\left(\frac{1}{|P^r|}\sum_{i\in P^i} 4\tilde{L}^2\mathbb{E}\|\mathbf{w}_{i,k}^r - \bar{\mathbf{w}}^r\|^2 + 4\tilde{L}^2\mathbb{E}\|\bar{\mathbf{w}}^r - \bar{\mathbf{w}}^{r-1}\|^2 + \frac{1}{|P^r|}\sum_{i\in P^r} 4\tilde{L}^2\mathbb{E}\|\mathbf{w}_{j',t'}^{r-1} - \bar{\mathbf{w}}^{r-1}\|^2\right)$$

$$+ 2\mathbb{E}\left[\frac{1}{R}\sum_r\frac{1}{|P^r|K}\sum_{i\in P^r,k}\frac{1}{|\mathcal{S}_1^i|}\sum_{\mathbf{z}\in\mathcal{S}_1^i}\|\mathbf{u}_{i,k}^r(\mathbf{z}) - g(\bar{\mathbf{w}}^r, \mathbf{z}, \bar{\mathbf{w}}^r, \mathcal{S}_2)\|^2\right]$$

$$+ 2\mathbb{E}\left[\frac{1}{R}\sum_r\frac{1}{|P^r|K}\sum_{j',t'}\frac{1}{|\mathcal{S}_1^i|}\sum_{\mathbf{z}\in\mathcal{S}_1^i}\|\mathbf{u}_{j',t'}^{r-1}(\mathbf{z}) - g(\bar{\mathbf{w}}_{t'}^{r-1}, \mathbf{z}, \bar{\mathbf{w}}_{t'}^{r-1}, \mathcal{S}_2))\|^2\right], \tag{116}$$

which by setting of $\eta$ and $\beta$ leads to

$$\frac{1}{RK}\sum_{r,k}\mathbb{E}\|\bar{G}_k^r - \nabla F(\bar{\mathbf{w}}_k^r)\|^2 \leq \frac{2\Delta_0^0}{\beta RK} + \frac{4\beta\sigma^2}{|P|} + 10\beta\tilde{\eta}^2 C_\ell^2 C_g^2 + 2\tilde{\eta}^2\frac{1}{R}\sum_r\|\nabla F(\bar{\mathbf{w}}^{r-1})\|^2$$

$$+ 5\frac{1}{R}\sum_r\frac{1}{NK}\sum_{i,k}\frac{1}{|\mathcal{S}_1^i|}\sum_{\mathbf{z}\in\mathcal{S}_1^i}\mathbb{E}\|\mathbf{u}_{i,k}^r(\mathbf{z}) - g(\bar{\mathbf{w}}^r;\mathbf{z},\mathcal{S}_2)\|^2$$

$$+ 5\frac{1}{R}\sum_r\frac{1}{NK}\sum_{j',t'}\frac{1}{|\mathcal{S}_1^i|}\sum_{\mathbf{z}\in\mathcal{S}_1^i}\mathbb{E}\|\mathbf{u}_{j',t'}^{r-1}(\hat{\mathbf{z}}_{j',t',1}^{r-1}) - g(\bar{\mathbf{w}}^{r-1};\hat{\mathbf{z}}_{j',t',1}^{r-1},\mathcal{S}_2))\|^2$$

$$+ 5\frac{1}{R}\sum_r\frac{1}{K}\sum_{t'}\mathbb{E}\|\bar{\mathbf{w}}^{r-1} - \bar{\mathbf{w}}_{t'}^{r-1}\|^2.$$

Using Lemma 3 yields

$$\frac{1}{R}\sum_r\frac{1}{NK}\sum_{i=1}^N\sum_{k=1}^K\frac{1}{|\mathcal{S}_1^i|}\sum_{\mathbf{z}\in\mathcal{S}_1^i}\mathbb{E}\|\mathbf{u}_{i,k}^r(\mathbf{z}) - g(\bar{\mathbf{w}}_k^r,\mathbf{z},\bar{\mathbf{w}}_k^r,\mathcal{S}_2)\|^2$$

$$\leq \frac{16MN}{\gamma|P^r|}\frac{1}{R}\frac{1}{NK}\sum_{i=1}^N\frac{1}{|\mathcal{S}_1^i|}\sum_{\mathbf{z}\in\mathcal{S}_1^i}\mathbb{E}\|\mathbf{u}_{i,0}^0(\mathbf{z}) - g(\bar{\mathbf{w}}_0^0,\mathbf{z},\bar{\mathbf{w}}_0^0,\mathcal{S}_2)\|^2$$

$$+ \frac{400M^2N^2}{\gamma^2|P^r|^2}\frac{1}{RK}\sum_{r,k}\tilde{L}^2\|\bar{\mathbf{w}}_{k-1}^r - \bar{\mathbf{w}}_k^r\|^2 + 150\gamma(\sigma^2 + C_0^2) + 256\beta^2K^2C_0^2$$

$$+ 128\tilde{L}^2\frac{|\mathcal{S}_1^i|}{\gamma}(\|\bar{\mathbf{w}}^r - \bar{\mathbf{w}}^{r-1}\|^2 + \|\bar{\mathbf{w}}^r - \bar{\mathbf{w}}^{r-1}\|^2)$$

$$+ 150(\gamma|\mathcal{S}_1^i| + 1)\tilde{L}^2\frac{1}{N}\sum_i\|\bar{\mathbf{w}}^r - \mathbf{w}_{i,k}^r\|^2 + 32(\gamma|\mathcal{S}_1^i| + 1)\tilde{L}^2\frac{1}{NK}\sum_{i=1}^N\sum_{k=1}^K\mathbb{E}\|\bar{\mathbf{w}}^{r-1} - \bar{\mathbf{w}}_{i,k}^{r-1}\|^2.$$

Combining this with previous five inequalities and noting the parameters settings, we obtain

$$\frac{1}{R}\sum_r\frac{1}{NK}\sum_{i=1}^N\sum_{k=1}^K\frac{1}{|\mathcal{S}_1^i|}\sum_{\mathbf{z}\in\mathcal{S}_1^i}\mathbb{E}\|\mathbf{u}_{i,k}^r(\mathbf{z}) - g(\bar{\mathbf{w}}_k^r,\mathbf{z},\bar{\mathbf{w}}_k^r,\mathcal{S}_2)\|^2$$

$$\leq O\left(\frac{MN}{\gamma RK|P|} + \eta^2\frac{M^2N^2}{\gamma^2|P|^2} + \gamma + \beta^2K^2 + \frac{M}{\gamma}\tilde{\eta}^2(\frac{1}{\beta RK} + \frac{\beta}{|P|}) + \gamma M\eta^2K^2 + \frac{1}{R}\sum_r\tilde{\eta}^2\|\nabla F(\bar{\mathbf{w}}^{r-1})\|^2\right)$$

and

$$\frac{1}{RK}\sum_{r,k}\mathbb{E}\|\bar{G}_k^r - \nabla F(\bar{\mathbf{w}}_k^r)\|^2$$

$$\leq O\left(\frac{MN}{\gamma RK|P|} + \eta^2\frac{M^2N^2}{\gamma^2|P|^2} + \gamma + \beta^2K^2 + \frac{M}{\gamma}\tilde{\eta}^2(\frac{1}{\beta RK} + \frac{\beta}{|P|}) + \gamma M\eta^2K^2 + \frac{1}{R}\sum_r\tilde{\eta}^2\|\nabla F(\bar{\mathbf{w}}^{r-1})\|^2\right).$$

$$(117)$$

Then using the standard analysis of smooth function, we derive

$$
\begin{aligned}
F(\bar{\mathbf{w}}^{r+1}) - F(\bar{\mathbf{w}}^r) &\leq \nabla F(\bar{\mathbf{w}}^r)^\top (\bar{\mathbf{w}}^{r+1} - \bar{\mathbf{w}}^r) + \frac{\tilde{L}}{2}\|\bar{\mathbf{w}}^{r+1} - \bar{\mathbf{w}}^r\|^2 \\
&= -\tilde{\eta}\nabla F(\bar{\mathbf{w}}^r)^\top \left( \frac{1}{NK}\sum_i\sum_k G_{i,k}^r - \nabla F(\bar{\mathbf{w}}^r) + \nabla F(\bar{\mathbf{w}}^r) \right) + \frac{\tilde{L}}{2}\|\bar{\mathbf{w}}^{r+1} - \bar{\mathbf{w}}^r\|^2 \\
&= -\tilde{\eta}\|\nabla F(\bar{\mathbf{w}}^r)\|^2 + \frac{\tilde{\eta}}{2}\|\nabla F(\bar{\mathbf{w}}^r)\|^2 + \frac{\tilde{\eta}}{2}\|\frac{1}{NK}\sum_i\sum_k G_{i,k}^r - \nabla F(\bar{\mathbf{w}}^r)\|^2 \\
&\quad + \frac{\tilde{L}}{2}\|\bar{\mathbf{w}}^{r+1} - \bar{\mathbf{w}}^r\|^2 \\
&\leq -\frac{\tilde{\eta}}{2}\|\nabla F(\bar{\mathbf{w}}^r)\|^2 + \tilde{\eta}\|\frac{1}{NK}\sum_i\sum_k (G_{i,k}^r - \nabla F(\bar{\mathbf{w}}_k^r))\|^2 \\
&\quad + \tilde{\eta}\|\frac{1}{K}\sum_k (\nabla F(\bar{\mathbf{w}}_k^r) - \nabla F(\bar{\mathbf{w}}^r))\|^2 + \frac{\tilde{L}}{2}\|\bar{\mathbf{w}}^{r+1} - \bar{\mathbf{w}}^r\|^2 \\
&\leq -\frac{\tilde{\eta}}{2}\|\nabla F(\bar{\mathbf{w}}^r)\|^2 + \tilde{\eta}\frac{1}{K}\sum_k\|\frac{1}{N}\sum_i (G_{i,k}^r - \nabla F(\bar{\mathbf{w}}_k^r))\|^2 \\
&\quad + \tilde{\eta}\frac{\tilde{L}^2}{K}\sum_k \|\bar{\mathbf{w}}_k^r - \bar{\mathbf{w}}^r\|^2 + \frac{\tilde{L}}{2}\|\bar{\mathbf{w}}^{r+1} - \bar{\mathbf{w}}^r\|^2.
\end{aligned}
\tag{118}
$$

Combining with (117), (113), (114), and (115), we derive

$$
\frac{1}{R}\sum_r \mathbb{E}\|\nabla F(\bar{\mathbf{w}}^r)\|^2 \leq O\left( \frac{MN}{\gamma RK|P|} + \eta^2\frac{M^2}{\gamma^2} + \gamma + \beta^2 K^2 + \frac{M}{\gamma}\tilde{\eta}^2(\frac{1}{\beta RK} + \frac{\beta}{|P|}) + \gamma M\eta^2 K^2 \right).
$$

By setting parameters as in the theorem, we can conclude the proof. Further, to get $\frac{1}{R}\sum_r \mathbb{E}\|\nabla F(\bar{\mathbf{w}}^r)\|^2 \leq \epsilon^2$, we just need to set $\gamma = O(\epsilon^2)$, $\beta = O(\frac{\epsilon^2}{\sqrt{M}})$, $K = O(\frac{\sqrt{M}}{\epsilon})$, $\eta = O(\frac{\epsilon^2}{M})$, $R = O(\frac{N}{|P|}\frac{\sqrt{M}}{\epsilon^3})$.  $\square$

# H  STATISTICS OF DATASETS AND MORE EXPERIMENTS

The statistics of the datasets we use are listed in Table 4.

Table 4: Statistics of the Datasets

|  | # of Training Data | # of Validation Data | # of Testing Data |
|---|---|---|---|
| Cifar10 | 24000 | 10000 | 10000 |
| Cifar100 | 24000 | 10000 | 10000 |
| CheXpert | 190027 | 1000 | 202 |
| ChestMNIST | 78468 | 11219 | 22433 |

Here we show some experiment results to verify the effectiveness of the larger buffers, the analysis of which is given in Appendix E and F. We focus on the task of one way partial AUC maximization optimized by FedX2 algorithm as in the experiment section. Recall that with the larger buffers, we just need to keep the last $\tau$ rounds of communicated history in $\mathcal{B}_{i,1}, \mathcal{B}_{i,2}$ instead of the just keeping the previous one round's history. With large buffers, it would provide each machines with a larger pool to sample when computing local gradients. It would possibly help enhance the performance in local steps. In Figure 2, $\tau = 1$ denotes the Algorithm 3 while large $\tau$ refers to the algorithms with larger buffers. We can see that by keeping some larger $\tau$ can improve the performance. And we have further verified that FedX2 can tolerate to skip a big number of communications.

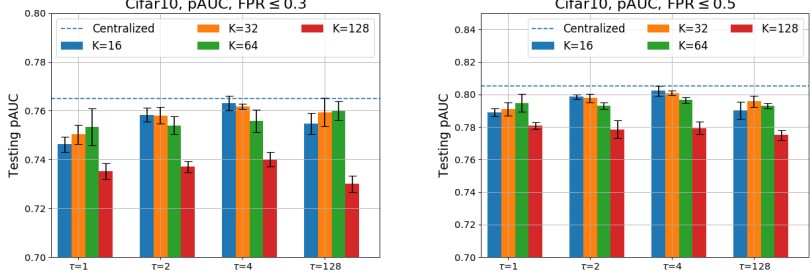

Figure 2: Fix $N$, Vary $K, \tau$

