# OpenReview forum: "FedX: Federated Learning for Compositional Pairwise Risk Optimization"
_ICLR.cc/2023/Conference — Submitted to ICLR 2023_

### Official Review · Reviewer_99hj · 2022-10-24

**Confidence:** 3
**Correctness:** 4
**Technical Novelty And Significance:** 1
**Empirical Novelty And Significance:** 2
**Recommendation:** 3

**Clarity, Quality, Novelty And Reproducibility:**

The novelty of this paper seems to be limited in the case that the authors didn't clearly specify their setting and the analysis is standard.

**Strength And Weaknesses:**

Strength:
1. This paper proposed two algorithms for federated CPR and established the best-known sample complexity results.

Weaknesses:
1.  The authors didn't clearly explain the distributed environment. In Page 2, the authors pointed out that a big challenge for CPR is that data are distributed on different machines and are prohibited to be moved to a central server. I don't think this would be a challenging point because most of the existing distributed algorithms do not ask the machines to share raw data, but instead they can share the sampled function value/gradient. It would be helpful if the authors could explain the challenges in distributed CPR more clearly.

2. In Page 4, the authors say that "It is not communication efficient that at each iteration all machines or a subset of machines sample data $z'$ and compute $h_w(z')$ and communicate them to all other machines in order to estimate/compute the blue terms above." However, as we can see from both FedX1 and FedX2 algorithms, these algorithms also require the machines share sampled stochastic information with other machines. For instance, FedX1 requires each machine to compute $H_{i,1}^r$ and $H_{i,2}^r$ and share them with the central machine in each round. This contradicts the above statement and makes me much confused. It would be helpful if the authors could explicitly explain what is allowed in this setting.

3. Absence of benchmark results: For the case when the outer function $f$ is nonlinear, the authors did not present the benchmark sample complexities for nondistributed CPR. Please provide the best-known non-distributed result so that I can understand the contributions of the distributed algorithm better.

4. The design of algorithms seems to be standard and techniques seem to be routine. FedX1 randomly samples the lazy parts from the previous round and FedX2 uses an additional weighted average technique, which has been well-understood in many ERM problems. These algorithms might be inefficient and lead to slow convergence.

5. In theorem 2, the authors set $K = O(M^{1/3} R^{1/3})$ where $M = \max_i |S_i|$ is the largest number of data on a single machine and $R$ is the total iteration. Following this strategy, because $| H_{i,2}^r | = K$,  in each round, the number of stochastic samples broadcasted to other machines may be huge if a high-level accuracy is required. This would cost huge memory and communications costs in practice and seems to be inefficient. What are the current best algorithms for handling nondistributed CPR? Is there any lower-bound result for the non-distributed case?

6. FedX2 requires an $O(M/\epsilon^2)$ per machine sample complexity, which grows linearly in $M$. Further, it seems that this result cannot be applied to the online streaming scenario because this scenario often does not require knowledge of the number of samples. In contrast, it looks like this result fits the finite sum case better where there are only a finite number of data. But the mathematical expression for the finite sum case is often different from the proposed one. Could the authors explain more about this point? Is it possible to develop more efficient algorithms for the finite-sum case?

7. The sample complexities seem to be not satisfactory. Both FedX1 and FedX2 algorithms require $O(1/\epsilon^4)$ samples to find an $\epsilon$-stationary point, which is indeed expensive compared with various complicated problems in the ERM setting having $O(1/\epsilon^2)$ sample complexities. I think a good survey of the current results in both distributed and nondistributed CPR and ERM is helpful.

8. In the numerical part, the authors use "iteration". Does it count as the product of round $r$ and communication interval $K$?

9. It would be helpful if the authors could provide some  applications of distributed CPR and write down their mathematical formulations to justify the importance and applicability of this problem.


**Summary Of The Paper:**

This paper studies distributed compositional pair risk optimization. The authors develop two algorithms to handle two different scenarios where the outer function $f$ is linear or nonlinear, and established their convergence rates. For linear $f$, they develop a FedX1 algorithm that achieves $O(1/N\epsilon^4)$ sample complexity on each machine. For nonlinear $f$, they design a FedX2 algorithm that achieves $O(M/\epsilon^4)$ sample complexity on each machine. The authors also conduct numerical experiments to test their algorithms.

**Summary Of The Review:**

This paper considers the distribution CPR problem. The main challenge is to design a communication-efficient algorithm in the distributed setting. However, the authors did not clearly explain their problem setting, especially which information can be communicated, making it hard to understand the challenges in algorithm design and evaluate the novelty of their algorithms. The FedX2 requires increasing per-iteration memory and communication costs if the algorithm targets obtaining a solution with high accuracy.  The technical analysis is standard The sample complexity results seem to be unsatisfactory compared with the existing vast works in ERM.

In summary, it looks to me that the key is to explain the problem setting and why it is hard to design efficient algorithms for CPR. The current contributions seem to be limited.

---

> ### Author Response · Authors · 2022-11-19
> **Author Response Part I (Q1-Q2)**
>
> Thank you for your time to review our submission. We carefully respond to your detailed review comments. We appreciate your precious time to read through.
>
> > **Q1:  In Page 2, the authors pointed out that a big challenge for CPR is that data are distributed on different machines and are prohibited to be moved to a central server.**
>
> We would like to first quote our original description in the following
>
> *"These differences pose a big challenge for optimizing CPR in the FL setting, where the training data are distributed on different machines and are prohibited to be moved to a central server. In particular, **the gradient of CPR cannot be written as the sum of local gradients at individual machines that only depend on the local data in those machines. Instead, the gradient of CPR at each machine not only depends on local data but also on data in other machines. As a result, the design of communication-efficient FL algorithms for optimizing CPR is much more complicated than that for ERM.** "*
>
> We did not mean that the big challenge for optimizing CPR is *``training data are distributed on different machines and are prohibited to be moved to a central server"*, which is used to explain the FL setting. We elaborate the the challenge in the sentences marked by the bold font. Please note that when we talk about the challenge of optimizing CPR in a FL setting, we are comparing with the modern FL for the standard empirical risk minimization, in which local gradient only depends on local data and does not depend on data in other machines. In addition, in the modern FL setting we do not like to communicate information (e.g., function value or gradients) at every iteration in order to reduce the communication costs. Hence, we have to consider balancing between skipping communications and using updated information for gradient estimator, which causes significant challenge of optimizing CPR in comparison with optimizing the traditional empirical risk. To the best of our knowledge, this is the first work that proposes  communication-efficient algorithm for the federated CPR problems with theoretical guarantee.
>
> > **Q2: Why is the proposed algorithm communication-efficient when it also requires the machines share sampled stochastic information with other machines**
>
> We would like to elaborate our original sentence *``It is not communication efficient that {\color{red}at each iteration} all machines or a subset of machines sample data $\mathbf{z}'$ and compute $h\_\mathbf{w}(\mathbf{z}')$ and communicate them to all other machines in order to estimate/compute the blue terms above."* What we emphasize here is {\color{red} not communicating at every iteration}.   When we consider communication costs, we should not just consider the bits of information to be shared, i.e., the number of bits for encoding $h\_\mathbf{w}(\mathbf{z})$, but also the significant overhead for communication (to establish and maintain  tcp/ip connection between different machines, to load data onto/off GPUs, to encode/decode the packet, to queue for network bandwidth, to retransmit the lost packets and so on). Roughly, we can decompose the each communication cost into two components, the communication time that directly depends on the number of bits of information to be communicated, and the communication overhead that is independent of the number of bits. Let us compare two communication models, i.e., the naive one  communicating $h\_\mathbf{w}(\mathbf{z}\_k)$ at every iteration for $k=1, \ldots, K$, and the proposed one that communicate $\{h\_\mathbf{w}(\mathbf{z}\_k), k=1, \ldots, K\}$ at once. The naive one has a total cost of $K C\_1(B) + K C\_o$, where $C\_1(B)$ denotes the communication time for sending $B$ bits of data (for $h(\mathbf{w}; \mathbf{z}\_k))$ and $C\_o$ denotes the communication overhead of one communication. In contrast, the proposed algorithm has a total cost of $C\_1(KB) + C\_o$. It is clear that the proposed algorithm could have much smaller communication costs. If we consider the possibility of failure of communications which could cause re-sending information,  the naive approach that communicates every iteration would increase the possibility of communication failure adding additional overhead. In addition, our algorithmic framework can allow for sending a subset of $K/N$ predictions among all $h(\mathbf{w}\_k, \mathbf{z}\_k), k=1, \ldots, K$. Please check the discussion on the bottom of Page 4. We have changed the original description to the following.
>
> ``*We would like to avoid communicating $h(\mathbf{w}; \mathbf{z}')$ at every iteration for estimating the blue terms as each communication would incur additional communication overhead.*"

---

> ### Author Response · Authors · 2022-11-19
> **Author Response Part II (Q3-Q6)**
>
> > **Q3: Absence of benchmark results when the outer function $f$ is non-linear**
>
> For nonlinear $f$, we have reported the state-of-the-art result in the remark under Theorem 2. The sample complexity is $\sum\_i |\mathcal{S}^1\_i|/\epsilon^4$, which is the best-known complexity for non-distributed CPR problem as established in Wang \& Yang (2022b). While under our algorithm, the sample complexity on each machine is $\max\_i |\mathcal{S}^1\_i|/\epsilon^4$.
> Consider the case when every machine has the same amount of data i.e., $ |\mathcal{S}^1\_i| = |\mathcal{S}^1\_j|$  $\forall i,j$, we have achieved a nice linear speed-up property.
>
> > **Q4: The design of algorithms seems to be standard and techniques seem to be routine**
>
> Although the design of algorithms seems straightforward, the differences from traditional FL for ERM caused significant challenges in terms of analysis. Please refer to response to Q1 of reviewer iaiV. We would like to repeat it here. As we stated in the introduction section, the standard analysis for FL of ERM only needs to ensure the gap error among local models to be bounded without harming the convergence of optimization. However, in our analysis of FedX, there is additional error on the lazy part. This is more challenging as the lazy part $h^{r-1}\_{1, \zeta}$ and $h^{r-1}\_{2,\xi}$ will cause the model parameter $\mathbf{w}^r\_{i, k}$ in the stochastic gradient estimator $G^{r}\_{i,k,1}$ and $G^r\_{i,k,2}$ to be **NOT independent** of  the randomness of data samples being used.  As a result, taking the expectation of stochastic gradient estimator given the current model parameter $\mathbf{w}^r\_{i,k}$ does not give the local gradient, which will fail the existing analysis  of FL. In order to handle this issue, we shift $\mathbf{w}^r\_{i,k}$ in $G^{r}\_{i,k,1}$ and $G^r\_{i,k,2}$ into $\bar{\mathbf{w}}^{r-1}$, which will incur **a latency error** $\|\bar{\mathbf{w}}^{r-1} - \bar{\mathbf{w}}^{r}\|^2$ and {\bf a gap error} among local models $\sum\_i \|\bar{\mathbf{w}}^r - \mathbf{w}^r\_{i,k}\|^2$. To the best of our knowledge, this technique is never used in the standard analysis of FL. We made this clear under Theorem 1 in the paper.  The analysis of FedX for non-linear $f$ is even more complicated as we also need to deal with the estimator error of local $\mathbf u(\mathbf{z})$ for estimating $g(\mathbf{w}; \mathbf{z}, \mathcal{S}\_2)$.
>
> > **Q5: The number of stochastic samples broadcast to other machines may be huge. Is there any lower bound for non-distributed CPR?**
>
> Please note that for many applications of interest (e.g., deep AUROC maximization, deep partial AUROC maximization), the prediction score $h(\mathbf{w}; \mathbf{z})\in\mathbb{R}$ is only a scalar. When we consider deep learning applications where the model $\mathbf{w}$ has tens of millions of parameters, the size of $K$ scalars for each communication compared with the size of model parameter $\mathbf{w}$  might not be a significant one. At the time of submission,  the best algorithm and result for solving non-distributed CPR are given in Wang \& Yang (2022b), which is already discussed in the remark under Theorem 2. Besides, there is currently no tight lower bound results for non-distributed CPR problem. We also notice that a better result is reported in a recent NeurIPS paper (Jiang et al. 2022), which proposes advanced variance reduction techniques for optimizing non-distribuetd CPR. A naive implementation of their algorithm will have even larger communication costs.
>
> Jiang et al. Multi-block-Single-probe Variance Reduced Estimator for Coupled Compositional Optimization. NeurIPS 2022.
>
> > **Q6: FedX2 requires an $O(M/\epsilon^2)$ per machine sample complexity, which grows linearly in $M$. It looks like this result fits the finite sum case better**
>
> Sorry for the confusion. The considered CPR problem is indeed the finite sum case for the outer summation, which is a special case of finite-sum coupled compositional optimization as mentioned in Page 1.  At the beginning of the Section 3, we have defined that $\mathbb{E}\_{\mathbf{z} \sim \mathcal{S}} = \frac{1}{|\mathcal{S}|} \sum\_{\mathbf{z} \sim \mathcal{S}}$ for the purpose of simplifying notations.

---

> ### Author Response · Authors · 2022-11-19
> **Author Response Part III (Q7-Q9)**
>
> > **Q7: The sample complexities seem to be not satisfactory. Both FedX1 and FedX2 algorithms require $O(1/\epsilon^4)$ samples to find an $\epsilon$-stationary point, which is indeed expensive compared with various complicated problems in the ERM setting having $O(1/\epsilon^2)$ sample complexities.**
>
> The $\epsilon$-stationary point is defined as $\mathbb{E}\|\nabla F(\mathbf{w})\|\leq \epsilon$. We would like to point out that the classic FL algorithms for ERM (e.g., FedAvg, FedProx) have the same complexity in terms of $\epsilon$, i.e., $O(1/\epsilon^4)$ (please refer to (Yu et. al., 2019b), and (Li et. al, 2020) for references). While we agree for finite-sum ERM problems, using variance-reduction techniques (e.g., SPIDER) one can reduce the sample complexity to $O(\sqrt{n}/\epsilon^2)$ for achieving $\epsilon$-stationary point in a non-distributed setting. However, even with some state-of-the-art techniques of FL for ERM, e.g., Scaffold (Karimireddy 2021), the sample complexity is still in the order of $O(1/\epsilon^4)$. It might be confusing that some existing literature (e.g., (Karimireddy 2021)) uses a different notion of $\epsilon$-stationary point, i.e.,  $\mathbb{E}\|\nabla F(\mathbf{w})\|^2\leq \epsilon$. Under this notion, our complexity is also $O(1/\epsilon^2)$. To the best of our knowledge, this is the first work considering distributed optimization of CPR.  We have provided more discussion of existing results of CPR and ERM in both distributed and non-distributed in the Appendix B of revision.
>
> > **Q8: Does the "iteration" count as the product of round $r$ and communication interval $R$?**
>
> Yes. It is the number of updates on each machine which is the product of round $R$ and communication interval $K$.
> Therefore, ``iteration" is also a notion for sample complexity, because a fixed-sized small batch of data is used in each iteration.
>
> > **Q9: Applications of the distributed CPR**
>
> Pleaser refer to response to reviewer gCnF.  We have provided more details of deep AUROC maximization, deep partial AUROC maximization in the Appendix A of revision.

---

### Official Review · Reviewer_gCnF · 2022-10-30

**Confidence:** 2
**Correctness:** 3
**Technical Novelty And Significance:** 3
**Empirical Novelty And Significance:** 3
**Recommendation:** 6

**Clarity, Quality, Novelty And Reproducibility:**

The main ideas in the paper are mostly clear, although the algorithms are a bit complicated, and their analysis seems quite involved. The authors had to introduce novel ideas to overcome technical obstacles.

**Strength And Weaknesses:**

Strengths:
- There are interesting ideas in the paper which could be useful for solving certain machine learning problems in distributed settings.


Weaknesses:
- The author did not do a very good job in motivating the setting in the sense of the types of problems it can be applicable to.

**Summary Of The Paper:**

This paper is about a Federated learning problem in which every input data point’s $z$ contribution to the objective is an arbitrary function $f$ or the sum of pairwise loss functions, each of which depends on $z$ and some other data point $z^\prime$. This is called compositional pairwise risk (CPR) optimization problem and is a generalization of the pairwise loss minimization problem, where $f$ is linear. Compared to the classical empirical risk minimization (ERM) problem, where the loss functions depend on a single data point, the challenge in the CPR setting is that the fact that data is distributed across different machines makes the computation hard to parallelize, which would not be the case with ERM. This makes the known centralized algorithms for CPR not directly adaptable in the federated learning setting by doing computation only with the data locally stored in each machine. As a result, in order to apply the proposed stochastic gradient descent algorithm, the terms of the gradient are split into “active” parts, which can be computed with local information within some machine, and “lazy” parts that depend on data stored across multiple machines. The latter terms are computed using “historical” data, which were communicated previously between the machines. As long as this “latency” error that is introduced along with the “gap” error from using local models versus the global model are comparable to the variance of the gradient estimator, similar convergence will be achieved with improved running time due to the parallelization.

**Summary Of The Review:**

Even though there are interesting and novel ideas in the paper, I believe there should be more discussion about the applicability of this particular setting, especially since it was introduced quite recently according to the related work section.

---

> ### Author Response · Authors · 2022-11-19
> **Thank you! Hope our revision and response can address your concern.**
>
> > **Q: The author did not do a very good job in motivating the setting in the sense of the types of problems it can be applicable to.**
>
> Thank you for acknowledging the novelty of our algorithms.  The compositional pairwise risks have been studied in several recent works as mentioned in the Introduction and Related Works, e.g., Qi et al. (2021), Wang et al. (2022).  Wang \& Yang (2022a),  Qiu et al. (2022), Zhu et al. (2022), Wang \& Yang (2022a), which include AUROC maximization, AUPRC maximization, NDCG maximization, partial AUC maximization. Below, we give details for two applications considered in our experiments, i.e., AUROC maximization and partial AUROC maximization, which has broad applications for handling imbalanced data.
>
> **AUROC maximization.** The area under ROC curve (AUROC) is defined
> (Hanley \& McNei, 1982) as
>     \begin{equation}
>         \text{AUROC}(\mathbf{w}) = \mathbb{E}[\mathbb{I}(h(\mathbf{w}, \mathbf{z}) \geq h(\mathbf{w}, \mathbf{z}'))|y=+1, y'=-1],
>     \end{equation}
>     where $\mathbf{z}, \mathbf{z}'$ are a pair of data features and $y,y'$ are the corresponding labels.
>     To maximize the AUROC,  we usually need to optimize the following pairwise surrogate loss:
>     \begin{equation}
>     \begin{split}
>         \min\limits\_{\mathbf{w}} \frac{1}{|\mathcal{S}\_1|}  \sum\limits\_{\mathbf{z}\_i\in S\_1} \frac{1}{|\mathcal{S}\_2|}
>          \sum\limits\_{\mathbf{z}\_j\in S\_2} \ell(h(\mathbf{w}, \mathbf{z}\_j) - h(\mathbf{w}, \mathbf{z}\_i)),
>     \end{split}
>     \end{equation}
>     where $\mathcal{S}\_1$ is the set of data with positive labels and $\mathcal{S}\_2$ is the set of data with negative labels. This is a CPR problem with a linear outer function.
>
> **Partial AUROC maximization.** In medical diagnosis,  high false positive rates (FPR) and low true positive rates (TPR) may cause a large cost. To alleviate this, we will also consider optimizing partial AUC (pAUC), which is the area under the ROC curve with a restriction of FRP less than some value. The recent work (Zhu et al. 2022)  proposes a DRO-based loss for maximizing pAUC with FPR  less than a small value, i.e.,
> \begin{equation}
>     \min\_{\mathbf{w}}\frac{1}{|\mathcal{S}\_1|}\sum\nolimits\_{\mathbf{z}\_i\in\mathcal{S}\_1} \lambda \log\frac{1}{|\mathcal{S}\_2|}\sum\nolimits\_{\mathbf{z}\_j\in\mathcal{S}\_2}\exp(\ell(h(\mathbf{w}; \mathbf{z}\_j) - h(\mathbf{w}; \mathbf{z}\_i))/\lambda),
>     \end{equation}
> where $\lambda$ is a hyperparameter. Hence,  it is CPR with non-linear $f$. We have provided the details of these two applications in the appendix.  For more applications, we refer the reviewer to a recent survey paper (Yang, 2022).
>
> Yang (2022). Algorithmic Foundation of Deep X-risk Optimization. https://arxiv.org/abs/2206.00439.

---

### Official Review · Reviewer_iaiV · 2022-10-31

**Confidence:** 4
**Correctness:** 4
**Technical Novelty And Significance:** 3
**Empirical Novelty And Significance:** 3
**Recommendation:** 6

**Clarity, Quality, Novelty And Reproducibility:**

This paper is clearly written and easy to follow. The problem studied in this paper is novel, although the proposed algorithms and analysis techniques are somewhat standard. The reproducibility of this work seems good.

**Strength And Weaknesses:**

Strengths:
1. Solving compositional pairwise risk minimization problems in a federated setting is a new and interesting problem.
2. The analysis of the "lazy" components necessitated by the communication restrictions in the federated setting is non-trivial.

Weaknesses:
1. Although the compositional pairwise risk minimization problem in the federated learning setting is new, the proposed FedX method appears to be a relatively straightforward extension of FedAvg-type algorithms to this new setting, and the novelty only comes from the complications arising from compositional pairwise risk function stochastic evaluations. Hence, the convergence analysis framework in this paper remains standard.

2. The local update number $K$ scales as $O(T^{-1/3}$, which shrinks as $T$ gets large and is unlike the $O(T)$ in state-of-the-art FL algorithms (see, e.g., [Yang et al., ICLR'21]. Is this necessary or just an artifact due to the proof technicality?

3. The proposed algorithms require full client participation, which is often infeasible in real-world FL systems. Thus, the applications of the proposed algorithms could be limited.

**Summary Of The Paper:**

This paper studied the problem of solving federated compositional pairwise risk minimization. This problem has several applications in AUROC maximization with a pairwise loss, partial AUROC maximization with a compositional loss, etc. The main challenge of solving this problem stems from the non-decomposability of the objective function and the interdependence between different machines. To address these challenges, the authors proposed two federated learning (FL) algorithms called FedX (X=1 or 2) for handling linear and nonlinear objective function $f$. The authors provided theoretical convergence analysis for the proposed algorithms and conducted experiments to verify their efficacies.

**Summary Of The Review:**

This paper studied the problem of solving compositional pairwise risk minimization in the federated learning setting. The authors proposed two algorithms to handle linear and nonlinear objective functions. The theoretical convergence results are solid and provide interesting insights, although the algorithms and proof techniques are somewhat standard.

---

> ### Author Response · Authors · 2022-11-19
> **Thank you for your comments! Here are our responses.**
>
>
> Thank you for your efforts to review this submission! We appreciate your favor of the significance and technical novelty of this work. Regarding your concerns of the several weaknesses, we address them as below.
>
> > **Q1: The convergence analysis framework in this paper remains standard.**
>
> We humbly disagree with your comment about the convergence analysis being standard. But we acknowledge that our analysis is built on existing analysis of FL. However, there are several key differences that cause challenges in the analysis.   As we stated in the introduction section, the standard analysis for FL of ERM only needs to ensure the gap error among local models to be bounded without harming the convergence of optimization. However, in our analysis of FedX, there is additional error on the lazy part. This is more challenging as the lazy part $h^{r-1}\_{1, \zeta}$ and $h^{r-1}\_{2,\xi}$ will cause the model parameter $\mathbf{w}^r\_{i, k}$ in the stochastic gradient estimator $G^{r}\_{i,k,1}$ and $G^r\_{i,k,2}$ to be **NOT independent** of  the randomness of data samples being used.  As a result, taking the expectation of stochastic gradient estimator given the current model parameter $\mathbf{w}^r\_{i,k}$ does not give the local gradient, which will fail the existing analysis  of FL. In order to handle this issue, we shift $\mathbf{w}^r\_{i,k}$ in $G^{r}\_{i,k,1}$ and $G^r\_{i,k,2}$ into $\bar{\mathbf{w}}^{r-1}$, which will incur **a latency error** $\|\bar{\mathbf{w}}^{r-1} - \bar{\mathbf{w}}^{r}\|^2$ and **a gap error** among local models $\sum\_i \|\bar{\mathbf{w}}^r - \mathbf{w}^r\_{i,k}\|^2$. To the best of our knowledge, this technique is never used in the standard analysis of FL. We made this clear under Theorem 1 in the paper.  The analysis of FedX for non-linear $f$ is even more complicated as we also need to deal with the estimator error of local $\mathbf u(\mathbf{z})$ for estimating $g(\mathbf{w}; \mathbf{z}, \mathcal{S}\_2)$. Please let us know if  any of these analysis are used in  the standard FL analysis.
>
> > **Q2: The local update number $K$ shrinks as $T$ gets larger.**
>
> We are sorry that there is  a typo in Theorem 1. The number of local updates between two communications $K$ should be $O(R^{1/3}/N)$, where $R$ is the number of communication rounds. Note that in the remark under Theorem 1, we have the correct order for $K$.  We have fixed this in Theorem 1 in the revision.
>
> > **Q3: The proposed algorithms require full client participation.**
>
> Our framework can be easily extended to allow partial client participation. We have provided the algorithm and analysis in the  Appendix G of  revision.

---

### Decision · Program_Chairs · 2023-01-20

**Decision:**

Reject

**Justification For Why Not Higher Score:**

The motivation for the problem is somewhat lacking, and the algorithm and analysis - while new - are somewhat standard.

**Justification For Why Not Lower Score:**

N/A

**Metareview: Summary, Strengths And Weaknesses:**

Summary: this paper considers FL for the problem of minimizing empirical risk where each term in the sum depends on two datapoints (instead of the customary one datapoint). When datapoints are spread out over multiple clients, this results in a coupled problem where stochastic gradients (i.e. gradient of the loss on a single pair) may not be locally computable because one of the datapoints may not be local. The paper proposes a way to locally approximate (with delay) the aggregate effect of all datapoints, and use that.

Strengths: The paper seems rigorously written

Weaknesses:
1. Lack of compelling motivation. For many of the problems mentioned (e.g. bipartite ranking or distance metric learning) there are problem-dependent approximate solutions available, or the ERM sum is not over all pairs in the data but rather each datapoint is paired with only a few others.
2. Lack of novelty: the algorithm is fairly obvious. the analysis is less obvious, but even there it follows known outlines in FL optimization